# Extreme floods of Venice: characteristics, dynamics, past and future evolution (review article)

Piero Lionello[1], David Barriopedro[2], Christian Ferrarin[3], Robert J. Nicholls[5], Mirko Orlić[4], Fabio Raicich[6], Marco Reale[7], Georg Umgiesser[3,10], Michalis Vousdoukas[8], Davide Zanchettin[9]

[1]University of Salento, DiSTeBA - Department of Biological and Environmental Sciences and Technologies, via per Monteroni, 165, Lecce, Italy and EuroMediterranean Center on Climate Change

[2]Instituto de Geociencias (IGEO), CSIC-UCM, C/Doctor Severo Ochoa 7, 28040 Madrid, Spain

[3]CNR - National Research Council of Italy, ISMAR - Marine Sciences Institute, Castello 2737/F, 30122, Venezia, Italy

[4]Department of Geophysics, Faculty of Science, University of Zagreb, Croatia

[5]Tyndall Centre for Climate Change Research, University of East Anglia. Norwich NR4 7TJ, United Kingdom

[6]CNR, Institute of Marine Sciences, AREA Science Park Q2 bldg., SS14 km 163.5, Basovizza, 34149 Trieste, Italy

[7] National Institute of Oceanography and Applied Geophysics – OGS and Abdus Salam ICTP, via Beirut 2-4, Trieste, Italy

[8]European Commission, Joint Research Centre (JRC), Ispra, Italy

[9]University Ca' Foscari of Venice, Dept. of Environmental Sciences, Informatics and Statistics, Via Torino 155, 30172 Mestre, Italy

[10] Marine Research Institute, Klaipeda University, Klaipeda, Lithuania

*Correspondence to*: Piero Lionello (piero.lionello@unisalento.it)

**Abstract**

Floods in the Venice city centre result from the superposition of several factors: astronomical tides, seiches and atmospherically-forced fluctuations, which include storm surges, meteotsunamis, and surges caused by atmospheric planetary waves. All these factors can contribute to positive water-height anomalies individually and can increase the probability of extreme events when they act constructively. The largest extreme water heights are mostly caused by the storm surges produced by the Sirocco winds, leading to a characteristic seasonal cycle, with the largest and most frequent events occurring from November to March. Storm surges can be produced by cyclones whose centres are located either north or south of the Alps. Historically, the most intense events have been produced by cyclogenesis in the western Mediterranean, to the west of the main cyclogenetic area of the Mediterranean region in the Gulf of Genoa. Only a small fraction of the interannual variability of extreme water heights is described by fluctuations in the dominant patterns of atmospheric circulation variability over the Euro-Atlantic sector. Therefore, decadal fluctuations of water-height extremes remain largely unexplained. In particular, the effect of the 11-year solar cycle does not appear to be steadily present if more than one hundred years of observations are considered. The historic increase in the frequency of floods since the mid-19[th] century is explained by relative mean sea-level rise. Analogously, future regional relative mean sea-level rise will be the most important driver of increasing duration and intensity of Venice floods through this century, overcompensating for the small projected decrease of marine storminess. The future increase of extreme water heights covers a wide range, largely reflecting the highly uncertain mass contributions to future mean sea-level rise from the melting of Antarctica and Greenland ice-sheets, especially towards the end of the century. For a high emission scenario (RCP8.5), the magnitude of 1-in-100 year water-height values at the North Adriatic coast is projected to increase by 26-

35 cm in 2050 and by 53-171 cm in 2100, with respect to the present value, and subject to continued increase thereafter. For a moderate emission scenario (RCP4.5), these values are 12-17cm in 2050 and 24-56cm in 2100. Local subsidence (which is not included in these estimates) will further contribute to the future increase of extreme water heights. This analysis shows the need for adaptive long term planning of coastal defenses using flexible solutions that are appropriate across the large range of plausible future water-height extremes.

**key words**: Venice, extreme events, floods, relative sea level rise, surges, climate change, trends

## 1.  Introduction

This paper reviews current understanding of the factors that are responsible for the damaging floods affecting the Venice city centre and for their future evolution. The event of 4th November 1966, with estimated damages of 400 million euros (De Zolt *et al.*, 2006), and of 12 November 2019 (Cavaleri *et al.*, 2020), with estimated damages of 460 million euros[1] and extensive global media coverage, highlight the risks that future extreme floods bring. Potential damages have often been linked to future relative sea level (RSL) rise. Costs of 7 billion euros have been estimated by the mid of this century if RSL rise continues at the rate observed in the 20th century (an unrealistic scenario based on recent trends and model projections) and can reach 8 and 16 billion euros for severe and high-end RSL rise scenarios, respectively (Caporin and Fontini, 2016). These estimates ignore adaptation options, but  show the large exposure and the values at stake. In order to prevent damages and losses of a unique monumental and cultural heritage, in 1994 the Italian government approved the construction of a system of mobile barriers (MoSE, Modulo Sperimentale Elettromeccanico) to prevent the flooding of Venice. MoSE's construction was initiated in 2003 and it has been successfully tested in October 2020.

The understanding of the dynamics leading to extreme floods and of the future evolution of their height and frequency is of paramount importance for a realistic assessment of present and future risks. This information is needed for efficient management of the implemented defence systems (see also Umgiesser et al., 2021, in this special issue), the assessment of their effectiveness in  from a climate change perspective and the development of new strategies to cope with future scenarios (see also Zanchettin et al., 2020, and Lionello et al., 2020 in this special issue).

The city of Venice is located in the centre of a large and shallow lagoon (Fig. 1), covering 500 km$^2$ with an average depth of about 1 m. Water is exchanged between the lagoon and the open sea  through three inlets (500-1000 m wide and from 8 to17 m deep) and it propagates to the city centre along a complex pattern of very shallow areas and canals (from 2 to 20 m deep). The lagoon is separated from the sea by two long (about 25 km in total) narrow (less than 200 m average width) sandy barrier islands, reinforced with artificial defences in the most vulnerable parts. The elevation of these islands is such that they are not submerged during the most extreme events, with the exception of the 4th November 1966 flood, when they were breached at several points.

A clear relationship exists between the frequency of floods and RSL rise, resulting from the superposition of vertical land motion (at multiple space and time scale) and mean sea level (MSL) rise, which is projected to greatly increase flood risks in the future (e.g. Lionello et al, 2021). The RSL rise issue is extensively discussed in a complementary review article in this special issue (Zanchettin et al., 2021), to which the interested reader is addressed for detailed information. However, RSL rise is not the only factor playing a role in flooding. Section 2.1 provides a general framework for the identification of the different factors acting at different time scales. Floods are caused by weekly to hourly atmospheric

---

[1] https://repository.tudelft.nl/islandora/object/uuid:ea34a719-79c1-4c6e-b886-e0d92407bc9d?collection=education

forcing, affected by long-term (seasonal to decadal) variability, and intensified by the long-term (multidecadal to centennial) RSL rise (section 2.1). The timing of the surges produced by the atmospheric forcing with respect to the phase of astronomical tide and free oscillations (seiches) can substantially affect floods (sections 2.1 and 2.2). In fact, the length of the basin (Fig. 1) and the average speed of barotropic shallow water waves combine in such a way that the period of the free oscillations is close to the diurnal and semidiurnal components. Therefore, the basin is close to resonant conditions and the North Adriatic has an astronomical tidal range of about 1 m at the northern end of the basin, which is relevant for the floods of Venice. The combination of all these forcings largely explains the historical floods, which are to some extent heterogeneous in terms of the leading factors (see section 2.2 and appendix A).

Storm surges, which are particularly important because they often produce the largest contribution to the floods, are caused by cyclones (see section 3.1). An important characteristic of the Adriatic Sea (particularly its northern area) is its proximity to the main cyclogenesis area of the western Mediterranean Sea, where cyclones initiate their south-eastward propagation along the Mediterranean storm track and (in a small number of cases) towards central Europe (e.g.. Lionello *et al.*, 2016). In autumn and winter, the area around the Adriatic Sea is frequently crossed by these cyclones. The resulting south-easterly wind (Sirocco) when channelled along the main axis of the basin by the action of the Apennines and Dinaric Alps is essential for producing the storm surge in the northern Adriatic Sea and floods of Venice. At longer time scales, the frequency and/or severity of extreme water heights have also been associated with large-scale atmospheric variability and astronomical (solar) forcing. Available evidence of these links and their dynamics is reviewed in sections 3.2 and 3.3.

A major concern is the future evolution of floods. Section 4 is devoted to past and future changes in the frequency and magnitude of extremes, and the relative roles of RSL rise and atmospheric forcing at different time scales. Section 4 also considers the most recent estimates of the future extremes and their dependence on the climate scenarios. The last section 5 provides a general assessment of the existing knowledge as well as indications of major gaps and needs for future research.

## 2. Dynamics and characteristics of extreme floods

Extreme floods of Venice are caused by extremes or high-end values of the local instantaneous thickness of the ocean, hereafter called water height. The water height is defined as the difference between the instantaneous sea level and a local reference level, both measured with respect to a fixed reference level (which could be the reference ellipsoid, the geoid, or a geocentric reference frame). In Venice, the local reference level moves vertically, because of land subsidence. The water height and the total thickness of the water column differ by a constant value, which is the depth of the sea bottom with respect to the local reference level (Fig. 2). Water height extremes and sea level extremes differ because the latter do not consider the effect of subsidence, which is important in Venice. Water height extremes result from contributions with different time scales and characteristics that are described in the next subsection.

### 2.1. Tides, seiches and atmospherically forced sea level anomalies

This section describes the factors that contribute to water-height fluctuations in the North Adriatic Sea: astronomical tides, seiches, atmospherically forced fluctuations, which consist of meteotsunamis, storm surges and surges caused by planetary atmospheric waves (PAW), interdecadal to seasonal (IDAS) sea level variations  and RSL rise. These factors are characterized by different dynamics and time scales. In general, they do not have the same importance in terms of

contribution to extreme water heights, which have mostly been attributed to large storm surges, whose effect can be reinforced or attenuated by the remaining factors. The classification of the atmospherically forced fluctuations in three categories is based on the scale of the meteorological forcing process: mesoscale for meteotsunamis, synoptic scale for storm surges and planetary scale for PAW surges (see section 2.2). At longer time scales, inter-decadal, inter-annual and seasonal (IDAS) sea level variability and local RSL rise also contribute to water-height extremes. Local RSL rise is the increase of local sea level relative to the local solid earth surface (Fig. 2 and Gregory et al., 2019) and it can be directly estimated by averaging local tide gauge data over a conveniently long period. It is caused by vertical land movements and changes of local MSL. The evolution of the RSL and IDAS in Venice, and of their different contributions is described in Zanchettin et al. (2021) in this special issue. The addition of storm surges, meteotsunamis and PAW surges represents the meteorological surge contribution to the water-height anomalies. The combined effect of seiches, astronomical tides and meteorological surge is generically referred to as detrended water height in this manuscript (Fig. 2), meaning that variability at seasonal and longer time scale is subtracted.

*Astronomical tides* in the Adriatic Sea have a mixed semidiurnal cycle with two high and two low tide levels of different height every day. There are 7 components with amplitude above 1 cm and only 3 above 10 cm, with the semidiurnal $M_2$ and $S_2$, and diurnal $K_1$ tides providing the largest contributions. The values of $M_2$, $S_2$ and $K_1$ are approximately 23, 14, 16 cm both outside the lagoon and in the Venice city centre. Tides consist of two Kelvin waves oppositely travelling along the basin at semidiurnal periods (Hendershott and Speranza, 1971) and of topographic waves travelling across the basin at diurnal periods (Malačičet al., 2000). They are adequately reproduced by a number of numerical models (e.g. Janeković and Kuzmić, 2005; Lionello et al., 2005; Ferrarin et al., 2017). Both diurnal and semidiurnal components have their maximum amplitude at the northern shore of the basin, in association to antinodes of seiches (see below). The semidiurnal components have an amphidromic point in the centre of the Adriatic (Franco *et al.*, 1982).

*Storm surges* in the Adriatic have been extensively studied due to the need to forecast the floods of Venice (Robinsonet al., 1973, Umgiesser *et al.*, 2021 in this issue for a review). The storm surge magnitude at the Venetian coast is mostly determined by the wind blowing over the shallow water areas over the North Adriatic Sea, whose contribution at the coast is typically 10 times larger than the inverse barometer effect (Bargagli *et al.*, 2002; Conte and Lionello, 2013; Lionello et al., 2019). Storm surges are produced by two main wind configurations: Sirocco blowing over the whole basin and a combination of Bora over the north Adriatic and Sirocco over the south Adriatic (Fig. 1). Depending on the structure of the wind field, flooding is more pronounced along the west or the east Adriatic coast (Međugorac *et al.*, 2018).

*Seiches* in the Adriatic are standing waves with a node at the southern boundary of the basin and an antinode at the northern shore. The periods of the basic modes are estimated at about 21.3 h and 10.8 h (Manca et al., 1974) and their patterns mimic those of the diurnal and semidiurnal tides, respectively. Seiches are commonly produced after a storm surge, when the wind drops or switches from Sirocco (south-easterly) to Bora (north-easterly) and the water accumulation in the north Adriatic ceases to be supported by the wind stress. The Adriatic seiches are slowly damped with the decay time of fundamental mode amounting to 3.2 days (Cerovečki et al., 1997), due to a weak frictional dissipation inside the basin and a small energy loss to the Mediterranean Sea. There is a long tradition of numerical modelling of the Adriatic seiches (e.g. Lionello et al., 2005), but more accurate predictions of their periods and decay are still needed, e.g., (Bajo *et al.*, 2019).

*Meteotsunamis* are meteorologically-generated long ocean waves in the tsunami frequency band (Vilibić and Šepić, 2009; Šepić et al., 2009). They are generated by mesoscale atmospheric pressure disturbances that resonantly generate a

traveling sea level anomaly, when their speeds of propagation approach that of the shallow-water barotropic waves. Adriatic meteotsunamis pose a major hazard on the eastern Adriatic coast, where their resonant periods are close to those of the normal modes of the bays/harbors.

*Long planetary atmospheric waves* propagate slowly and with wavelengths ranging from 6000 to 8000 km. They produce a long-term meteorological forcing and eventually long-lasting sea level anomalies (*PAW surges*), which establish favourable background conditions for flooding (Pasarić and Orlić, 2001).

The factors considered so far allow an interpretation of a typical flood ("aqua alta", e.g. Robinson *et al*., 1973). When a cyclone moves from western Mediterranean towards the Adriatic, low atmospheric pressure and Sirocco wind support an increase of water height in the northern Adriatic Sea and potential flooding of the area. When the cyclone leaves the Adriatic area, atmospheric pressure increases while the Sirocco slackens or changes to Bora. Consequently, sea level decreases and seiches may be generated. Therefore, in the Adriatic storm surges and seiches represent two distinct phases of the response to the atmospheric forcing, one in which sea level rises under direct atmospheric forcing, and the other in which sea level relaxes – possibly through a series of damped oscillations. If a successive storm surge develops before the attenuation of the seiches induced by a previous event, a constructive or destructive superposition may occur (Bajo *et al.*, 2019). Analogously, the phase of tide during the period when the storm surge is large, can substantially increase or decrease the actual sea level maximum. The contribution of meteotsunamis and PAW surges to extreme sea-level events in Venice has not been thoroughly investigated to date. However, the recent 12 November 2019 event uncovered their important role in flooding in Venice (Ferrarin et al., 2021). Therefore, in general, the hazard and probability of an extreme sea level should also include these two contributions (see section 2.2).

Water-height values are further modulated by IDAS sea-level variability (caused by changes in marine circulation, characteristics of the water masses and by the action of teleconnection patterns) and RSL changes. *RSL changes* represent a long-term process and RSL rise has been the dominant factor responsible for the significant increase of frequency of floods of the Venice city centre (Lionello *et al*., 2012b). In the last century, it has been caused almost equally by the increase of the mean level of the sea surface and the decrease of the land level because of natural and anthropogenic subsidence (see Zanchettin *et al*. 2021, in this special issue, for a comprehensive review of its past and future evolution).

The floods of Venice do not occur because water overtops coastal barriers or defences. In fact, the elevation of the natural barriers separating the lagoon from the Adriatic Sea has so far prevented wave overtopping, with the unique exception (already mentioned in the introduction) of the 1966 flood, when waves may have contributed to increase water height in the lagoon. Therefore, wave run-up and infra-gravity waves and nearshore processes (though certainly relevant along the sea-side front of the barrier islands under some conditions) have never been considered when computing water-height extremes inside the lagoon. It cannot be excluded that these factors will become relevant under extreme sea level rise in the future, but present evidence is that waves do not need to be considered for computing the water height in the Venice city centre (Roland et al., 2009), as long as barrier islands continue being protected by coastal defences and maintained by beach nourishment. It is well known that tidal and non-tidal components have a certain degree of interaction in shallow water areas with large tidal excursions where non-linear effects are significant (e.g Horsburgh and Wilson, 2007, and references therein). However, in a recent global scale investigation on the non-linear interactions between the tide and non-tidal residuals (Arns et al., 2020) only a small negative effect on extreme sea levels in the northern Adriatic Sea has been found. In fact, in the northern Adriatic Sea, given the relatively small importance of tidal excursions (about 1 m) compared to the local water depth (average depth of about 35 m), it is reasonable to neglect the effect of tides on the

storm surge propagation. Such assumption is confirmed by a long past prediction practice with hydrodynamic models, where only the meteorological forcing was used and the astronomical tide was either added to the model results to get the actual prediction or subtracted to the observations for model validation. Examples of this approach and of its success are Lionello et al. (2006), Bajo et al. (2007), Mel and Lionello (2014) among many others. Such an assumption has been confirmed by several high-resolution numerical studies demonstrating that tide-meteorological surge interactions are small, even during the most severe events (Roland et al., 2009; Cavaleri et al., 2019). An example of such simulations can be found in Appendix B and it shows that nonlinear interactions are lower than 5% at the peak of the water height.

### 2.2. A description of the largest past events

Regular tide gauge observations in Venice started in 1871. Since 1919 observations have been referenced to their mean level over the 1884-1909 period (central year 1897), which is the local reference level used for water-height values and is usually called 'Zero Mareografico Punta Salute' (ZMPS). The history of Venice tide gauges, their reference planes and the related geodetic connections have been described and discussed by Dorigo (1961a). Battistin and Canestrelli (2006) reviewed the observations from 1872 to 2004 and provided a complete list of daily maxima and minima with the relevant primary data sources. Tide gauge data are also available in the web sites of Istituto Superiore per la Protezione e la Ricerca Ambientale (ISPRA), Servizio Laguna di Venezia (www.venezia.isprambiente.it), and Centro Previsione e Segnalazione Maree of the Venice municipality (www.comune.venezia.it/content/centro-previsione-e-segnalazione.maree).

The Venice Municipality defines large events ('aqua alta') when the water height exceeds 80 cm, severe and exceptional events when it exceeds 110 cm and 140 cm, respectively. Since 1872, there have been 18 exceptional events. Depending on the phase of the astronomical tide and of other factors, large water heights can or cannot correspond to very high storm surges. Table 1 list the largest water heights[2] alongside the contributions of various factors (in a similar way as previously done by Orlić, 2001):

WATER HEIGHT = STORM SURGE + PAW SURGE + METEOTSUNAMI and MAV setup+ ASTRONOMICAL TIDE + SEICHE + IDAS Variability + RSL

In order to compute the values in Table 1, the long-term time series of Punta della Salute was processed with a tidal harmonic analysis tool based on the least squares fitting (Codiga, 2011) to separate the tidal from the other contributions. The residuals were detrended using a 10-year centred running mean to determine the RSL rise.

The contributions of storm surge, PAW surge, meteotsunamis and mesoscale atmospheric variability (MAV), seiches, IDAS variability have been estimated using band-pass digital filters in the time domain assuming Fourier decomposition, following Ferrarin *et al*., 2021.The procedure is straightforward for seiches and tides, which can be isolated by applying band-pass filters around their known frequencies. The criteria are more complicated when considering the response of sea level to the atmospheric forcing because it is characterized by a continuous spectrum. A general distinction by Holton (2004), based on the different space-time scales of the atmospheric phenomena, considers planetary-scale (order of $10^7$ m), synoptic-scale (order of $10^5$-$10^6$ m) and mesoscale motions (order of $10^4$-$10^5$ m). At the planetary scale Rossby waves move westwards against the eastward zonal flow and are therefore characterized by relatively small speeds (1–10 m/s) and long time scales (from 10 days to 100 days). Synoptic-scale systems (mostly driven by baroclinic instability) tend to

---

[2] Some significant surges may have been missed before 1933 due to lack of information, while all the high RSL events are available since 1872.

move eastwards with the mean flow and are marked by relatively large speeds (typically 10 m/s) and time scales of about
a few days. Mesoscale systems (which are topographically forced or are driven by instabilities operating at that scale)
have also relatively large speeds, of the order of 10 m/s (Markowsky and Richardson, 2010) and characteristic time scales
in the range from 10 minutes to few hours. A 10-day period for the separation between planetary and synoptic scales is
supported by the cross-spectral analysis of the 500 hPa geopotential height and sea level for the Adriatic (Orlić, 1983),
which show indeed high (low) coherence above (below) this threshold. A 10-hour cut-off period allows to distinguish
between synoptic-scale and MAV setup (including Meteotsunamis), as the latter has time scales in the range from the 10-
min period of a pure buoyancy oscillation to the 17-hour period of mid-latitude inertial oscillations (Markowsky and
Richardson, 2010). On a practical basis, Ferrarin *et al*. (2021) have used the 10-hour threshold for separating responses
to a cyclone moving in an eastward direction above the Mediterranean from a low-pressure meso-scale system travelling
in a northwestward direction along the west Adriatic coast in their analysis of the 12 November 2019 even. The separation
between PAW surges and IDAS variability was achieved by applying a low pass filter with the cut-off period placed at
120 days. The values of the contributions to water heights above 140 cm are shown in Table 1. Overall, the list agrees
with the ones compiled in other studies since the beginning of instrumental observations (Dorigo, 1961b; Livio, 1968;
Canestrelli *et al.*, 2001). The frequency of water heights follows a strong seasonal cycle (Lionello et al., 2012b). The most
intense events (with maxima above the 99th percentile) occur in November and December, with November concentrating
the largest number of intense events. However, severe events (maxima above the 80th percentile) can occur from late
September to early May and, very rarely also in summer.
The event of 4 November 1966 corresponds to both the highest storm surge and the largest water height ever recorded in
Venice. Other outstanding events are those observed on 22 December 1979 and 12 November 2019. The event of 29
October 2018 consists of two peaks separated by 6 hours, with similar water-height values (148 and 156 cm), but quite
different phases of the astronomical tide, so that the higher water level corresponds to the lower storm surge. This is the
only example in 147 years of two such high water level peaks in such a short time interval. November 2019 is also peculiar
because four water-height peaks with at least 140 cm height occurred on 12, 13, 15 and 17. The event of 12 November
2019 was particularly severe, reaching 189 cm. This was the second highest ever-recorded water height. In this case the
storm surge was relatively modest, and the exceptional water level was caused by the superposition of PAW surges,
positive astronomical tide and an unprecedented contribution caused by a meteotsunami. After the exceptionally high
water on 12 November, three successive events with water height above 140 cm occurred in just five days. As reported
in Ferrarin *et al*. (2021), these events were driven by three separate Sirocco wind episodes in the Adriatic Sea, which did
not trigger any significant seiche. These flood events were determined by the overlapping of the maximum meteorological
contribution, the tide peak and a persistent high monthly mean sea level in the northern Adriatic. Four of the eight largest
water heights since 1872 were observed during the autumn seasons of 2018 and 2019.
The astronomical tide is an important contribution to the actual water-height extremes and the time lag between the surge
peak and the nearest astronomical tide maximum may make a substantial difference. Considering the events described in
Appendix A, if surge and tide had peaked together, the observed water height, based on the linear superposition of the
different factors (a reasonable first-order approximation for the Adriatic Sea) would have approximately been 220 cm,
both on 4 November 1966 and 29 October 2018 (second peak), and  215 cm on 12 November 1951. Particularly, for the
second peak of 29 October 2018 the large negative contribution of the astronomical tide played an essential role limiting
the severity of the event. On the contrary, the coincidence of a moderate storm surge with a preexisting seiche and a high
astronomical tide level produced the sixth highest water height in Table 1. In conclusion, storm surge represents often the

largest contribution, but, in several cases, also other factors play a fundamental role. Particularly, in the case of 12 November 2019 (the second highest ever-recorded water height) several other factors exhibited contributions comparable to the storm surge, whose value was rather moderate.

The meteorological and marine conditions that led to major storm surge events have been assessed with reanalyses and dedicated model simulations, including the catastrophic storm surges of 4 November 1966 (De Zolt *et al.*, 2006; Roland *et al.*, 2009; Cavaleri *et al.*, 2010), 22 December 1979 (Cavaleri *et al.*, 2010), 1 December 2008 (Međugorac et al., 2016), 1 November 2012 (Međugorac et al., 2016), 12 November 2019 (Ferrarin et al., 2021). Bertotti *et al.* (2011) modelled five important events that occurred between 1966 and 2008. Appendix A describes the largest water-height events and related meteorological situations.

### 2.3. The propagation of the sea-level signal in the interior of the lagoon

North Adriatic water-height anomalies first propagate into the lagoon through the three inlets, and then follow the tidal channels (Fig. 1, right panel). The major channels inside the lagoon are up to 10 m deep, and this results in a propagation speed of about 10 m/s (Umgiesser *et al.*, 2004). The water then expands laterally into the shallow flats, where propagation of the wave is much slower. Astronomical tides in the southern and central basins of the lagoon are slightly amplified with respect to the inlets, because of resonance effects between the tide (both diurnal and semidiurnal) and the size of the basin. In the northern part of the lagoon, characterized by mud flats, islands and salt marshes, dissipative processes dominate over the resonance condition, so that the tidal wave shows an attenuation of about 50 % of the incoming tide (Ferrarin *et al.*, 2015). As a consequence of natural and anthropogenic morphological changes that occurred in the lagoon in the last century, the amplitude of major diurnal and semi-diurnal tidal constituents grew significantly, with a consequent increase in extreme high sea levels in Venice (Ferrarin *et al.*, 2015).

The surge signal, once it has entered the lagoon, propagates nearly without damping to the city centre, where water levels are comparable to the ones close to the inlets with a typical 1 hour delay (Umgiesser *et al.*, 2004). Other more remote areas of the lagoon show a higher phase shift with respect to the inlets of up to 3 hours. With strong NE (Bora) or SE (Sirocco) winds, the difference between water levels in the south and the north side of the lagoon may exceed 50 cm (Mel et al., 2019). The Venice city centre is relatively little affected by these differences, since it is close to the node of the oscillations. However, the strong setup at the southern part of the lagoon can lead to flooding in the city of Chioggia.

Figure 3 shows the amplification factor (percentage, values higher/lower than 100 correspond to amplification/attenuation) of sea level oscillations in the Venice city centre with respect to their amplitude at the lagoon inlets as a function of their period. This computation is based on the model of Umgiesser *et al.* (2004). In the present situation long period oscillations (≥24 hours) at the inlets propagate undisturbed into the lagoon, short ones (≤3hours) are very effectively damped and at intermediate periods they reach an amplification maximum of about 120% at 9 hours. Numerical experiments with the same model and no frictions suggest that this effect is caused by the combination of internal resonances occurring in the range from 10 to 5 hours with the strong friction inside the shallow lagoon. In the hypothetical case with very shallow inlets (maximum depth equal to 6 m) all periods below 12 hours are heavily damped. This shows that lowering the depth of the inlets would lower the water-height maxima inside the lagoon, though with problematic consequences in terms of reduced shipping, water exchange and strong erosion inside the inlets. A 1 m RSL rise (without any change in the morphology of the lagoon) would amplify the lagoon response, showing the possibility of higher extremes in the future.

### 3. Atmospheric patterns associated with extreme storm surges

#### 3.1. Characteristics of cyclones producing storm surges and floods of Venice

The Mediterranean region is characterized by a high frequency of cyclone due to a wide range of factors and mechanisms that favour cyclogenesis (Trigo et al., 1999; Lionello *et al.*, 2006a; Ulbrich et al., 2009; Lionello *et al.*, 2012a; Ulbrich *et al.*, 2012; Lionello *et al.*, 2016). These systems are often associated with extreme weather events (Jansa *et al.*, 2001; Lionello *et al.*, 2006a; Toreti *et al.*, 2010; Ulbrich *et al.*, 2012; Reale and Lionello, 2013), storm surges along the Mediterranean coastline and floods of Venice (Canestrelli *et al.*, 2001; Trigo and Davies, 2002; De Zolt *et al.*, 2006; Lionello *et al.*, 2012b; Lionello et al., 2019). Cyclones produce storm surges by two mechanisms: the inverse barometric effects caused by the decrease of atmospheric pressure during their transit over the area, and the wind set-up caused by the intense surface wind that piles up water masses against the coast of the Northern Adriatic (Lionello et al., 2019).

Figure 4 shows the temporal evolution of mean sea-level pressure (MSLP) and 10 meter wind fields during intense storm surge events. It is a composite based on the floods with a storm surge contribution higher than 50 cm in the period 1979-2019 (Table 1) using ERA5 reanalysis (Hersbach *et al.*, 2020). The time lags chosen for the composites are 36, 24, 12 hours before and 12, 24 hours after the peak of the events reported in Table 1. Figure 5 shows the same information, though it is based on the remaining events in Table 1 (with a storm surge contribution lower than 50cm). In both figures the pressure minimum is located in the Gulf of Genoa at the peak of the event, but in Figure 4 the cyclone is deeper and the MSLP gradient along the Adriatic Sea is larger. These differences have strong impacts on the intensity of the wind fields, their spatial structure and direction in the Adriatic Sea (small panels in Figs. 4 and 5), modulating the part of the Adriatic coastline that is most affected by the storm surge (Međugorac *et al.*, 2018). Indeed, the first predictions of floods in Venice were based on an autoregressive model considering as inputs the MSLP cross-basin differences (Tomasin and Frassetto, 1979). Further, the evolution of the cyclone before and after the water peak of the storm surge is different in Figs. 4 and 5. In Figure 4 cyclogenesis occurs close to the Iberian coast in the western Mediterranean Sea (as noted in Lionello *et al.*, 2012b), with a MSLP minimum well separated from the background field. In Fig. 5 cyclogenesis occurs in the north-western Mediterranean Sea within the flow produced by a pre-existing cyclone, whose centre is located north of the Alps. In both composites the lee cyclogenesis processes and the generation of a secondary minimum is evident (Trigo and Davies, 2002; Lionello, 2005; Lionello *et al.*, 2012b; Lionello et al., 2019) and the pressure gradient along the Adriatic Sea intensifies and becomes almost parallel to the basin coastlines. This synoptic configuration produces a decrease of the atmospheric pressure above northern Italy and an increase of intensity of the atmospheric flow in the Adriatic Sea directed towards its northern coast, which results in the increase of sea level in Venice.

Figure 6 shows the density (contours) of tracks of cyclones (measured in percentage relative to the total frequency of cyclones in each cell of 1.5°) producing a water height higher than 110 cm (https://www.comune.venezia.it/it/content/grafici-e-statistiche) in the period 1979-2019. Figure 6 also reports the tracks of the cyclones associated to all events that are listed in Table 1 (cyan colour), with the events of 4 November 1966, 29 October 2018 (Vaia storm) and 12 November 2019 in blue, red and green lines, respectively. Cyclone tracks shown in Fig. 6 have been identified with an automatic detection and tracking scheme (Lionello et al., 2002) applied to the ERA5 MSLP fields at a spatial resolution of 0.25° and a temporal resolution of 6 hours. The tracking scheme partitions the MSLP fields in depressions, which can be considered candidates for independent cyclones, by merging all steepest descent paths leading to the same pressure minimum. Shallow secondary minima with a small area are absorbed in the large nearest system, whose trajectory is computed by associating the location of the low-pressure centres in successive maps within a minimum distance criterion until the system disappears (cyclolysis). In that way, the method detects the

formation of cyclones inside the Mediterranean and, at the same time, avoids the inflation in the number of cyclones that would result from considering small, short-lived features as independent systems. This method has been extensively described in previous works (Lionello, Dalan et al. 2002, Reale and Lionello 2013, Lionello et al. 2016) and already used in numerous studies assessing the climatology of Mediterranean cyclones, such as Lionello et al., (2016) and Flaounas et al., (2018), in the IMILAST tracking scheme intercomparison analysis (Neu et al., 2013) and in a dedicated study on the synoptic patterns leading to high water levels along the coast of the Mediterranean Sea (Lionello et al., 2019). Readers are addressed to those studies for details.The density of tracks shown in Fig. 6 is characterized by a north-west/south-east direction in the Atlantic sector, which is different from the usual south-west/north-east pattern of the regional storm track (Neu *et al.*, 2013; Ulbrich *et al.*, 2013; Reale *et al.*, 2019). Moreover, it has a maximum in the western Mediterranean. As also shown in Lionello *et al.*, 2012 the tracks of cyclones producing the largest floods (Table 1 and Fig. 3) have distinctive characteristics with respect to the majority of cyclones crossing the Mediterranean Sea. Many of these systems enter the region from the west/southwest and follow a north-eastward direction. Differently, the majority of Mediterranean cyclones originate in the gulf of Genoa and follow a south-east direction (Trigo et al., 2002; Trigo et al., 1999; Lionello *et al.*, 2006b; Ulbrich *et al.*, 2012; Lionello *et al.*, 2016). In fact, the position of the pressure minimum, the spatial structure of cyclone-induced wind fields over the Adriatic Sea and the MSLP cross-basin differences largely affect the characteristics of the storm surge. More recent studies confirm that the position of the cyclone with respect to the basin is critical for storm surges in the north Adriatic, and its variation induces a veering of the onshore wind and even negative responses in sea level (Lionello et al., 2019).

The peculiarity of cyclones triggering storm surges is also evidenced from a cluster analysis of the daily atmospheric fields associated to the peaks above the 99.5th percentile of the daily mean detrended water height obtained with a 6-month high pass filter (Fig. 7). Only peaks that are separated by at least 3 days are considered to ensure the selection of independent extreme events. To ensure a large sampling size, the analysis uses the NCEP/NCAR reanalysis data for the 1948-2018 period (Kalnay et al., 1996). A k-means clustering (e.g. Wilks 2006) of the standardized anomalies of MSLP over the Euro-Atlantic sector and 10-m wind vectors over the Mediterranean Sea has been applied to group events with similar spatial patterns. Clusters are constructed so that differences between the daily patterns are minimized within the same cluster and maximized between the clusters, using the sum of squared distances as metric. Each cluster is characterized by its centroid (the composited spatial pattern of MSLP and 10-m wind standardized anomalies for all days in the cluster). The root mean squared difference (RMSD) between the daily standardized fields of MSLP and 10-m wind vector of all considered events and their corresponding centroid measures the total spread of the partition. When all extremes are considered (Figure 6a), the resulting centroid pattern resembles that of Figs. 4 and 5 at the peak of the event. However, the composite has a considerable spread (large RMSD), which can be reduced by progressively discriminating types of events (i.e. increasing the number of clusters, Fig. 6b). Two clusters bring the steepest decrease in the RMSD distribution and capture the distinction between cyclones to the north and south of the Alps (Figs. 7c,d) already reported by Lionello (2005).

### 3.2.   **Links to large scale patterns**
Several studies have investigated links between the main modes of atmospheric circulation variability and floods in Venice (Fagherazzi *et al.*, 2005; Lionello, 2005; Zanchettin *et al.*, 2009; Barriopedro *et al.*, 2010; Martínez-Asensio et al., 2016). The negative phase of the North Atlantic Oscillation (NAO) has been associated with both high mean sea level and floods in Venice (Zanchettin et al., 2009), although this signal is absent in autumn (when storm surges are larger).

Indeed, the large-scale circulation pattern associated with Venice floods (Lionello, 2005) is different to the NAO, being the East Atlantic (EA; Martínez-Asensio *et al.*, 2014; Martínez-Asensio et al., 2016) or the East AtlanticWestern Russia (EAWR; Fagherazzi *et al.*, 2005) the teleconnection patterns that exert the largest influence on their seasonal characteristics. Differences in the large-scale seasonal mean atmospheric circulation between active years (autumns with at least one large meteorological surge[3]) and quiet years (autumns with no large meteorological surge) have also been reported (Barriopedro *et al.*, 2010). The favourable seasonal pattern for the occurrence of large meteorological surges in autumn displays little resemblance to the NAO, but a negative pressure centre in central Europe, similar to that found in the daily-based composite of Fig. 7a.

The aforementioned relationships are often weak, though, and hence potentially sensitive to metrics, thresholds and analysed periods. This blurred influence of teleconnection patterns is not surprising, taking into account that seasonally averaged indices do not necessarily capture short-term fluctuations, and that favourable synoptic conditions (see Fig. 7) might occur under different large-scale configurations. To avoid this, a Weather Regime (WR) approach is adopted herein, which predefines a number of recurrent large-scale atmospheric circulation patterns and assigns each day to one of them. Following (Garrido-Perez *et al.*, 2020), we considered eight WRs, which yield a fair representation of the variability all-year round. Almost half of the extreme events[4] in Venice are associated with the Atlantic Low (AL) WR (Fig. 8a), whose canonical pattern (Fig. 8b) strongly resembles that of Fig. 7d and of Fig. 8.6 of (Lionello 2005). The remaining cases (arguably many of the Mediterranean cyclones included in Fig. 7c) occur under different WRs without a clear preference, although some anticyclonic WRs (e.g. the Atlantic High) are unfavourable for extreme meteorological surges. Despite the strong association with AL on daily scales, the Spearman's rank correlation r between the seasonal frequency series of AL days and extreme events is low (r=0.26 for 1948-2018, $p<0.05$,where p is the significance level) and similar to that obtained from other less influential WRs (e.g. Zonal Regime, r=0.27; $p<0.05$). This illustrates that the interannual variability of detrended water-height extremes in autumn cannot be well described by fluctuations in the dominant patterns of atmospheric circulation variability over the Euro-Atlantic sector.

### 3.3. The role of solar cycles on extreme floods

Some studies have reported decadal fluctuations in the frequency of floods in phase with the 11-yr solar cycle during the second half of the 20th century, such that periods of high solar activity have coincided with more frequent and persistent floods in Venice (Tomasin, 2002; Lionello, 2005; Barriopedro *et al.*, 2010) and other Mediterranean coastal stations (Martínez-Asensio et al., 2016). This signal results from the atmospheric forcing on sea level, as revealed by hindcasts of a barotropic ocean model forced with observed atmospheric pressure and winds (Martínez-Asensio et al., 2016).

An unanswered question is how such a small solar forcing could modulate the tropospheric circulation over the Euro-Atlantic sector. Several hypotheses have been proposed, including decadal variations of the regional atmospheric circulation that promote the constructive interference with the favourable pattern for the occurrence of extreme floods during periods of high solar activity (Barriopedro *et al.*, 2010). Other studies claim for a solar modulation of the stratospheric polar vortex and a lagged response of the NAO, e.g. (Thiéblemont *et al.*, 2015) and reference therein). However, this mechanism would mainly affect the winter NAO, rather than the decadal variability of autumn floods in Venice. In addition, modeling studies reveal negligible impacts of the 11-yr solar cycle on the NAO and demonstrate that

---

[3] Events in which the meteorological surge exceeds the 95th percentile of the total distribution

[4] Events with daily mean detrended water height above the 99.5th percentile of the 1948-2018 distribution

decadal variations of the NAO can eventually vary in phase with the 11-yr solar cycle by random chance (Chiodo *et al.*, 2019). Given the lack of mechanistic understanding, the null hypothesis of internal variability cannot be rejected.

Indeed, an updated analysis of autumn extreme events (99.5th percentile) from the longest series of daily mean detrended water height in Venice based on the data of Raicich (2015), covering the period 1872-2018, shows that the 11-yr solar signal is no longer evident since the ~2000s, nor it was present before the ~1950s (Fig. 9 top panel). Significant correlations are limited to the period from 1970 to 2000 (Fig. 9, bottom panel) and give rise to strong co-variability during the second half of the 20th century, coinciding with the Grand Solar Maxima covered by most studies. Further, there is no indication of the presence of an 11-year periodicity in the series of autumn mean water height (Figs. C1 in Appendix C ) and when extreme events are defined using different thresholds (Fig. C2 in Appendix C). These results suggest that if there is a solar signal it would likely be non-stationary (arguably masked by other sources variability) and/or non-linear (e.g. confined to Grand Maxima of solar activity). The alternative hypothesis is that the decadal variability of extreme surges is due to other causes, including internal variability. It is plausible that, superimposed on the uncontroversial increasing frequency of Venice flooding due to the RSL rise, the frequency of extreme water heights will experience large interannual-to-decadal variations in the future, as it has been observed in the recent period. However, the causes of this variability are still uncertain.

## 4. Past and future evolution
### 4.1. Past evolution and recent trends of floods and extreme sea levels

Enzi and Camuffo (1995) presented the most complete compilation of pre-instrumental extreme water heights observed in Venice by reviewing hundreds of historical documents, thus obtaining a sequence of over 100 events in the 787-1867 period. The long-term evolution has been studied by Camuffo and Sturaro (2004) combining information from documentary sources and instrumental observations. From 1200 to 1740 the flood frequency was <0.1 $yr^{-1}$, except for the Spörer period (1500-1540), when it was 0.63 $yr^{-1}$. Subsequently, the frequency increased from 0.19 $yr^{-1}$ in 1830-1930 to 1.97 $yr^{-1}$ in 1965-2000. Considering specific thresholds, the number of floods above the 120 cm threshold has increased from 1.6 $decade^{-1}$ (average frequency during the first half of the 20th century) to 40 $decade^{-1}$ in the last decade (2010-2019). Considering a lower (110 cm) threshold the number of events has increased from 4.2 $decade^{-1}$ to 95 $decade^{-1}$.

Former studies of recent trends (Trigo and Davies, 2002) found that in the second half of the 20th century the local RSL rise compensated for the decreasing frequency of storms, leading to no change in the frequency of floods. Other studies, found a significant positive trend of moderate floods in Venice and Trieste during the second half of the 20th century (1951-1996), that was attributed to increases in the frequency of Sirocco wind conditions over the central and southern Adriatic (Pirazzoli and Tomasin, 2002). A more recent study considered data in the period 1940-2007, reporting a 4% reduction of all water-height events, but no significant increase in the frequency or intensity of the most extreme events if the effect of RSL rise is subtracted from the data (Lionello *et al.*, 2012b). According to Ferrarin *et al.* (2015), the detected increase in amplitude of the tidal waves enhanced the occurrence of severe water heights in Venice in the period 1940-2014, while changes in storminess had no significant long-term impact.

Observations made in Venice and Chioggia allowed to extend the series of water-height data back to the second half of the 18th century (Raicich, 2015). For this longer period, the time series of meteorological surge frequency does not exhibit a significant long-term trend, but strong inter-annual and inter-decadal variability. In summary, the amount of current evidence shows that while the frequency of floods has clearly progressively increased in time after the mid-twenty century, there is no clear indication of a sustained trend at multi-decadal time scales in either the frequency or the severity of extreme meteorological surges. The presence of a substantial interannual and interdecadal variability explains

differences among studies, which have considered different periods and different thresholds. The long term increase of flood frequency is largely caused by RSL rise connected to both climatic change and land subsidence (see Zanchettin *et al* 2021 in this special issue).

### 4.2. **Future evolution of extreme water heights**

Several past studies considered the future evolution of meteorological surges and water heights in the Adriatic Sea. A first analysis was based on a doubled-$CO_2$ scenario and a single climate simulation (Lionello, Nizzero and Elvini, 2003). Successive studies adopted the SRES scenarios and multiple simulations (Marcos *et al.*, 2011; Lionello, Galati and Elvini, 2012c; Troccoli *et al.*, 2012; Mel, Sterl and Lionello, 2013). The most recent studies have considered the whole Mediterranean Sea or large parts of it and an ensemble of simulations for high (RCP8.5) and moderate (RCP4.5) emission scenarios (Conte and Lionello, 2013; Androulidakis *et al.*, 2015; Vousdoukas *et al.*, 2016; Lionello *et al.*, 2017; Mentaschi *et al.*, 2017; Vousdoukas *et al.*, 2017; Vousdoukas *et al.*, 2018). These studies are not fully comparable in that some of them (e.g Lionello *et al.*, 2017) considered separated contributions from RSL rise and changes of meteorological surges, whereas others (e.g. Vousdoukas *et al.*, 2017 and 2018) addressed the overall change of sea level extremes. Further, Vousdoukas *et al.* considered the 100-year return values, while Lionello *et al.* (2017) considered annual maxima and 5 and 50-year return values. Studies assessing only meteorological surges suggest non-significant changes or a significant reduction of their intensity, which might reach about 5% for high emissions at the end of the 21st century (with consistent attenuation also of the wind wave height). This weak climate change signal is consistent with the future prevalent decrease of cyclone intensity and related wind speeds in the Mediterranean region that is suggested by most studies in spite of model-related uncertainty and sub-regional differences (see Reale et al, 2021 for a recent comprehensive update, and Lionello et al., 2008, Zappa et al., 2013, Nissen et al., 2014, Zappa et al. 2015). However, there is substantial agreement that the future RSL rise will be the dominant factor that will increase frequency and height of floods (Lionello *et al.*, 2017; Jackson and Jevrejeva, 2016; Jevrejeva *et al.*, 2016; Vousdoukas *et al.*, 2017; Vousdoukas *et al.*, 2018). Only a very low rate of future RSL rise, such as that hypothesized in Troccoli *et al.* (2012), might prevent future increase of floods. However, such a low future RSL rise is very unlikely (Jordà et al., 2012), because it is lower than the global sea-level rise under the RCP2.6 scenario in the IPCC SROCC (Oppenheimer *et al.*, 2019) and it would require the RSL rise in Venice during the 21$^{st}$ century to be lower than observed during the 20$^{th}$ century (see also Zanchettin *et al*. 2021, in this issue).

The future variation of amplitude of tides and surges in response to sea-level rise will depend on the adaptation strategy of coastal defences (Bamber and Aspinall, 2013) – protection versus retreat. (Lionello et al., 2005) showed that a full compensation strategy (protection), preserving the present coastline by dams, would reduce the amplitude of tides and storm surges, while a no compensation strategy, allowing permanent flooding of the low coastal areas (retreat), would increase the amplitude of the diurnal components and the amplitude of storm surges at the North Adriatic coast. These effects are small, but not negligible, being about 10% for the diurnal component in case of 1-m sea-level rise.

Projections of extreme sea levels (ESLs) were produced combining dynamic simulations of all relevant components during the present century, and under moderate and high emission scenarios. They include: MSL rise and variations of future tides, meteorological surges and wind wave set-up (Vousdoukas *et al.*, 2017; Mentaschi *et al.*, 2017; Vousdoukas *et al.*, 2018), but do not include the effect of local subsidence. ESLs were produced through a probabilistic process-based

framework (Jackson and Jevrejeva, 2016; Jevrejeva *et al.*, 2016), incorporating the large uncertainties originating from the Greenland and Antarctic ice sheets under high emission scenario (Bamber and Aspinall, 2013). Values for different return periods were estimated using non-stationary extreme value statistical analysis (Mentaschi *et al.*, 2016) and variations with respect to the 2001-2020 baseline. Here, the spatially averaged values along the north-west Adriatic coast[5] are considered.

The 100-year extreme sea level (100y-ESL) (Fig. 10) in the North-West Adriatic Sea[6] by 2050 is very likely (5-95th percentile) to rise by 12 to 17 cm under the RCP4.5 moderate-emission-mitigation-policy scenario and by 26 to 35 cm under the RCP8.5 high emissions scenario (Vousdoukas *et al.*, 2018). Similarly, rise to 24-56 cm and 53-171 cm, respectively, by the end of the century. By the year 2050, the frequency of present-day 100-year events is projected to increase by 2 or 10 times (i.e. one event per 50 or 10 years) depending on the emissions scenario. By the end of this century, events with the severity of current 1-in-100-year extremes would occur at least every 5 and 1 year, under moderate and high emissions, respectively.

Breaking down the contributing factors to the increase in 100y-ESLs in the North-West Adriatic Sea (Fig. 11), thermal expansion accounts for 58% and 32% (median values) of the projected increase towards the end of the century, under moderate and high emissions, respectively while the Antarctica and Greenland ice sheet melting contribution vary from 14% to 20% (median values). However, the combined contributions from ice mass-loss from glaciers, and ice-sheets in Greenland and Antarctica together, are the dominant factor by 2100, contributing to 61% and 51% (median values) of the 100y-ESL increase under moderate-and high emissions, respectively.

While the above paragraphs discuss changes due to climatic and meteorological factors, the future dynamics of tides and surges in response to sea-level rise will also depend on the evolution of the shoreline in the area. As sea levels rise, societies will have to decide whether to protect the coast and maintain the current shoreline (e.g. with coastal dams), or allow shoreline retreat. Previous studies (Lionello et al., 2005) have shown that a protection strategy would reduce the amplitude of tides and storm surges and increase that of Adriatic Sea seiches, while allowing for permanent flooding of the low coastal areas and retreat, would increase the amplitude of the diurnal tide components and storm surges. These effects are small, but not negligible, being about 10% for the diurnal component in the case of 1 m RSL rise.

## 5.  Conclusions and outlook

There is a widespread view that in the floods of the Venice city centre are mostly caused by storm surges and that the actual maximum water height depends substantially on the timing of the storm surge peak with respect to the phase of the astronomical tide. Consequently, efforts have traditionally focused on the correct simulation of the intensity, timing and spatial variability of the wind (mainly the Sirocco) for the accurate reproduction of water-height extremes. This review confirms the paramount importance of storm surge, which produced the highest recorded flood (4 November 1966), but also identifies other phenomena that, though they individually produce lower water-height anomalies than storm surge, can act constructively and yield extreme events. The event of 12 November 2019 (the second highest ever recorded flood) provides a good example. Therefore, research is required on PAW surges and meteotsunamis, the other contributions to meteorological surges, including their joint distributions, in order to better understand the likelihood of compound events

---

[5] The area in the red box in Figure 1 is from lon 12.1 W to  12.9°W; and from lat 43.8 N and 45.8°N
[6] The area in the box from lon 12.1 W to  12.9°W; and from lat 43.8 N and 45.8°N is considered

as that of November 2019. Furthermore, a multivariate statistical model that describes extreme water heights as a function
of the various contributions would provide a more complete characterization of extreme events.
The actual effect of wave-set up on the water height inside the Venice lagoon remain uncertain. Some studies have
computed it during individual storms affecting Venice (Bertotti and Cavaleri, 1985; Lionello, 1995; De Zolt *et al.*, 2006)
and for 100y-ESL projections (Vousdoukas *et al.*, 2016; Vousdoukas *et al.*, 2017) and have shown that the wave set-up
contribution at the Adriatic shoreline can exceed 10 cm, but its relevance for the flooding of Venice city centre would
require that it initiates sufficiently offshore to affect the sea level at the lagoon inlets.This remains to be investigated.
The occurrence of floods, beside from long-term RSL rise, is modulated by IDAS, sea-level variability at shorter time
scales (from seasonal to decadal). Similarly, also the occurrence of meteorological surges displays strong interannual to
decadal variability. Evidences linking this variability with astronomical (e.g. the solar cycle) and climate patterns (e.g.
North Hemisphere teleconnections) remain elusive, from both statistical and theoretical approaches. These issues are
important for the development of seasonal predictions of sea-level extremes, understanding of observed trends and their
attribution to long term anthropogenic climate change (and local subsidence). Longer records and better understanding
of the sea-level responses to atmospheric forcing and remote influences would contribute to fill these knowledge gaps.
The synoptic conditions leading to extreme storm surges at Venice are clearly documented, as they are produced by
cyclogenesis occurring in the western Mediterranean Sea. There is consensus on the secondary role that the
meteorological forcing plays in the long-term changes of major floods. Its contribution may decrease further in the future
because of their projected attenuation. However, the confidence on future weakening depends on the capability of climate
models to correctly reproduce the full set of meteorological contributions under climate change, including  storm surges,
Meteotsunamis and PaW surges. Literature on projections of PAW surges and meteotsunamis is presently unavailable
and progress on these factors is urgently required as their changes may have different signs and magnitude from those of
storm surges. Therefore, while presently available studies agree on the future attenuation of meteorological surges,
analyses understanding the role of the different meteorological forcings on extreme sea-level events are missing and
deserve investigation.
This review confirms the consensus concerning the key control of the frequency and severity of floods in Venice exerted
by historic and future RSL rise. Hence, understanding and predicting the future evolution of extreme water heights in
Venice depends critically on the availability of RSL rise projections with lower uncertainty than at present. A large
fraction of such uncertainty is related to the future emission scenario. Adopting a moderate-emission-mitigation-policy
scenario (RCP2.6), or a high emissions scenario (RCP8.5) would imply a 30% difference in the projected 100y-ESL at
the end of the 21st century. Another major source of uncertainty concerns the melting of ice-sheets, which accounts for
the largest increase of the 100y-ESL at the end of this century, particularly for a high emission scenario. Further, scenarios
for local anthropogenic and long term natural subsidence needs to be developed, as they can further contribute to the
future increase of extreme water heights. Other factors, such as changes in storminess or the deviation of the
Mediterranean mean sea level from that of the Subpolar North Atlantic (caused by steric effects and redistribution of mass
within the Mediterranean Sea) appear to be less important (see Zanchettin *et al.*, 2021 in this special issue).
Reducing uncertainty in the future projections of water-height extremes is only one aspect of the research needed. The
other aspect is adaptive planning of coastal defences to consider the large uncertainty on future evolution.. A moderate
scenario suggests a 10% and 30% increase of 100y-ESL in 2050 and 2100, respectively. A high emission scenario shows
a 25% increase already in 2050, reaching 65% in 2100. These ranges are further enlarged by the uncertainty in scenario
projections (leading to 100y-ESL increase up to 65% and a 160% in 2050 and 2100, respectively), which should be further
expanded to higher values including high-end scenarios (see Zanchettin *et al.*, 2021 in this issue). Further, the inclusion
of uncertainties on future subsidence is required to assess the likely range of future extreme water heights, which provide
the actual information for the hazard to be faced by coastal defences, the environment of the city and of the lagoon. In
other words, the uncertainty range on extreme water heights is larger than on ESLs and should be detailed at a finer spatial
scale. The large range of possible changes, especially after 2050 is not expected to be reduced substantially in the
upcoming years, as it largely relies on human decisions and pervasive modelling uncertainties, which limits the generation
of constrained climate information and poses major challenges for policy-making decisions on the development of
effective adaptation measurements. These results (see also Lionello *et al.* 2021 in this special issue) stress the need for
planning and implementing defence strategies of Venice that can be adapted to face the large range of plausible future
sea-level extremes.
**Acknowledgements**
M. Reale has been supported in this work by OGS and Cineca under HPC-TRES award number 2015-07 and by the
project FAIRSEA (Fisheries in the Adriatic Region - a Shared Ecosystem. Approach) funded by the 2014 - 2020 Interreg
V-A Italy - Croatia CBC Programme (Standard project ID 10046951).The work of M. Orlić has been supported by
Croatian Science Foundation under the project IP-2018-01-9849 (MAUD). Scientific activity by DZ and GU performed
in the Research Programme Venezia2021, with the contribution of the Provveditorato for the Public Works of Veneto,
Trentino Alto Adige and Friuli Venezia Giulia, provided through the concessionary of State Consorzio Venezia Nuova
and coordinated by CORILA. D. Barriopedro was supported by the Spanish government through the PALEOSTRAT
(CGL2015-69699-R) and JEDiS (RTI2018-096402-BI00) projects.
**Author contribution**
PL coordinated the paper. Specific contributions to the sections are as follows (LA = leading author, CA = contributing
author). Section 1: LA: PL; CA: RJN, DZ. Section 2: LA: MO and FR; CA: PL, GU, CF. Section 3: LA: MR and DB,
CA: FR and PL. Section 4: LA: MV, CA: FR, PL. Section 5: LA: PL, CA: RJN, DZ, DB. Figure 1, 3,4 and 5 MR, Figure
2 PL, Figure 3 GU,  Figures 6.7 and 8 DB; Figures 9 and 11 MV. Table 1:CF; table 2: FR; Appendices: AI: FR; AII :MR;
AIII: DB ; Figure III1  and III2, DB.
**Competing Interest**
The authors declare that they have no conflict of interest.
**6.    Appendix A: Selected major events**
Here we present a short description of extreme sea-level events based on original reports. Each description is based on
the cited sources, which often include synoptic weather maps and diagrams of relevant meteorological parameters (see
table A1)
**A.0. 15 January 1867**
On 15 January 1867, that is just few years before the beginning of regular sea-level records a remarkable storm surge
occurred. Although no tide gauge data are available, contemporary sources reported measurements taken at local
hydrometers.

Zantedeschi (1866-67), quoting the local Civil Engineering Office (Ufficio del Genio Civile), reported that the maximum observed water height was 1.59 m 'above the common ordinary high water marked at the royal hydrometer in the Grand Canal'. The 'common ordinary high water' is also known as the 'comune marino' (CM), that is the upper edge of the green belt formed by algae on quays and walls, often indicated by an engraved horizontal mark and/or a 'C' (Rusconi, 1983; Camuffo and Sturaro, 2004). According to Dorigo (1961a) the ZMPS is 22.46 cm below the CM of 1825, upon which the tide gauge zero at S. Stefano was based. Therefore, under the hypothesis that the same CM was adopted at the royal hydrometer and at S. Stefano, the maximum water height should have been approximately 181 cm above ZMPS.

However, later sources gave different valuess. Annali (1941) reported 132 cm above the 1825 CM, therefore the height would turn out to be 154 cm (153 is reported, maybe due to rounding). Dorigo (1961) also reported 153 cm, probably quoting Annali (1941).

If the 181-cm height was correct, the 1867 height would be the third largest water height ever measured in Venice, not too far from the 187 cm of 12 November 2019 and the 194 cm reached on 4 November 1966. Note, however, that in the 1860's the relative MSL was about 30 cm lower than at present, which makes the 1867 event very remarkable.

**A.1. 16 April 1936**

A cyclone affected the western and central Mediterranean, with a minimum MSLP around 990 hPa in the Gulf of Lions, causing strong southerly winds blew over the Adriatic. In Venice wind mostly blew from the first quadrant but it veered to SSW near the surge peak, with gust speed over 25 m s$^{-1}$; in the meantime the MSLP dropped to 990 hPa.

The water height reached 147 cm; at that time it represented the second highest value ever recorded, the first having been observed on 15 January 1867 (see above). The water-height peak occurred about 2 hr after the astronomical tide maximum. The meteorological surge contribution was about 91 cm.

**A.2. 12 November 1951**

From 10 to 12 November a deep cyclone formed in the Ligurian Sea where MSLP dropped from 1008 to 984 hPa. On the Ionian Sea and the Balkans MSLP was higher than 1012 hPa, and the strong MSLP gradient induced strong southerly winds over the Adriatic Sea, up to over 20 m s$^{-1}$ in Venice. As a result, the water height in Venice increased both because of the wind-induced surge and the local inverse barometer effect. Luckily, the surge peak occurred at the astronomical tide minimum. If it had occurred at the next high tide, 5 hr later, the observed water height would have been about 65 cm higher. The water -height peak was 151 cm and it exceeded the official danger level of 110 cm for about 9 hr. The meteorological surge peak attained 86 cm.

**A.3. 4 November 1966**

On 3 and 4 November 1966 the MSLP field over the Mediterranean was characterized by a cyclone to the west and an anticyclone to the east. The cyclone centre deepened and slowly moved from the northwest Mediterranean to northeast Italy, while the zonal MSLP gradient increased over the Adriatic. As a consequence, strong and persisting southerly wind affected the Adriatic Sea. In Venice Sirocco speed reached 20 m s$^{-1}$ with gusts up to 28 m s$^{-1}$, and the MSLP dropped to 992 hPa.

The water height of 194 cm and the meteorological surge height of 143 cm are the highest values in the whole instrumental record. The water height remained over 110 cm for 22 hr. Economic losses for the city of about 400 hundred millions euros have been estimated.

Note that two elements limited the water-height peak, namely the fact that the astronomical tide was negative, though near zero, at the time of the maximum surge, and that in those days the Moon phase was close to last quarter, making the astronomical tide amplitude relatively small, around 30 cm. Had the surge peak occurred 5 hr earlier, the water height would have attained about 220 cm.

**A.4. 22 December 1979**

This event was connected with a cyclone whose minimum was less than 990 hPa, that moved on 21 and 22 December from the Algerian coast to the Gulf of Genoa. The combination with higher MSLP over the Balkans enabled southerly wind blow over the central and southern Adriatic, with gusts up to 20 m s$^{-1}$, while in the northern Adriatic Bora prevailed with gusts over 20 m s$^{-1}$. The local MSLP was not particularly low (1001 hPa), thus the surge was almost entirely attributed to wind.

The meteorological surge peak reached 106 cm and came 3 hr before the astronomical tide maximum: nevertheless, the water height was remarkably high, namely 166 cm which represents the third highest observed value. A water height higher than 110 cm lasted for 7 hr.

**A.5. 1 February 1986**

The synoptic situation consisted of cyclone over the western Mediterranean, this time centred in the Gulf of Lions, and an anticyclone over eastern and northern Europe. A southerly wind flow affected the whole central Mediterranean, including the Adriatic Sea, but a Bora component was present over the northern Adriatic. Southerly wind was particularly strong in the southern Adriatic (almost 30 m s$^{-1}$ gusts in Bari), while in Venice Bora gusts were faster than 20 m s$^{-1}$.

This event is characterised by the fourth highest water height ever measured in Venice, that is 159 cm. The event severity was the result of a moderate meteorological surge of 70 cm, that occurred just 1 hr after a 35 cm astronomical tide maximum and close to the peak of a large seiche.

**A.6. 6 November 2000**

This event was caused by the combined effect of a large cyclone affecting the whole western Europe and an anticyclone over eastern Europe. The lowest MSLP was observed in the English Channel with values lower than 970 hPa. The eastward movement of the cyclone caused the whole Adriatic to experience a remarkable MSLP decrease in the 24 hr preceding the surge, up to a 27-hPa drop in Venice.

As on 1 February 1986, during this event the storm surge and the astronomical tide maximum almost coincided. The observed water height attained 144 cm and the surge grew up to 87 cm. The water height remained above 100 cm for over 7 hr.

**A.7. 1 December 2008**

An intense cyclone, with strong westerly flow affected the western Mediterranean Sea. The day before the flood a small-scale cyclonic circulation developed over the Gulf of Genoa and moved eastward into the River Po valley. This caused surface wind over the Tyrrhenian and Adriatic Seas to veer from W to SW, then to S, intensifying in the meantime and reaching the maximum intensity in the early hours of 1 December. In the afternoon, the cyclonic circulation began weakening and the intensity of the associated wind in the Adriatic Sea progressively decreased.

From the late afternoon of 30 November to the early morning of 1 December, MSLP in Venice dropped by about 13 hPa in 9 hr, reaching 994 hPa. The wind veered from NNE to SE around 01:30 UTC, with speed between 15 and 20 m s$^{-1}$ for the following 7-8 hr.

The water height attained 156 cm, that is the fifth highest value since 1872. The maximum meteorological surge height was 62 cm and it occurred less than 1 hr before the astronomical tide maximum.

**A.8. 29 October 2018**

The event was caused by the combined action of a cyclone, centred between the Gulf of Lions and the Gulf of Genoa, whose minimum MSLP was lower than 985 hPa, and an anticyclone over northeastern and eastern Europe. This configuration enabled strong Sirocco along the Adriatic, with speed around 15 m s$^{-1}$ and gusts up to 25 m s$^{-1}$ from the late morning to the evening in Venice, where MSLP reached a minimum of 996 hPa.

The strength and persistence of southerly winds caused the meteorological surge to remain particularly high. The highest water height was reached at 13:40 UTC with 156 cm (fifth value in the history of the observations in Venice), a couple of hours later than the astronomical tide maximum, then the water height decreased to 119 cm at 16:35 UTC at rose again up to 148 cm at 19:25 UTC. The meteorological surge level peaked at 91 cm together with the maximum water height and to 117 cm at 19:20 UTC, in coincidence with the astronomical tide minimum. The 117 meteorological surge level represents the second highest ever observed and the 119 cm value the highest minimum water height. Overall, the water height was higher than 120 cm for 14 hr, as on 4 November 1966.

**A.9 12, 15 and 17 November 2019**

On November 12$^{th}$, 2019, an exceptional flood event took place in Venice, second only to the November 4$^{th}$ event, 1966. Moreover, with 15 water heights above 110 cm and 4 above 140 cm, November 2019 was the worst month for flooding in Venice since the beginning of sea-level records.

The extreme high sea level recorded in Venice was due to the combination of the following large-scale and local dynamics:

- the in-phase timing between the peak of the storm surge and an astronomical tide maximum;
- the PAW surge produced by a standing low-pressure and wind systems over the Mediterranean Sea persisting for the whole month of November (which determined a high monthly mean sea level in the northern Adriatic Sea);
- the storm surge produced by a deep low-pressure system over the central-southern Tyrrhenian Sea that generated south-easterly winds along the main axis of the Adriatic Sea;
- the meteotsunamis produced by a fast-moving mesoscale cyclone travelling in the north-westward direction along the Italian coast of the Adriatic Sea;
- a local set-up produced by very strong south-westerly winds over the Lagoon of Venice.

The MSLP minimum of the cyclone on the Tyrrhenian Sea was about 990 hPa. A small deep secondary MSLP minimum formed in the afternoon, reaching 988 hPa at Venice around 21 UTC. Initially, moderate northeasterly wind was blowing over the north Adriatic (about 10 m s$^{-1}$ at Venice), but between 21 and 22 UTC it veered to S,E then to SW, and sustained wind reinforced up to 20 m s$^{-1}$ at Tessera airport.

The highest water height was reached at 21:50 UTC with 189 cm, that represents the second highest value in the history
of the observations in Venice, and it almost exactly coincided with the astronomical tide peak. The meteorological surge
level peaked at 100 cm, representing the fourth highest value ever observed. The peak water height was similar to the
1966 value (namely 194 cm), but, while in 1966 it was mainly the result of a huge meteorological component (143 cm,
see A.3 above), in 2019 the astronomical tide contribution also played a significant role. Moreover, in 2019 the RSL was
11 cm higher with respect to 1966.
On 15 November another storm surge developed in connection with a large cyclone over west Europe, having a 995 hPa
MSLP minimum over France, and extending into Algeria. Local pressure in Venice reached 1001 hPa and wind blew
from SE at less than 10 m s$^{-1}$. The water height peaked at 154 cm at 10:40 UTC.
**Appendix B: Simulation of the tide- meteorological surge interactions**
This short appendix is dedicated to an experiment carried out using the model framework of Cavaleri et al. (2019). Three
different simulations have been performed using only the astronomical tidal forcing (SHYFEM Tide), the meteorological
forcing (SHYFEM Surge) and the full forcing (SHYFEM Total). The difference between the "SHYFEM Total"
simulation and the sum of "SHYFEM Surge" and "SHYFEM Tide" represents the effect of the nonlinear interactions
between the astronomical tide and the meteorological surge. Figure B1 shows the results of the simulation and the small
magnitude of the nonlinear effect, which is about only 5% of the total surge in correspondence with the highest peak in
the simulated period.
**Appendix C: Wavelet of the storm surge frequency**
In order to integrate the discussion in section 5.3 on the presence of a 11-yr periodicity of extreme detrended water heights,
Figs. C1 and C2 show the amplitude of the wavelets of the time series of the autumn mean water height for 1924-2018
and of its daily meteorological surge extremes. Missing values in the time series before 1928 prevented the computation
of the wavelet transform in Fig. C1 for the whole period 1872-2018 covered with the data. Figure C2 has been limited to
the same period for coherence with Fig. C1. In both graphics, a decadal signal consistent with the 11-year solar cycle is
present only for a few decades from the 1970s to the 1990s and absent before and after this period.

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

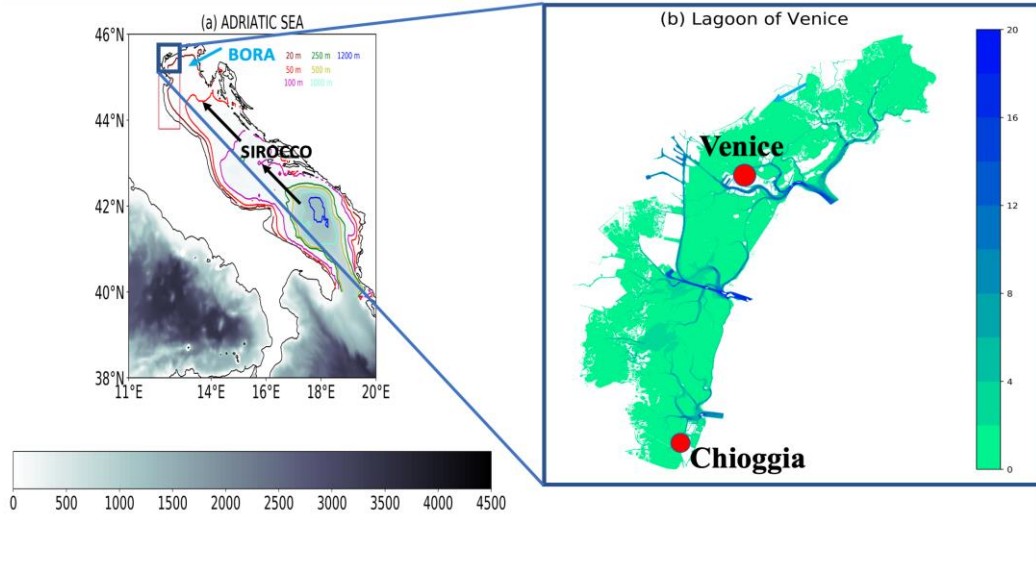

**Figure 1: left panel: bathymetry of the Adriatic Sea with the position of Venice and arrows denoting the directions of the two main wind regimes affecting the North Adriatic. The red box (whose northern part includes the whole lagoon) denotes the area represented by the data in Figs.8 and 9. Right panel: morphology of the lagoon of Venice with the three inlets connecting it to the Adriatic Sea, and the position of the city and of Chioggia (the bathymetric data are for year 2002 and are based on original data provided by Magistrato alla Acque di Venezia and elaborated by Sarretta et al. (2010),**

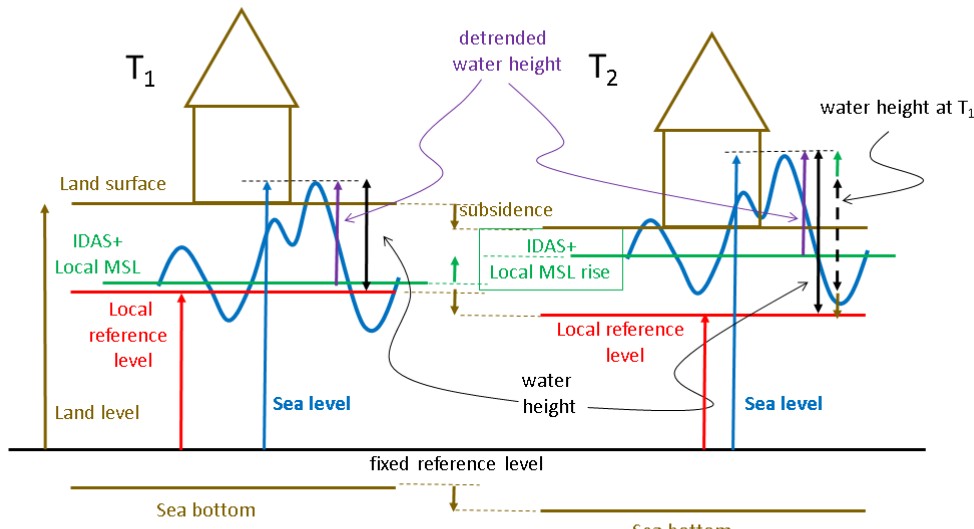

**Figure 2 schematic showing changes of water height for a hypothetical identical event occurring at time $T_1$ (mid of 20th century) and $T_2$ (first decades of the 21st century). Local subsidence has shifted to a lower level the land surface, the reference level and the sea bottom. RSL rise and IDAS have shifted to an upper level the sea surface. The water height of the same event hypothetically measured at $T_1$ and $T_2$ differ by the IDAS and RSL rise contribution. The latter is split in local subsidence and mean sea level rise. The detrended water height is the addition of the meteorological surge (Storm surge, PAW surge, meteotsunamis), astronomical tide and seiches.**

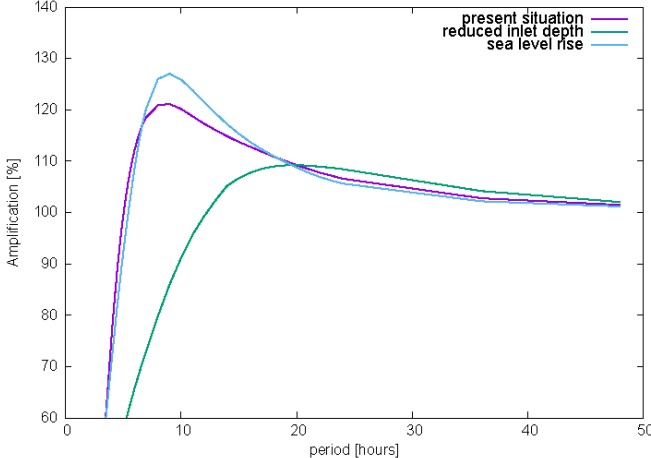

1019

Figure 3 Amplification (percentage, y-axis, values higher/lower than 100 correspond to amplification/attenuation) of sea level oscillations in the Venice city centre with respect to their amplitude at the lagoon inlets as a function of their period (hours, x-axis). The curves show the present situation (violet), a hypothetical reduction to 6 meters of the depth of the three inlets of the lagoon (green), a RSL rise of 1 meter without any change in the morphology of the lagoon.

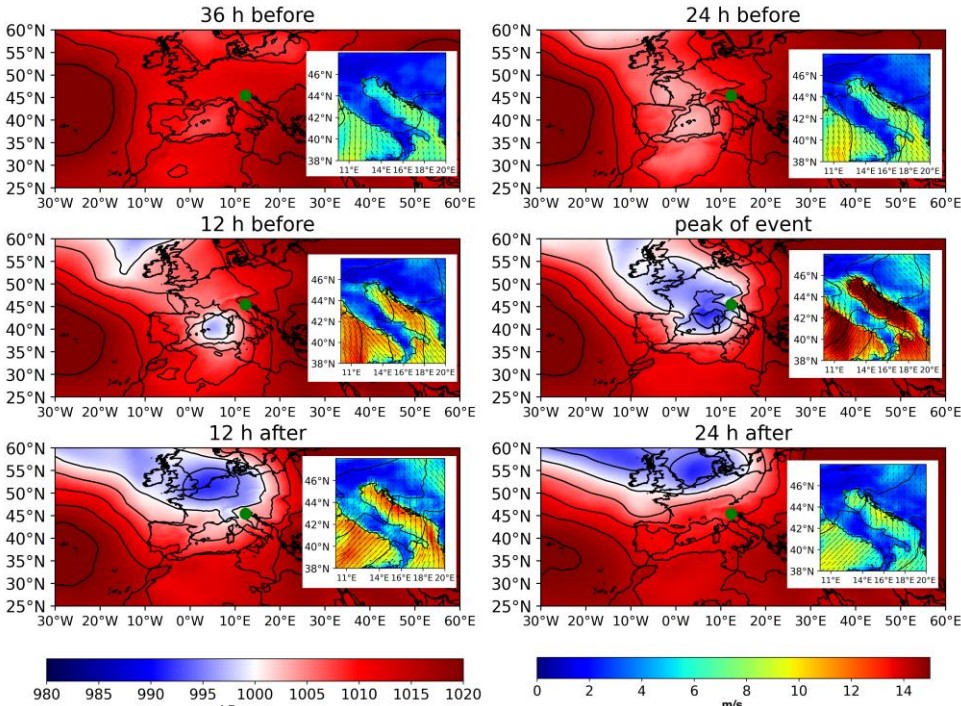

1024

Figure 4 large panels show the composite of MSLP fields based on ERA5 (in hPa, left colo  bar) datasets associated with storm surges higher than 50 cm in Venice (see table 1). Small panels show the corresponding wind fields over the Adriatic Sea (m/s, right color bar). The time lags chosen for the composites is 36, 24 , 12 hours before  and 12, 24 hours after the peak of the event. The green dot shows the location of the city of Venice

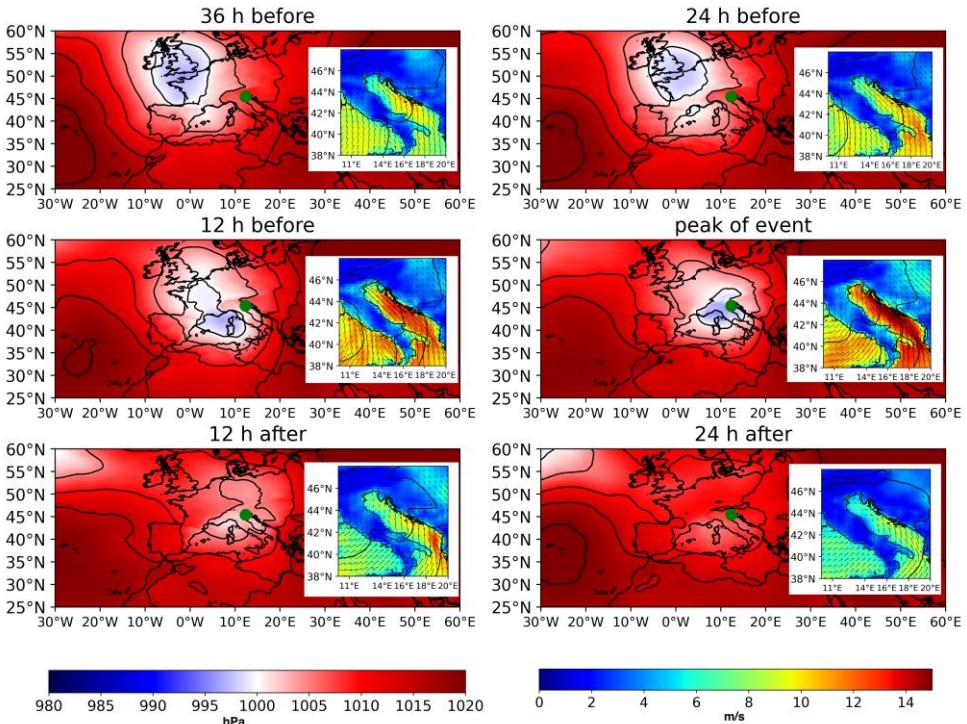


**Figure 5 Same as figure 2, except it is based on the events in table 1 with storm surge height lower than 50cm**

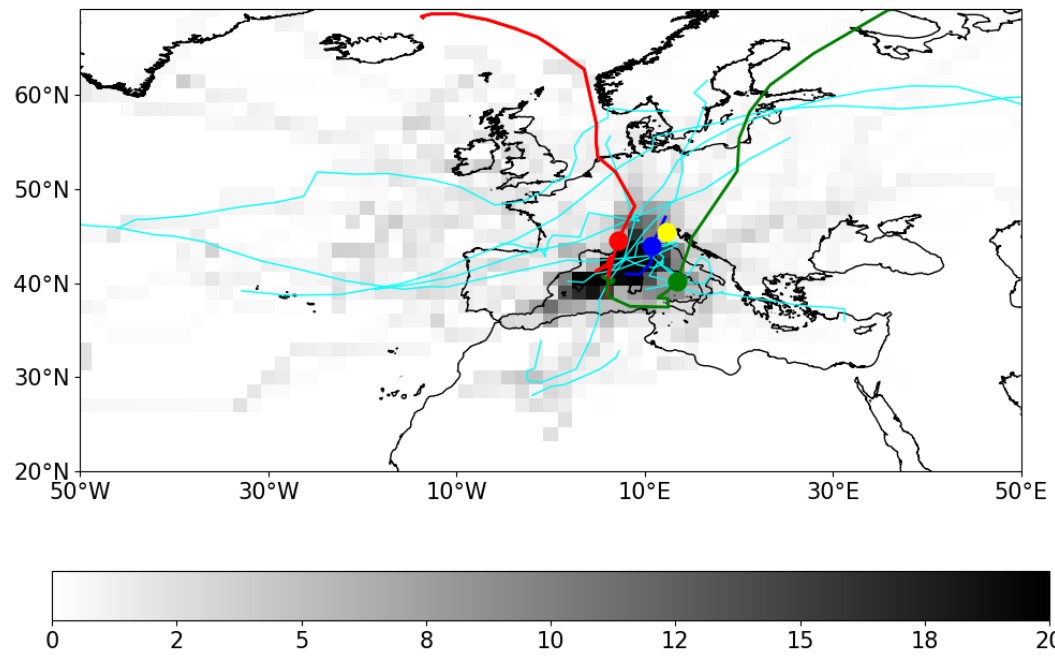

**Figure 6 Density of tracks of cyclones associated with storm surges contributing to water-height maxima above 110 cm (relative**
**frequency of cyclones for each cell of 1.5 in percentage of total, grey bar in the panel, based on ERA5). Cyan tracks represent**
**the events reported in Table 1 with water-height maxima140 cm (see Table 1), the red, green and blue tracks represent the**
**29/10/2018, 19/11/2019, and 6/11/1966 events (the blue track is based on ERA40data). Yellow dot represents the location of the**
**city of Venice**

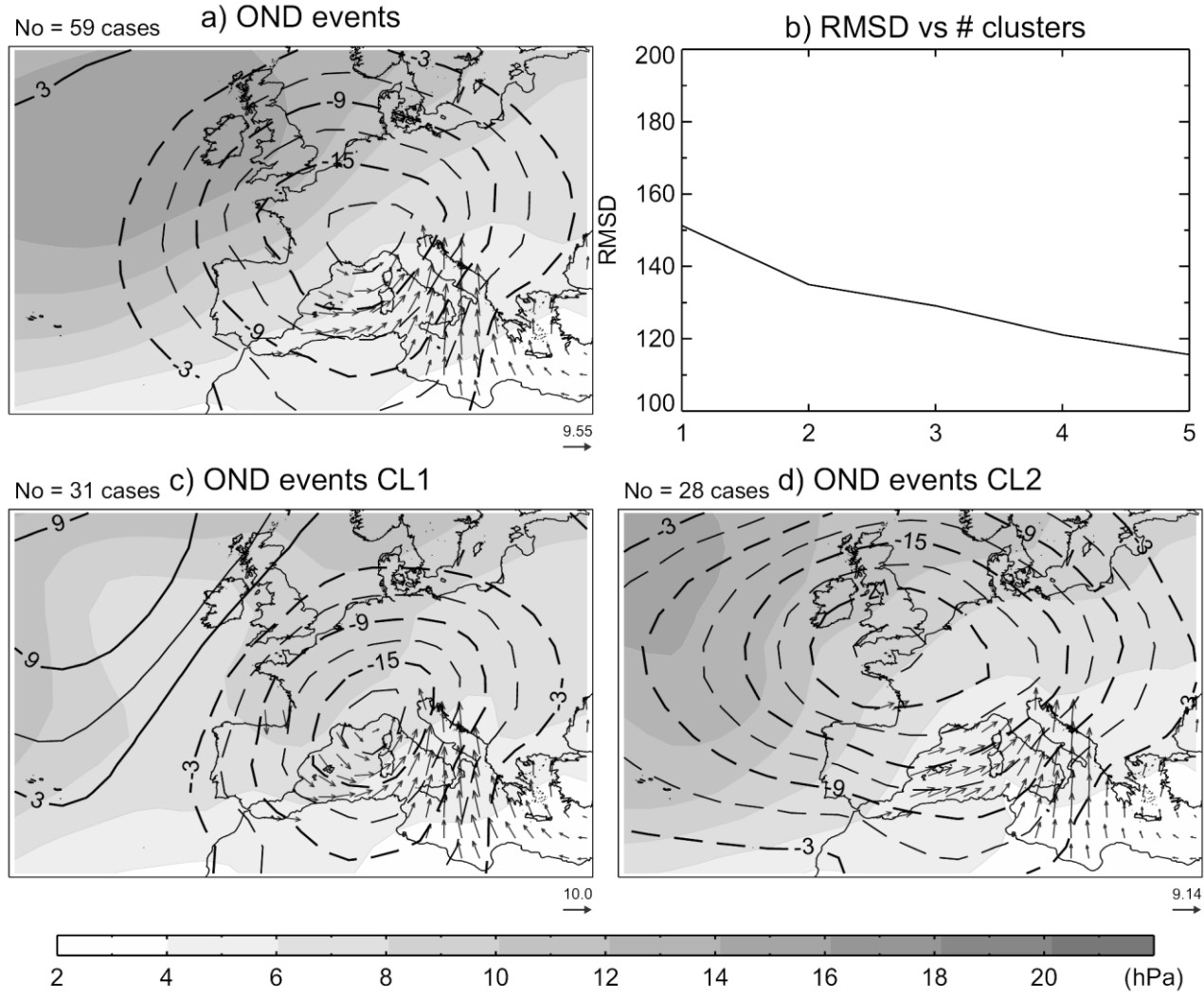


**Figure 7 a) Composite of daily anomalies of sea level pressure (MSLP) over [30,60]°N, [30°W,30°E] (contours, hPa) and 10 m wind vector over the Mediterranean sea (arrows, ms[-1]) for autumn (October-November-Decmber, OND) daily mean detrended water heights above the 99.5th percentile of the 1948-2018 distribution. Shading shows the standard deviation of the composited MSLP fields. The number of cases is shown in the top left corner. The modulus of a reference wind speed vector is shown in the bottom right corner. Panel b) Root mean squared difference (RMSD) of the daily standardized anomalies of MSLP and 10 m wind vector as a function of the number of clusters. RMSDs are computed with respect to the centroid of the respective cluster; Panels c, d) as a) but when surge events are split in two groups, referred to as cluster one (CL1) and two (CL2), which correspond to the choice of two clusters in b). Note that a) is equivalent to considering one cluster with all events. Data sources: NCEP/NCAR reanalysis (Kalnay et al. 1996) and Fabio Raicich (Raicich 2015)**

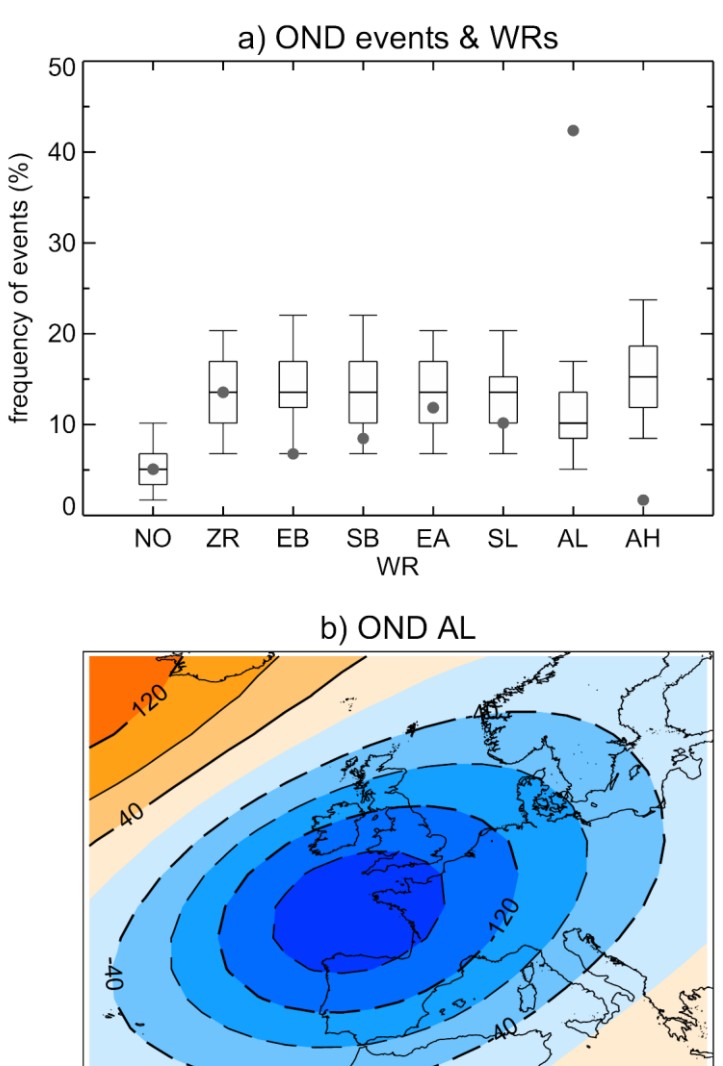

1048

**Figure 8. Top panel: Relative frequency of autumn extreme daily mean detrended water heights above the 99.5th percentile of the 1948-2018 (in % with respect to the total number of events) occurring under the given Weather Regime (WR). Whiskers denote the random distributions obtained from a bootstrap of 5000 trials, each one containing the same number of autumn days of the 1948-2018 period as surge events. Boxes denote the inter-quartile distribution, with the median in between, and bars extend from the 5th to the 95th percentile of the random distributions. WRs are defined from daily fields of geopotential height at 500 hPa of the NCEP/NCAR reanalysis over the Euro-Atlantic sector [30, 65]ºN, [30°W, 25°E]. Acronyms stand for: NO: No,(i.e. undefined) WR; ZR: Zonal Regime; EB: European Blocking; SB: Scandinavian Blocking; EA: East Atlantic; SL: Scandinavian Low; AL: Atlantic Low; AH: Atlantic High. See Garrido-Pérez et al. (2019) for further details. Bottom panel: The AL pattern that is associated to the occurrence of more than 40% of extremes. Data sources: NCEP/NCAR reanalysis (Kalnay et al. 1996) and Fabio Raicich (Raicich 2015).**

1059

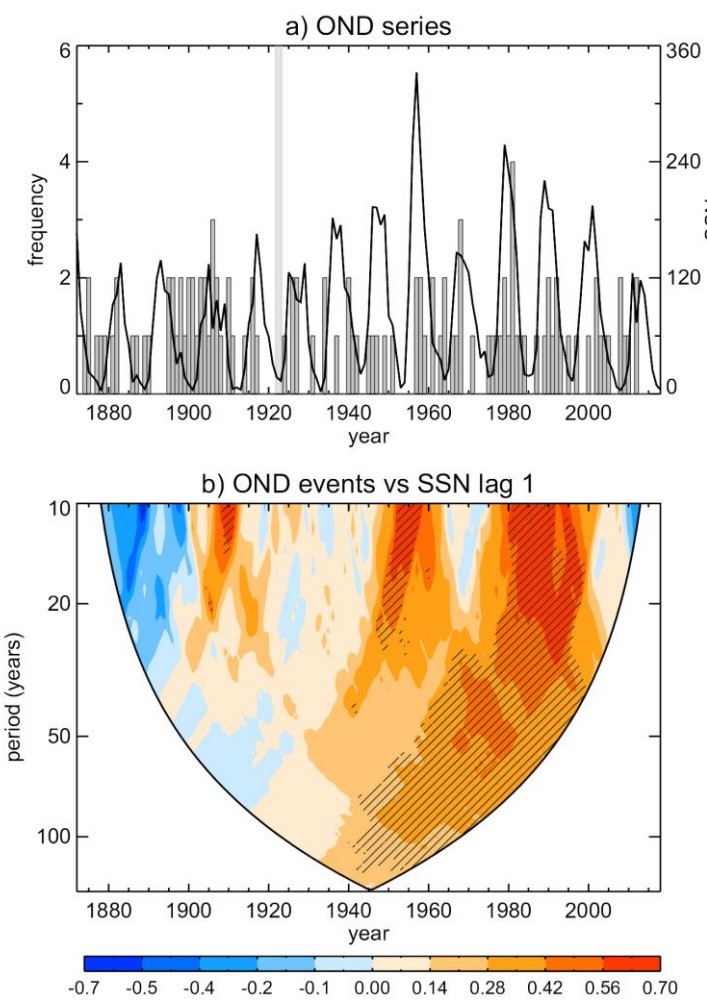

1060

**Figure 9 top panel: Time series of the autumn frequency of independent daily mean detrended water height (IDMWH) extremes for 1872-2018, defined as daily peaks above the 99.5th percentile (35.0 cm) of the total distribution for the period 1924-2018 Daily peaks are required to be separated by more than 72h. Black line shows the autumn mean time series of the SunSpot Number (SSN). Bottom panel: Rank Spearman's (r) correlations between the autumn frequency of independent daily mean detrended water-height extremes and the SSN) for running windows of different width (y-axis) centred at each year of the 1872-2018 period (x-axis). Hatching denotes statistically significant correlations (p<0.05). Correlations are only computed when the sample size is equal or larger than 10 and it exceeds the half size of the window. Panel b) shows the correlation pattern for SSN leading by 1-yr, which produces the largest values. Data sources: WDC-SILSO, Royal Observatory of Belgium, Brussels (see Clette et al. 2014) and Raicich (2015). This figure follows the same approach adopted in Barriopedro et al. (2010), which considered the much shorter 1948-2008 period and the frequency of meteorological surge extremes.**

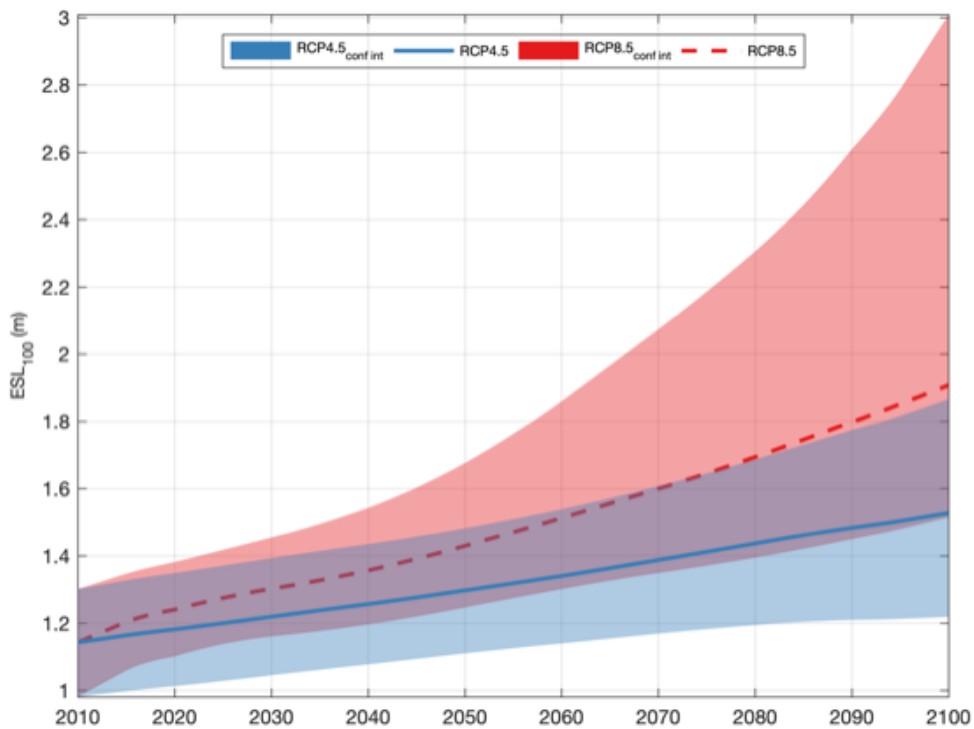


**Figure 10 Time evolution of the 100y-ESL in the North-West Adriatic Sea under RCP4.5 (blue) and RCP8.5 (red). Lines show the corresponding medians and coloured areas express the 5th-95th percentiles (very likely range). This figure follows the same graphic format and it is based on the same simulations shown in Vousdoukas et al (2017), but it specifically refers to the area marked with the red box in figure 1, panel a.**




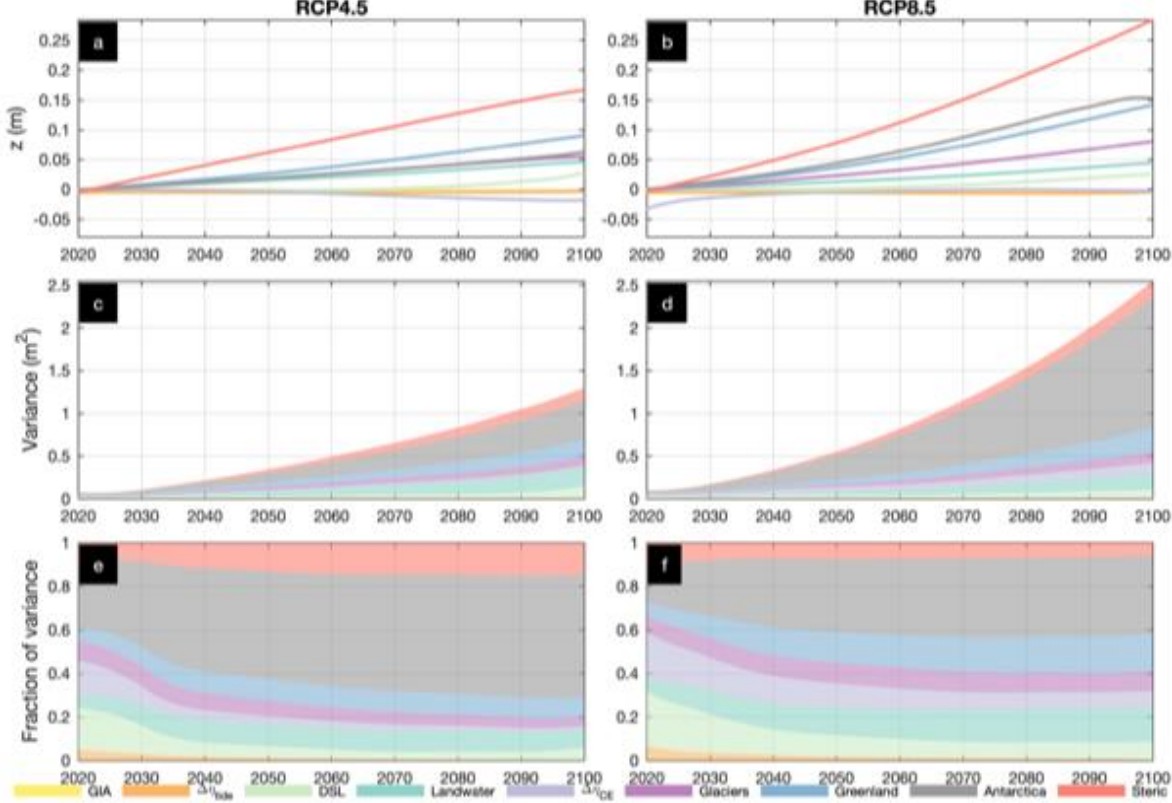


**Figure 11 Break-down of projected 100y-ESL contributions in the North-West Adriatic Sea and of their uncertainty, under RCP4.5 (a, c, e) and RCP8.5 (b, d, f). Projected increase of the 100y-ESL (with respect to the 2001-2020 baseline) from changes in climate extremes, the high tide water level, as well as from SLR contributions from Antarctica, land-water, Greenland, glaciers, dynamic sea level (DSL), glacial isostatic adjustment (GIA), and steric-effects (a, b); variance (in m2) in components (c, d) and fraction of components' variance in global 100y-ESL change. Colors represent different components as in the legend and values express the median. This figure follows the same graphic format and it is based on the same simulations shown in Vousdoukas et al (2018), but it specifically refers to the area marked with the red box in figure 1, panel a.**




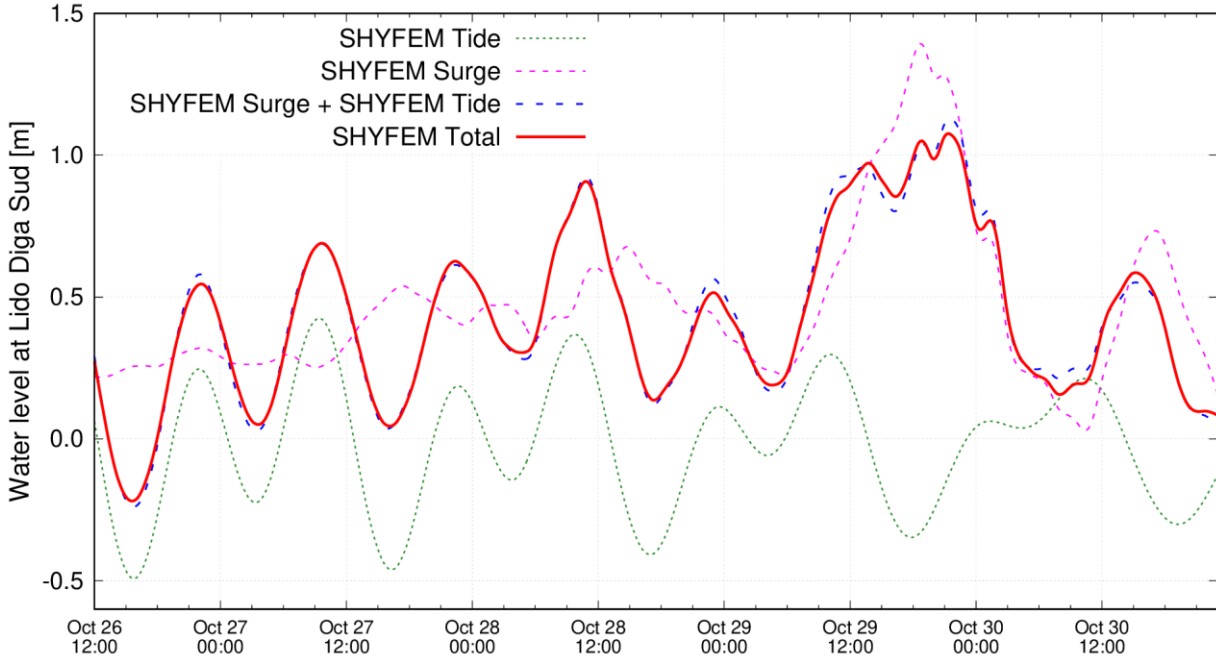

**Figure B1 Simulation of the event A.8 described in Appendix 1 performed with the SHIFEM model (see Cavaleri**
**et a., 2019) for details. The different curves represent the model results using only the astronomical tidal forcing**
**(SHYFEM Tide), the meteorological forcing (SHYFEM Surge) and the full forcing (SHYFEM Total). The dashed**
**line (SHYFEM Surge + SHYFEM Tide) is the algebraic sum of the SHYFEM Tide and SHYFEM Surge results.**

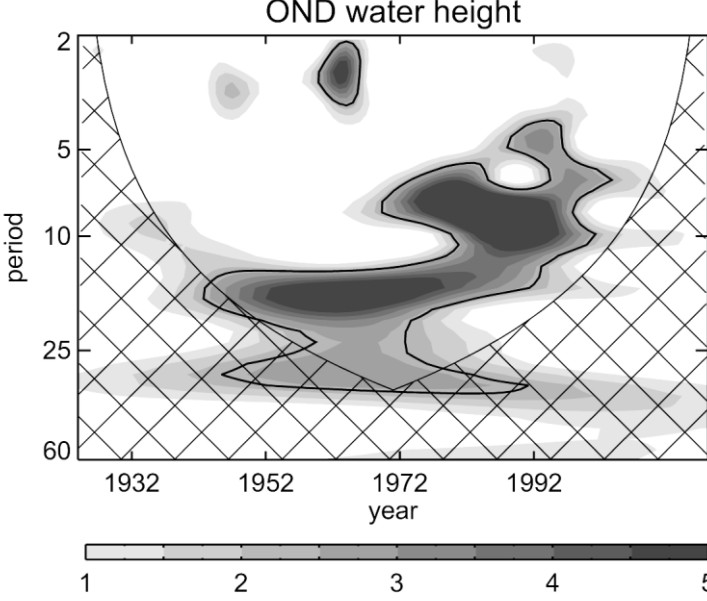


**Figure C1 Wavelet of time series of the autumn (October-November-December) mean water height for 1924-2018,**
**expressed as power values normalized by the variance. Seasonal values are obtained from monthly means of the**
**daily series. All months of this period have less than 10% missing days. Significant power density at 90%**
**confidence level is highlighted by contours. This figure follows the same approach adopted in Barriopedro et al.**
**(2010), which considered the much shorter 1948-2008 period.**

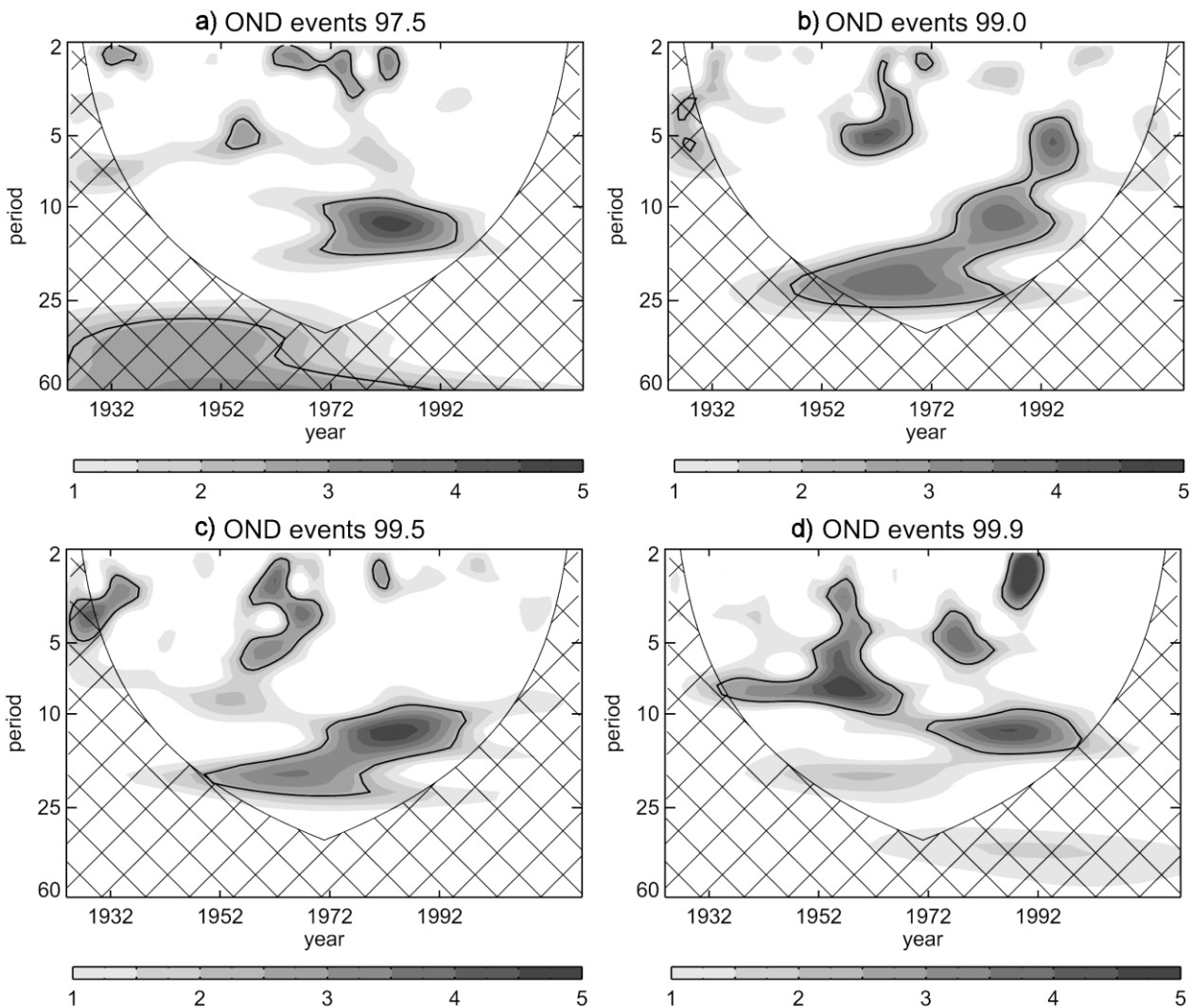


Figure C2 As figure A.2 but for the autumn frequency series of daily detrended water heights above the 97.5th, 99th , 99.5th
and 99.9th percentiles. This figure follows the same approach adopted in Barriopedro et al. (2010), which considered the much
shorter 1948-2008 period and hourly data.

| Datetime [UTC] | Water Height | Astronomical Tide | Seiche | Storm surge | Meteotsunami + MAV setup | PAW surge | IDAS | RSL (19y running mean) | | Meteorological Surge | Detrended water height |
|---|---|---|---|---|---|---|---|---|---|---|---|
| Level (cm) | | | | | | | | | | | |
| 1936-04-16 20:35:00 | 147 | 21 | 15 | 63 | 2 | 26 | 10 | 10 | | 91 | 127 |
| 1951-11-12 07:05:00 | 151 | 43 | 1 | 44 | 3 | 39 | 6 | 15 | | 86 | 130 |
| 1960-10-15 06:55:00 | 145 | 31 | 4 | 63 | 3 | 11 | 13 | 20 | | 77 | 112 |
| 1966-11-04 17:00:00 | 194 | -12 | 22 | 107 | 16 | 20 | 20 | 21 | | 143 | 153 |
| 1968-11-03 06:30:00 | 144 | 35 | 10 | 47 | 2 | 21 | 7 | 22 | | 70 | 115 |
| 1979-02-17 00:15:00 | 140 | 34 | -2 | 39 | 8 | 26 | 13 | 22 | | 73 | 105 |
| 1979-12-22 08:10:00 | 166 | 17 | 15 | 77 | 15 | 14 | 6 | 22 | | 106 | 138 |
| 1986-02-01 03:00:00 | 159 | 30 | 22 | 48 | 4 | 18 | 15 | 22 | | 70 | 122 |
| 1992-12-08 09:10:00 | 142 | 42 | 8 | 30 | 2 | 34 | 3 | 23 | | 66 | 116 |
| 2000-11-06 19:35:00 | 144 | 16 | 7 | 69 | 1 | 17 | 8 | 26 | | 87 | 110 |
| 2002-11-16 08:45:00 | 147 | 44 | -8 | 47 | 1 | 22 | 14 | 27 | | 70 | 106 |
| 2008-12-01 09:45:00 | 156 | 36 | 22 | 41 | 1 | 20 | 6 | 30 | | 62 | 120 |
| 2009-12-23 04:05:00 | 143 | 20 | 32 | 22 | 4 | 18 | 16 | 31 | | 44 | 96 |
| 2009-12-25 03:00:00 | 145 | 30 | 23 | 21 | 3 | 21 | 16 | 31 | | 45 | 98 |
| 2010-12-24 00:40:00 | 144 | 35 | 2 | 39 | 3 | 22 | 12 | 31 | | 64 | 101 |
| 2012-11-01 00:40:00 | 143 | 20 | 1 | 54 | 1 | 27 | 9 | 31 | | 82 | 103 |
| 2012-11-11 08:25:00 | 149 | 47 | -4 | 63 | 2 | 2 | 8 | 31 | | 67 | 110 |
| 2013-02-11 23:05:00 | 143 | 38 | 14 | 39 | 0 | 6 | 15 | 31 | | 45 | 97 |
| 2018-10-29 13:40:00 | 156 | 25 | 2 | 50 | 12 | 29 | 4 | 34 | | 91 | 118 |
| 2018-10-29 19:25:00 | 148 | -31 | 24 | 75 | 13 | 29 | 4 | 34 | | 117 | 110 |
| 2019-11-12 21:50:00 | 189 | 36 | 5 | 42 | 37 | 21 | 14 | 34 | | 100 | 141 |
| 2019-11-13 08:30:00 | 144 | 48 | 4 | 14 | 7 | 23 | 14 | 34 | | 44 | 96 |
| 2019-11-15 10:35:00 | 154 | 47 | 4 | 25 | 2 | 27 | 15 | 34 | | 54 | 105 |
| 2019-11-17 12:10:00 | 150 | 34 | 0 | 35 | 10 | 22 | 15 | 34 | | 67 | 101 |
| 2019-12-23 08:45:00 | 144 | 39 | 39 | 6 | 1 | 14 | 11 | 34 | | 21 | 99 |
| Percentage (%) | | | | | | | | | | | |
| 1936-04-16 20:35:00 | 147 | 14 | 10 | 43 | 1 | 18 | 7 | 7 | | 62 | 86 |
| 1951-11-12 07:05:00 | 151 | 28 | 1 | 29 | 2 | 26 | 4 | 10 | | 57 | 86 |
| 1960-10-15 06:55:00 | 145 | 21 | 3 | 43 | 2 | 8 | 9 | 14 | | 53 | 77 |
| 1966-11-04 17:00:00 | 194 | -6 | 11 | 55 | 8 | 10 | 10 | 11 | | 74 | 79 |
| 1968-11-03 06:30:00 | 144 | 24 | 7 | 33 | 1 | 15 | 5 | 15 | | 49 | 80 |
| 1979-02-17 00:15:00 | 140 | 24 | -1 | 28 | 6 | 19 | 9 | 16 | | 52 | 75 |
| 1979-12-22 08:10:00 | 166 | 10 | 9 | 46 | 9 | 8 | 4 | 13 | | 64 | 83 |
| 1986-02-01 03:00:00 | 159 | 19 | 14 | 30 | 3 | 11 | 9 | 14 | | 44 | 77 |
| 1992-12-08 09:10:00 | 142 | 30 | 6 | 21 | 1 | 24 | 2 | 16 | | 46 | 82 |
| 2000-11-06 19:35:00 | 144 | 11 | 5 | 48 | 1 | 12 | 6 | 18 | | 60 | 76 |
| 2002-11-16 08:45:00 | 147 | 30 | -5 | 32 | 1 | 15 | 10 | 18 | | 48 | 72 |
| 2008-12-01 09:45:00 | 156 | 23 | 14 | 26 | 1 | 13 | 4 | 19 | | 40 | 77 |
| 2009-12-23 04:05:00 | 143 | 14 | 22 | 15 | 3 | 13 | 11 | 22 | | 31 | 67 |
| 2009-12-25 03:00:00 | 145 | 21 | 16 | 14 | 2 | 14 | 11 | 21 | | 31 | 68 |
| 2010-12-24 00:40:00 | 144 | 24 | 1 | 27 | 2 | 15 | 8 | 22 | | 44 | 70 |
| 2012-11-01 00:40:00 | 143 | 14 | 1 | 38 | 1 | 19 | 6 | 22 | | 57 | 72 |
| 2012-11-11 08:25:00 | 149 | 32 | -3 | 42 | 1 | 1 | 5 | 21 | | 45 | 74 |
| 2013-02-11 23:05:00 | 143 | 27 | 10 | 27 | 0 | 4 | 10 | 22 | | 31 | 68 |
| 2018-10-29 13:40:00 | 156 | 16 | 1 | 32 | 8 | 19 | 3 | 22 | | 58 | 76 |
| 2018-10-29 19:25:00 | 148 | -21 | 16 | 51 | 9 | 20 | 3 | 23 | | 79 | 74 |
| 2019-11-12 21:50:00 | 189 | 19 | 3 | 22 | 20 | 11 | 7 | 18 | | 53 | 75 |
| 2019-11-13 08:30:00 | 144 | 33 | 3 | 10 | 5 | 16 | 10 | 24 | | 31 | 67 |
| 2019-11-15 10:35:00 | 154 | 31 | 3 | 16 | 1 | 18 | 10 | 22 | | 35 | 68 |
| 2019-11-17 12:10:00 | 150 | 23 | 0 | 23 | 7 | 15 | 10 | 23 | | 45 | 67 |
| 2019-12-23 08:45:00 | 144 | 27 | 27 | 4 | 1 | 10 | 8 | 24 | | 15 | 69 |
| AVERAGE | | 20 | 7 | 30 | 4 | 14 | 7 | 18 | | 48 | 75 |

**Table 1 List of the extreme water heights (higher than 140 cm) alongside the contributions (see section 2.1 and 2.2):**
**astronomical tide, seiches, storm surge, meteotsunami and lMesoscale Atmospheric Valraibility (MAV) set-up, PAW surge,**
**IDAS variability, Relative Mean Sea Level, meteorological surge and total surge. The water-height values are referenced to the**
**'Zero Mareografico Punta Salute' (ZMPS). The upper panel shows the actual values (cm), the lower panel the percentage (%)**
**of each contribution.**

| Date | MaxWH (cm) | Sources |
|---|---|---|
| 1966-11-04 | 194 | DO68, CA01, BC06, CPSM |
| 2019-11-12* | 189 | ISP20 |
| 1979-12-22 | 166 | CA01, BC06, CPSM |
| 1986-02-01 | 159 | CA01, BC06, CPSM |
| 2018-10-29* | 156 | BC06, ISPRA, CPSM |
| 2008-12-01 | 156 | BC06, ISPRA, CPSM |
| 2019-11-15 | 154 | CPSM |
| 1951-11-12 | 151 | DO61, CA01, BC06, CPSM |
| 2019-11-17 | 150 | CPSM |
| 2012-11-11 | 149 | BC06, ISPRA, CPSM |
| 2018-10-29* | 148 | BC06, ISPRA, CPSM |
| 2002-11-16 | 147 | BC06, CPSM |
| 1936-04-16 | 147 | DO61, BC06, CPSM |
| 2009-12-25 | 145 | BC06, CPSM |
| 1960-10-15 | 145 | DO61, CA01, BC06, CPSM |
| 2019-12-23 | 144 | ISP19 |
| 2019-11-13* | 144 | CPSM |
| 2010-12-24 | 144 | BC06, ISPRA, CPSM |
| 2009-12-23 | 144 | BC06, CPSM |
| 2000-11-06 | 144 | CA01, BC06, CPSM |
| 1968-11-03 | 144 | CA01, BC06, CPSM |
| 2013-02-12 | 143 | BC06, ISPRA, CPSM |
| 2012-11-01 | 143 | BC06, ISPRA |
| 1992-12-08 | 142 | CA01, BC06, CPSM |
| 1979-02-17 | 140 | ISP19 |



**Table A1 List of the surge events higher than 100 cm alongside the respective water-height maxima. The asterisk indicates the**
**two RSL peaks during the same event on 29 October 2018. (AN41 = Annali, 1941; CA01 = Canestrelli et al., 2001; CA19 =**
**Cavaleri et al., 2019, CA20 = Cavaleri et al., 2020, CPSM = CPSM, 2020; DE06 = de Zolt et al., 2006; DO61 = Dorigo, 1961b;**
**DO68 = Dorigo, 1968; ISPRA = ISPRA, 2008-2018; ISPRA/CPSM/CNR = ISPRA et al., 2020)**