# Peer review of "Extreme floods of Venice: characteristics, dynamics, past and future evolution (review article)"

_Natural Hazards and Earth System Sciences, 2020_

## Referee Comment (RC1) · Anonymous Referee #1 · 18 Dec 2020

The paper touches an interesting and relevant topic because of the recent flooding events in Venice and the start of the operation of the Mose barriers. In general terms I think the paper would convey a stronger message provided that:

a. It emphasizes what is actually new in the paper and the advances with respect to previous publications and the state of the art.

b. It discusses the various contributions to sea level in Venice, distinguishing between internal and external variability.

c. The same regarding the non-linear and non-stationary effects in the different sea level components.

More specifically, I include some remarks to strengthen the message of the paper:

[Figure]

1. When talking about the timing of the surges with respect to the astronomical tide and free oscillations, some further discussion on the possibility to lower high sea level by modifying the resonance period could be interesting. As a matter of fact, part of the literature on how to avoid harbour long wave resonance could be applied in here.

2. When presenting (section 2.1) the astronomic tides and other components, it would be nice to relate the maximum tidal amplitudes at the Northern shore of the basin to antinodes.

3. When presenting (section 2.2) the different contributions to relative extreme sea level the paper would benefit from a more in depth discussion regarding the non-linear feeding between the different components. In particular discussing the cases of constructive versus distractive interference.

4. When presenting (section 2.2) the RSL peaks with at least 1.40m that occurred in November 2019 on 12, 13, 15, 17 I think this deserves a further discussion on the possibility of a 2-day resonance period.

5. When discussing the propagation of the sea level signal into the interior of the lagoon (section 2.3) it appears that the surge signal propagates nearly without damping while other sea level components appear to experience significant damping. This should be discussed more in depth and explained in "physical" terms.

6. When presenting the evolution of mean sea level pressure fields during intense surge events (section 3.1) further discussion on the cause-effect relationship between the position of the low pressure centre and the pressure gradient could provide an alternative to numerical model predictions and also advance the understanding on these phenomena.

7. When discussing the future evolution of extreme sea levels (section 4.2) some further discussion on the projection of surges should be included. In fact, in a former section, mention is made on the possibility that surges will get smaller while here the

possibility of non-significant changes or significant reduction are presented on an equal basis. This evolution of surges should also be related to the projections of storm trajectories and peak intensities in this part of the Mediterranean, since both are expectedly related.

8. When discussing (section 5) the limited consideration of wave set up so far I think it would be worthwhile to add the role of infra gravity waves both cross shore and edge waves since they can also contribute small variations to mean sea level which may be critical for the resulting flooding damages.

9. When presenting (lines 430 and following) the projected attenuation of storm surges again it should be discussed whether the projection represents a decreasing trend and relating that to the projected wave conditions.

10. Finally there are some "typos" that should be corrected (e.g. line 36 or line 265).

---

## Referee Comment (RC2) · Anonymous Referee #2 · 25 Jan 2021

The preprint paper (#nhess-2020-359), submitted to EGU's journal *Natural Hazards and Earth System Sciences (NHESS)*, presents an effort to review several aspects of the flood generating mechanism in Venice city center, i.e., by superposition of astronomical tides, seiches, storm surges, meteotsunamis, etc. All these factors are reviewed in terms of individually and/or cooperatively contributing to intense and/or extreme sea level elevation events, respectively. The main outcome of the review focuses on the following findings:

Extreme sea level events are mostly related to storm surges due to Sirocco winds, revealing a characteristic seasonal cycle, with the most important events occurring from November to March. The most intense historical events have been produced by western Mediterranean cyclogenesis, e.g., the Gulf of Genoa. Only a few extreme events of sea levels are caused by atmospheric circulation patterns deriving from the Euro-Atlantic sector. Tidal effects of the 11-year solar cycles appear to mildly contribute to sea level extremes, hidden by the rest of the factors. Relative sea level rise seems to drive a frequency increase of extreme sea levels 1850. Consecutively, it is assessed that the intensity and duration of flood events on Venice in the 21st century, will be affected by possible regional mean sea level rise (MSLR) equalizing and even overcoming the probable enfeeblement of extremes due to the projected storminess attenuation until 2100. High uncertainty of the evolution of global-scale factors inducing MSLR, such as mass contributions in the Mediterranean due to Antarctica and Greenland ice melting, does not help in robustness of future projections. Extreme value analysis based on RCP climate projections provides estimations of increase up to 65% and 160% in 2050 and 2100, respectively, for the 100-year return level events at the North Adriatic coast. Geological and geotechnical factors, such as local subsidence due to tectonics or coastal aquifer drainage or overexploitation, are not discussed at all in terms of future increase of extreme flooding.

This is an interesting overall review endeavor of a very significant scientific and social issue with particular local interest, but the paper in its current form does not support scientific innovation, as it does not add new knowledge of permanent value on the subject of coastal flooding in general; it presents only a few new insights on previous findings for the Venice study area. The paper mainly recapitulates and tries to interblend existing knowledge from very remarkable past articles, with a specialized focus on certain aspects of the presented

problem, by world experts on the field. Yet, in its current form, it does not build robust new arguments on the investigated subject. I believe that if the Editors should consider its publication, at least a major revision should take place, rewriting most of its parts supported by novelty aspects and fresh findings. Some graphs should be omitted (as they are reported elsewhere or refer to previously published literature) and new methodologies of interconnecting the existing knowledge should be proposed and applied. Moreover, some clarifications on the followed approaches are also deserved.

In the following, I present my major comments and some specific remarks in tandem with editorial changes and spellcheck needed.

**Major Comments:**

1) The paper is actually a full **review** of all the met-ocean physical parameters and mechanisms contributing to high sea levels and eventually the generation of Venice floods. This should be **clearly stated in the Title**. This is not a Research Article.

2) Flooding phenomena are by default considered as **extreme events** in literature, yet the authors present proper **analysis** based on univariate **extreme value theory only for the storm surges** and not all the other components of sea level variations. This perspective undermines the notion of a compound event. Moreover, the **wave-induced** component, i.e., the **run-up**, adding to the total sea-surface height, especially near the coastal front, is left out. Of course, its influence is limited to areas near the waterfront, whereas all the other components (surges, RSLR, etc.) can cause spatially extended inundation, yet all the above need to be discussed and explained to the reader.

3) Figure 4: Please elaborate on the **storm track algorithm** and its previous validation. Does it treat **proper identification of secondary lows in the wake of e.g., "Medicanes"**, as NASA's storm track algorithm to **avoid double backing** of storm center on itself over the course of 24 hours. Please further discuss the use of storm tracking technique.

4) The **wind patterns** are in the Adriatic are defined as a crucial factor of surge-driven flood dynamics, but **no data is presented to back this up**. Thus, some kind of wind maps in extreme cases or anything else would help to relate wind set-up to certain flood events.

5) The **geological and geotechnical aspects** of Venice floods are totally overlooked, yet the **low elevation of terrestrial land** in the Venice area is the **main factor for inundation**, rather than changes in storminess patterns. It is reported in many papers that constant **geotectonic land subsidence** and potential overexploitation of coastal aquifers may drive sediment **settlement** and the urban environment's ever-evolving **land sinking below MSL**.

6) Lines 169-176: present a classic methodology for signal processing of timeseries to separate storm surges, PAWs, meteotsunamis and IDAS, but the **choice of cut-off frequency** seems arbitrary, as the eventual durations of the reported phenomena are not "physically" fixed. These are known to occur at similar frequency bands (**overlapping frequencies between several components**) and this makes it difficult to discriminate between different phenomena, especially between surges and meteotsunamis. This should be at least discussed in terms of results' robustness.

7) Lines 183-185: Are the authors sure that these are separate events? Which is the **methodology of discrimination** used? Defining the same event (with several peaks) as multiple cases may insert bias to the statistics of extremes.

8) In general, the submitted paper **feels more like a report** (Figures can be enhanced) or a **review** more than a new research paper. Therefore, all past data on the reported phenomena should be "sewed" together in a comprehensive narrative with new clear scientific insight on the specifics of coastal inundation in Venice city center.

**Specific Comments:**

1) Some literature of storm surges, waves, climatology, cyclogenesis, extremes etc. in the Mediterranean could be added to the state-of-the-art:

*Bengtsson et al. (2006). Storm tracks and climate change. J Clim 9(15): 3518-3543.*

*Calafat et al. (2012). Comparison of Mediterranean sea level variability as given by three baroclinic models. J Geophys Res 117, C02009.*

*Campins et al. (2011). Climatology of Mediterranean cyclones using the ERA-40 dataset. Int J Climatol 31(11): 1596-1614.*

*Makris et al. (2016). Climate Change Effects on the Marine Characteristics of the Aegean and the Ionian Seas. Ocean Dyn, 66(12): 1603–1635.*

*Fernández-Montblanc et al. (2019). Towards robust pan-European storm surge forecasting. Ocean Mod, 133: 129-144.*

Line 44: No keywords are provided.

Lines 23, 93, 98: The authors refer to planetary waves (e.g., Rossby and Kelvin waves) in the Mediterranean. Maybe this terminology could be avoided to prevent possible misinterpretations. The Kelvin waves' mechanics could be approximately used to interpret small sea-level oscillations induced by large-scale **tidal motions** in the **elongated Adriatic**, but **classic planetary wave** motions usually refer to **equatorial Rossby waves** in global scale basins, such as the Atlantic Ocean etc., rather than a closed, marginal, regional aquatic body, as the Mediterranean Sea. Planetary waves depend heavily on global thermoclines etc. and their periods of oscillation are of monthly or yearly scales. This is hardly the case in the Mediterranean. The PAWs referred to in Lines 130 and on, are well-established motions, but more likely treated as meteorologically driven long waves of fine temporal scales (hours to days) rather than actual planetary waves. The authors themselves do a good job clarifying that in Lines 133-135.

Line 24 and *elsewhere in the text*: The authors use the term Sea Level Anomaly (SLA) for any sea level variation investigated in the paper. However, in literature, the SLA term usually refers to large-scale long-term (even to climatological scales of analysis) deviations of the Mean Sea Level (MSL) from earth's geoid, not the episodic, short-term, meteorologically induced, coastal sea level elevations that the paper mainly discusses. According to NOAA, *"A sea level anomaly reveals the regional extent of anomalous water levels in the ocean that occurs when the 5-month running average of the interannual variation is at least 0.1 meters (4 inches) greater than or less than the long-term trend. The interannual variation is the monthly MSL after the trend and the average seasonal cycle are removed. The anomalies are usually mapped by month, using the mid-point of the 5-month running average. When the 5-month average is more than 0.1 meters above the trend, it is indicated as a positive anomaly..."*. Thus, I would recommend using the term sea surface height or anything similar.

Line 51 and 149: please provide the reference period of determining the RSLR in Venice. It is essential information for determining the robustness of the values presented, depending on the timeframe of continuous observations.

Lines 276-282: Please elaborate on the methodology used here (RMSD of which parameter, explain k-means analysis, etc.). Are the authors sure that these are extreme events? Is the analysis based on some robust EVA method? Moreover, please explain how this correlates with the Venice flood events? It seems more of a cyclogenesis-surge association.

Line 334-337: these statements seem like speculations, not numerical facts. Please elaborate or rephrase.

Line 354-358: This is an issue of time-framing, i.e., the choice of the right temporal window to trace statistically significant trends. Moreover, the approach based on stationarity or non-stationarity is also a big issue. Please elaborate and discuss further.

Line 407: All the presented material is in the form of an inventory of past hard work reported in older papers, but it does not integrate all the datasets together in a composite, coherent way to produce new knowledge on Venice flooding.

Lines 418-422: This analysis seems irrelevant to Venice city center floods and inundation of the surrounding areas. The wave set-up is a surf zone sea-level parameter in nearshore areas, but what would be important for flooding is the wave run-up on the coast. This is a different task to perform as it would require a huge amount of beachfront and coast cross-sections treated with several different empirical relations depending on run-up calculation over engineered or natural beach types. Furthermore, as high waves are dissipated by depth-limited breaking and the specifics of the Venice lagoon topography do not allow very high waves to attack the waterfront, wave-induced would not be a crucial factor.

Figure 7: The Figure's results are most likely reported as is in Barriopedro et al. 2010.

Figure 8: This kind of information is already reported in Vousdoukas et al. (2017).

Figure 9: **Plagiarism detected**. This is the exact same Figure as Vousdoukas et al. (2018)'s Fig. 5. Moreover, it is not clear how this Figure's results correlate with local or even regional projections of mean Sea Level Rise focused on the northern Adriatic.

Figure A.1: **Plagiarism detected**. This is the same Figure as Barriopedro et al. (2010)' Fig. 2. It is also not clear how these results relate to Venice city center floods.

Table 1: This is an interesting feature. Percentages of contribution by each component to the total RSL would be beneficial to the reader for supervisory purposes. Is that new knowledge or reported elsewhere? It should be clarified.

**Editorial Comments:**

Lines 25-26: correct "by storm surges produced by the inverse barometer effect and enhanced locally by the Sirocco winds".

Lines 50-55: Repetition of "*damages and losses of a unique monumental and cultural heritage*" expression.

Line 72: Correct "*relevant **to** extreme sea levels*".

Line 74: "*factors(see*" typo.

Line 111: "***is ceases***" typo.

Lines 117-126: better use *atmospheric pressure* than *air pressure*.

Line 162: fix RSL term repetition (use only abbreviation), "…*when it exceeds 80 cm*"

Line 200: change expression to something like "*with a rather moderate storm surge value*"

Lines 232-233: please rephrase (*region is one of the areas…. in the region..* repetition)

Line 266: correct *in **the** Atlantic*

Correct the way of citations' writing in the main text and specifically the use of parentheses e.g., in Lines 78, 251, 258, 268, 304, 340, 352, etc.

Lines 272-275: This sentence is rather vague, please elaborate further on the reported findings of previous studies and how they specifically correlate with Venice city center floods.

Lines 295-296: Please rephrase or clarify. Do the authors refer to the synoptic-scale wave train of very low pressures in central Europe and how they drive frequent high autumn surges?

Lines 308-309: Define *rho* (the Spearman's correlation?) in the main text. In terms of which parameters?

Line 320: correct "how such **a** small"

Line 344-345: How is the frequency determined? As event/year?

Line 375: maybe use "optimistic" than "low" for RCP2.6

Line 378: correct "depend **on**"

Line 390: correct "spatially **averaged**"

Line 387-392: More information about the employed EVA method is needed.

Line 863: Explain OND as October-November-December seasonal period.

Figure 5: Write No. of cases on the graphs before the actual numbers.

Figure 7: Clette et al. 2014 is missing from the Reference list.

Line 897: rephrase "Heavy".

Line 918: correct "As Figure **A.1**".

---

## Author Comment (AC1) · 18 Mar 2021

**Author responses to Reviewer#2 comments for the manuscript:**
**"Extremes floods of Venice: characteristics, dynamics, past and future evolution"**

*In the text below, Reviewer's comments are in bold characters, authors' responses are in slant characters.*

**The preprint paper (#nhess-2020-359), submitted to EGU's journal *Natural Hazards and Earth System Sciences (NHESS)*, presents an effort to review several aspects of the flood generating mechanism in Venice city center, i.e., by superposition of astronomical tides, seiches, storm surges, meteotsunamis, etc. All these factors are reviewed in terms of individually and/or cooperatively contributing to intense and/or extreme sea level elevation events, respectively. The main outcome of the review focuses on the following findings:**

**Extreme sea level events are mostly related to storm surges due to Sirocco winds, revealing a characteristic seasonal cycle, with the most important events occurring from November to March. The most intense historical events have been produced by western Mediterranean cyclogenesis, e.g., the Gulf of Genoa. Only a few extreme events of sea levels are caused by atmospheric circulation patterns deriving from the Euro-Atlantic sector. Tidal effects of the 11-year solar cycles appear to mildly contribute to sea level extremes, hidden by the rest of the factors. Relative sea level rise seems to drive a frequency increase of extreme sea levels 1850. Consecutively, it is assessed that the intensity and duration of flood events on Venice in the 21st century, will be affected by possible regional mean sea level rise (MSLR) equalizing and even overcoming the probable enfeeblement of extremes due to the projected storminess attenuation until 2100. High uncertainty of the evolution of global scale factors inducing MSLR, such as mass contributions in the Mediterranean due to Antarctica and Greenland ice melting, does not help in robustness of future projections. Extreme value analysis based on RCP climate projections provides estimations of increase up to 65% and 160% in 2050 and 2100, respectively, for the 100-year return level events at the North Adriatic coast. Geological and geotechnical factors, such as local subsidence due to tectonics or coastal aquifer drainage or overexploitation, are not discussed at all in terms of future increase of extreme flooding.**

**This is an interesting overall review endeavor of a very significant scientific and social issue with particular local interest, but the paper in its current form does not support scientific innovation, as it does not add new knowledge of permanent value on the subject of coastal flooding in general; it presents only a few new insights on previous findings for the Venice study area. The paper mainly recapitulates and tries to interblend existing knowledge from very remarkable past articles, with a specialized focus on certain aspects of the presented problem, by world experts on the field. Yet, in its current form, it does not build robust new arguments on the investigated subject. I believe that if the Editors should consider its publication, at least a major revision should take place, rewriting most of its parts supported by novelty aspects and fresh findings. Some graphs should be omitted (as they are reported elsewhere or refer to previously published literature) and new methodologies of interconnecting the existing knowledge should be proposed and applied. Moreover, some clarifications on the followed approaches are also deserved.**

**In the following, I present my major comments and some specific remarks in tandem with editorial changes and spellcheck needed.**

*We thank Reviewer#2 for the comments on our manuscript. Please, find below our answers.*

**Major Comments:**
**1) The paper is actually a full review of all the met-ocean physical parameters and mechanisms contributing to high sea levels and eventually the generation of Venice floods. This should be clearly stated in the Title. This is not a Research Article.**

*This is a review article. In fact, the manuscript is classified as a "Review article" in the journal submission system. We thought this was clear considering the initial sentence of the manuscript "This paper reviews current understanding on the extreme water levels that are responsible for the damaging floods affecting the Venice city center ...". Further, the description of this special issue, which is available in the journal web page, clearly writes that "This special issue is composed of three review papers, addressing three different and complementary aspects of the hazards causing the flood of Venice. Review paper 1 describes the tools [...] Review paper 2 describes the factors leading to extreme events, their past evolution, and expected future levels under a climate change perspective. Review paper 3 considers the evolution of the mean relative sea level [...]". We apologize for having missed "Review article:" in the manuscript title and sorry for the following confusion. In order to further emphasize that this is a review article and avoid any misunderstanding, we agree that this information should be added to the revised title.*

**2) Flooding phenomena are by default considered as extreme events in literature, yet the authors present proper analysis based on univariate extreme value theory only for the storm surges and not all the other components of sea level variations. This perspective undermines the notion of a compound event. Moreover, the wave-induced component, i.e., the run-up, adding to the total sea-surface height, especially near the coastal front, is left out. Of course, its influence is limited to areas near the waterfront, whereas all the other components (surges, RSLR, etc.) can cause spatially extended inundation, yet all the above need to be discussed and explained to the reader.**

*A main novelty resulting from this review is the importance of the superposition of several different factors for understanding extreme sea levels. However, this is a review and NOT a research paper and being based on existing literature cannot include a multivariate probabilistic analysis, as this is not available in the scientific literature. We agree that a multivariate approach is important and we have added to the conclusions that "Furthermore, a multivariate statistical model that describes extreme water levels as a function of the various contributions would provide a more complete characterization of extreme water levels.". However, to perform such an analysis is beyond the scope of this review paper.*

*The comment of the reviewer shows that, indeed, a more extended explanation on the lack of relevance of wave induced components for the floods of Venice will be beneficial.*
*The city of Venice is located in the center of a large and shallow lagoon (Fig.1), with an approximate extension of 500km2 and an average depth of about 1 meter. The lagoon is connected to the Adriatic Sea by three inlets (500-1000m. wide and from 8 to17m. deep), through which high water levels propagate from the open sea, along a complex pattern of very shallow areas and canals (from 2 to 20 meters deep) to the city center. The lagoon is separated from the sea by two long (about 25 km in total) narrow (less than 200m average width) and sandy islands, reinforced with artificial barriers in the most vulnerable parts. The elevation of these islands is such that they separate the lagoon from the open sea also during the most extreme events, with the exception of the 4th November 1966 flood, when they were breached in several points.*
*The floods of Venice do not occur because water overtops coastal barriers or defenses. Therefore, wave run-up and infra-gravity waves and nearshore processes (though certainly relevant along the sea-side front of Lido under some conditions) have never been considered when computing sea level extremes inside the lagoon. Wave set-up at the Adriatic shore has been estimated only during some extreme events (e.g De Zolt et al., 2006), but not inside the lagoon inlets.*
*The elevation of the natural barriers separating the lagoon from the Adriatic Sea has so far prevented overtopping caused by wave run-up and infragravity-waves, except in the 1966 flood when waves may have contributed to total water levels. In the future this is unlikely to change as barrier islands will continue being protected by coastal defences and maintained by beach nourishment. Hence, waves do not need to be considered. It cannot be excluded that these factors will become relevant in case of extreme sea level rise in the future, but present evidence is that waves do not need to be considered.*

*A new version of Fig. 1 with a map of the lagoon and the three paragraphs above will be added to the introduction, section 2.3 and the conclusions, respectively.*

[Figure]

*Figure 1: left panel: bathymetry of the Adriatic Sea with the position of Venice and arrows denoting the directions of the two main wind regimes affecting the North Adriatic. Right panel: morphology of the lagoon of Venice with the three inlets connecting it to the Adriatic Sea, and the position of the city and of Chioggia. The red box (which includes the whole lagoon in its northern part) denotes the area represented by the data in Figs. 8 and 9.*

**3) Figure 4: Please elaborate on the storm track algorithm and its previous validation. Does it treat proper identification of secondary lows in the wake of e.g., "Medicanes", as NASA's storm track algorithm to avoid double backing of storm center on itself over the course of 24 hours. Please further discuss the use of storm tracking technique.**

*The tracking scheme analyses MSLP or geopotential gridded fields, it identifies the pressure minima, the location where a cyclogenesis occurs and the following trajectory of the pressure minimum by associating low-pressure centers in successive maps by a minimum distance criterion. Shallow secondary minima with a small area are absorbed in the large nearest system. It has been already used in numerous studies assessing the climatology of Mediterranean cyclones such as Lionello et al. (2016) and Flaounas et al. (2018), in the IMILAST tracking scheme intercomparison analysis (Neu et al., 2013) and in a dedicated study considering the synoptic patterns leading to high water levels along the coast of the Mediterranean Sea (Lionello et al., 2019). We suggest that readers are addressed to those studies for details on the method and how it compares with other tracking schemes. We will add this information when discussing Fig. 4.*

*References*

- *Lionello P., Isabel F. Trigo, Victoria Gil, Margarida L. R. Liberato, Katrin M. Nissen, Joaquim G. Pinto, Christoph C. Raible, Marco Reale, Annalisa Tanzarella, Ricardo M. Trigo, Sven Ulbrich & Uwe Ulbrich (2016) Objective climatology of cyclones in the Mediterranean region: a consensus view among methods with different system identification and tracking criteria, Tellus A: Dynamic Meteorology and Oceanography, 68:1, DOI: 10.3402/tellusa.v68.29391*
- *Flaounas E., Fanni Dora Kelemen, Heini Wernli, Miguel Angel Gaertner, Marco Reale, Emilia Sanchez-Gomez, Piero Lionello, Sandro Calmanti, Zorica Podrascanin, Samuel Somot, Naveed Akhtar, Raquel Romera, Dario Conte. (2018) Assessment of an ensemble of ocean–atmosphere coupled and uncoupled*

*regional climate models to reproduce the climatology of Mediterranean cyclones. Climate Dynamics 51:3, pages 1023-1040.*

- *Lionello, P., Conte, D., and Reale, M. (2019) The effect of cyclones crossing the Mediterranean region on sea level anomalies on the Mediterranean Sea coast Nat Hazards Earth Syst Sci 19:1541–1564, DOI:10.5194/nhess-19-1541-2019*
- *Neu U, Akperov MG, Bellenbaum N, Benestad R, Blender R, Caballero R, Cocozza A, Dacre HF, Feng Y, Fraedrich K, Grieger J, Gulev S, Hanley J, Hewson T, Inatsu M, Keay K, Kew SF, Kindem I, Leckebusch GC, Liberato MLR, Lionello P, Mokhov II, Pinto JG, Raible CC, Reale M, Rudeva I, Schuster M, Simmonds I, Sinclair M, Sprenger M, Tilinina ND, Trigo IF, Ulbrich S, Ulbrich U, Wang XL, Wernli H (2013) IMILAST – a community effort to intercompare extratropical cyclone detection and tracking algorithms: assessing method-related uncertainties. Bull Am Met Soc, 94:529-547. doi:10.1175/BAMS-D-11-00154.1*

*None of these studies was meant to analyze medicanes. The capability of this  tracking algorithm on detecting small features, such as medicanes, depends on the tuning of the parameter controlling the merging of small secondary systems in large circulation structures. However, we feel that this discussion is not really relevant for this subsection, though it might be considered in future studies in the wake of the analysis of the 19 November 2019 event.*

**4) The wind patterns in the Adriatic are defined as a crucial factor of surge-driven flood dynamics, but no data is presented to back this up. Thus, some kind of wind maps in extreme cases or anything else would help to relate wind set-up to certain flood events.**

*Following this comment of the Reviewer,  Figs. 2 and 3 have been modified in order to show the wind fields over the Adriatic Sea during the evolution of the floods*

[Figure]

*Figure 2: The large panels show the composite of SLP fields based on ERA5 (in hPa, left color bar) datasets associated with storm surges higher than 50 cm in Venice (see Table 1). Small panels show the corresponding wind fields over the Adriatic Sea (m/s, right color bar). The time lags chosen for the composites are 36, 24 , 12 hours before and 12, 24 hours after the peak of the event. The green dot shows the location of the city of Venice.*

[Figure]

*Figure 3: Same as figure 2, except it is based on the events in Table 1 with storm surge height lower than 50 cm*

**5) The geological and geotechnical aspects of Venice floods are totally overlooked, yet the low elevation of terrestrial land in the Venice area is the main factor for inundation, rather than changes in storminess patterns. It is reported in many papers that constant geotectonic land subsidence and potential overexploitation of coastal aquifers may drive sediment settlement and the urban environment's ever-evolving land sinking below MSL.**

*Loss of land level, subsidence and, in general, vertical land motions have been a key factors for increasing the vulnerability of the city in the past century contributing to about 50% of the observed RSL rise. This review article is a part of a special issue where another contribution (Zanchettin et al., 2021, nhess-2020-351) discusses extensively relative sea level rise including estimates of the past and future role of subsidence. This was, probably too shortly, mentioned in the RSLR paragraph in section 2.1. We will add a reference to it in the introduction.*

**6) Lines 169-176: present a classic methodology for signal processing of timeseries to separate storm surges, PAWs, meteotsunamis and IDAS, but the choice of cut-off frequency seems arbitrary, as the eventual durations of the reported phenomena are not "physically" fixed. These are known to occur at similar frequency bands (overlapping frequencies between several components) and this makes it difficult to**

**discriminate between different phenomena, especially between surges and meteotsunamis. This should be at least discussed in terms of results' robustness.**

*A number of processes contribute to the sea-level variability and their separation enables the extreme sea levels to be interpreted. When it comes to the signals characterized by peaks in the sea-level spectra (tides, seiches), the procedure is straightforward: they are readily isolated by applying the band-pass filters around the known frequencies. The situation is more complicated when considering the response of sea level to the atmospheric forcing because it is characterized by a continuous spectrum. However, it is still possible to distinguish between various processes contributing to the continuum due to their relationship with the mid-latitude atmospheric phenomena. Most often, the distinction is made on the basis of different space scales of the phenomena. Among a number of classifications developed for the purpose, one of the simplest appears to be that proposed by Holton (2004): he makes a distinction between the planetary-scale motions, of $O(10^7 m)$, the synoptic-scale motions, of $O(10^6 m)$, and the mesoscale motions, of $O(10^4 m) - O(10^5 m)$. Moreover, he points out that the planetary atmospheric waves tend to move westward against the eastward zonal flow and are therefore characterized by relatively small speeds (1–10 m/s) whereas the synoptic-scale atmospheric systems tend to move eastward in the mean flow and are consequently marked by relatively large speeds (typically 10 m/s). As for the mesoscale atmospheric systems, their speeds are also relatively large, of $O(10 m/s)$ (Markowsky and Richardson, 2010). By allowing for these space scales and speeds, it is easy to show that the time scales of various processes also differ, being for the planetary-scale atmospheric motions of $O(10 days) - O(100 days)$, for the synoptic scale atmospheric motions of $O(1 day)$, and for the mesoscale atmospheric motions of $O(10 minutes) - O(1 hour)$. The differences between the space and time scales and the related speeds reflect the different dynamics controlling the atmospheric phenomena. At the planetary scale, the Rossby wave dynamics prevails. At the synoptic scale, motions are mostly driven by baroclinic instability. Mesoscale processes are either topographically forced or are driven by one of a number of instabilities operating at that scale.*

*The proposed filters are meant to be effective at isolating the processes related to the planetary-scale, synoptic-scale and mesoscale atmospheric phenomena. The selection of filters is further supported by some other findings. Orlić (1983) has performed cross-spectral analysis of the geopotential height of 500 hPa surface above the Adriatic and the sea level. The results showed that the coherence is high at periods surpassing 10 days (at which planetary-atmospheric-wave dynamics dominates) and is much weaker at periods smaller than 10 days (at which baroclinic instability in the atmosphere operates). On the other hand, Markowsky and Richardson (2010) stated that the time scales of mesoscale atmospheric phenomena range from the period of a pure buoyancy oscillation (roughly 10 minutes) to the inertial period (roughly 17 hours in the midlatitudes). That the 10-hour cutoff period allows one to distinguish between the processes related to the synoptic-scale and mesoscale atmospheric variability is also confirmed by Ferrarin et al. (2021): it enabled them to separate a cyclone moving in an eastward direction above the Mediterranean from a low-pressure system travelling in a northwestward direction above the west Adriatic coast and to reveal the difference between the responses of the sea to the two forcing mechanisms.*

*Therefore, while we agree with the Reviewer that establishing fixed thresholds is not possible, the adopted values allow for an effective separation of the different components in the northern Adriatic Sea. A paragraph with these arguments will be inserted in section 2.2*

*References:*

- *Ferrarin, C., Bajo, M., Benetazzo, A., Cavaleri, L., Chiggiato, J., Davison, S., Davolio, S., Lionello, P. Orlić, M., Umgiesser, G. (2021): Local and large-scale controls of the exceptional Venice floods of November 2019, Progress in Oceanography, submitted.*
- *Holton, J. R. (2004): An Introduction to Dynamic Meteorology (Fourth Edition), Elsevier, Amsterdam, 535 pp.*
- *Markowsky, P., Richardson, Y. (2010): Mesoscale Meteorology in Midlatitudes, Wiley-Blackwell, Chichester, 407 pp.*

- *Orlić, M. (1983): On the frictionless influence of planetary atmospheric waves on the Adriatic sea level, Journal of Physical Oceanography, 13, 1301-1306.*

**7) Lines 183-185: Are the authors sure that these are separate events? Which is the methodology of discrimination used? Defining the same event (with several peaks) as multiple cases may insert bias to the statistics of extremes.**

*After the exceptionally high water on 12 November, three successive events with water level values higher than 1.40 m occurred in just five days. As stated in Ferrarin et al. (2020), these events were driven by separate Sirocco wind episodes in succession in the Adriatic Sea, which did not trigger any significant seiche oscillations in the Adriatic Sea. Similarly to what happened on 12 November, these flood events were determined by the overlapping of the maximum meteorological contribution, the tide peak and a persistent above average mean sea level during the month  in the northern Adriatic. This comment will be added in section 2.2*

**8) In general, the submitted paper feels more like a report (Figures can be enhanced) or a review more than a new research paper. Therefore, all past data on the reported phenomena should be "sewed" together in a comprehensive narrative with new clear scientific insight on the specifics of coastal inundation in Venice city center.**

*Several figures will be redrawn considering our answers to the comments of the Reviewers. We insist that this is indeed a review article, whose utility is presenting the available knowledge and gaps,  repeating past analyses using new datasets and achieving a deep insight by merging the outcomes of published papers. We appreciate the synthesis that the Reviewer provided in the second and third paragraphs of the submitted comments and we feel that it shows the effectiveness of our effort.*

**Specific Comments:**
**Some literature of storm surges, waves, climatology, cyclogenesis, extremes etc. in the Mediterranean could be added to the state-of-the-art:**
**Bengtsson et al. (2006). Storm tracks and climate change. J Clim 9(15): 3518–3543.**
**Calafat et al. (2012). Comparison of Mediterranean sea level variability as given by three**
**baroclinic models. J Geophys Res 117, C02009.**
**Campins et al. (2011). Climatology of Mediterranean cyclones using the ERA-40 dataset. Int J Climatol 31(11): 1596–1614.**
**Makris et al. (2016). Climate Change Effects on the Marine Characteristics of the Aegean**
**and the Ionian Seas. Ocean Dyn, 66(12): 1603–1635.**
**Fernández-Montblanc et al. (2019). Towards robust pan-European storm surge forecasting.**
**Ocean Mod, 133: 129-144.**

*Actually we think that these references are not really relevant for this review article. Calafat et al (2012) can eventually be relevant for the companion review on sea level rise and Fernández-Montblanc et al. (2019) for that on the prediction models. Bengtsson et al (2006) and Campins et al. (2011) consider storm tracks and cyclones and opening the related issues would require analysing a large number of papers and deserve a dedicated review article. Makris et al (2016) considers a different geographical area.*

**Line 44: No keywords are provided.**
*Proposed key words are: Venice, extreme events, floods, sea level, climate change, trends*

**Lines 23, 93, 98: The authors refer to planetary waves (e.g., Rossby and Kelvin waves) in the Mediterranean. Maybe this terminology could be avoided to prevent possible misinterpretations. The Kelvin waves'**

mechanics could be approximately used to interpret small sea-level oscillations induced by large-scale tidal motions in the elongated Adriatic, but classic planetary wave motions usually refer to equatorial Rossby waves in global scale basins, such as the Atlantic Ocean etc., rather than a closed, marginal, regional aquatic body, as the Mediterranean Sea. Planetary waves depend heavily on global thermoclines etc. and their periods of oscillation are of monthly or yearly scales. This is hardly the case in the Mediterranean. The PAWs referred to in Lines 130 and on, are well established motions, but more likely treated as meteorologically driven long waves of fine temporal scales (hours to days) rather than actual planetary waves. The authors themselves do a good job clarifying that in Lines 133-135.

*A distinction should be made between the planetary oceanic waves (POWs) and the planetary atmospheric waves (PAWs). We have documented the response of Adriatic and Mediterranean Seas to the forcing provided by PAWs, not the POWs in the Mediterranean and Adriatic Seas and the Reviewer acknowledges that the manuscript is very clear on this. The use of the PAW terminology is very well established in the literature and we do not think this is a source of confusion.*

Line 24 and *elsewhere in the text*: The authors use the term Sea Level Anomaly (SLA) for any sea level variation investigated in the paper. However, in literature, the SLA term usually refers to large-scale long-term (even to climatological scales of analysis) deviations of the Mean Sea Level (MSL) from earth's geoid, not the episodic, short-term, meteorologically induced, coastal sea level elevations that the paper mainly discusses. According to NOAA, "*A sea level anomaly reveals the regional extent of anomalous water levels in the ocean that occurs when the 5-month running average of the interannual variation is at least 0.1 meters (4 inches) greater than or less than the long-term trend. The interannual variation is the monthly MSL after the trend and the average seasonal cycle are removed. The anomalies are usually mapped by month, using the mid-point of the 5-month running average. When the 5-month average is more than 0.1 meters above the trend, it is indicated as a positive anomaly...*". Thus, I would recommend using the term sea surface height or anything similar.

*We agree that SLA can be used for large scale and long term deviations of the mean sea level, but we do not think that our use of SLA in a more general sense is a source of confusion. We have checked our text and it is not possible to replace sea level anomaly(ies) with sea level height(s) in our manuscript, without changing the meaning of the sentences.*

Line 51 and 149: please provide the reference period of determining the RSLR in Venice. It is essential information for determining the robustness of the values presented, depending on the timeframe of continuous observations.

*This is described in the first paragraph of section 2.2: "Since 1919 sea-level values have been referred to the mean sea level over the 1884-1909 period (central year 1897),which is usually called 'Zero Mareografico Punta Salute' (ZMPS), and referred to as relative sea level (RSL)."*

Lines 276-282: Please elaborate on the methodology used here (RMSD of which parameter, explain k-means analysis, etc.). Are the authors sure that these are extreme events? Is the analysis based on some robust EVA method? Moreover, please explain how this correlates with the Venice flood events? It seems more of a cyclogenesis-surge association.

*We used daily time series of MSL from 1872 to 2018 and retained the daily residuals after removing a low-frequency component (6-month filtered MSL). Then, we selected events (peaks above the 99.5th percentile threshold of the residuals series that are separated by at least 3 days), and saved the first day of the event. Considering the typical duration of surges, this approach would capture the occurrence of independent high surge events.*

*For the so-selected days, we used standardized anomalies of SLP over the Euro-Atlantic sector and 10-m wind vector over the Mediterranean Sea. A k-means clustering (e.g. Wilks 2006) of these daily fields was applied to group surge events with similar spatial patterns. Iterating from different initial random seeds, the algorithm proceeds until all days are classified in a given cluster. Clusters are constructed so that differences between the daily patterns are minimized within the same cluster and maximized between the clusters, according to a given distance metric (the sum of squared distances). Each cluster is characterized by its centroid (the composited spatial pattern of SLP and 10-m wind standardized anomalies for all days in the cluster). The RMSD shown in Fig. 5b is therefore the root mean squared difference between the daily standardized fields of SLP and 10-m wind vectors of all surge days and their corresponding centroid. In the revised version we will provide this additional details on the methodology before describing figure 5.*

**Line 334-337: these statements seem like speculations, not numerical facts. Please elaborate or rephrase.**

*Our analyses and our sentence do not demonstrate or reject a solar influence. The statement relies on the new evidence provided in the manuscript (updated assessment of Figure 7). The correlation between solar activity and the frequency of autumn surges reported elsewhere for the late 20$^{th}$ century is herein captured, but it is lost when we use a longer record, and recent measurements for the 21$^{st}$ century. This suggests that solar influences are non-stationary (effects detected for specific periods only), non-linear (e.g. confined to the recent Grand Modern Maximum), or that the apparent association between the 11-yr solar cycle and surge events was only circumstantial (i.e. caused by other factors). We have rephrased the sentence to emphasize that we are describing possible explanations for the new results: "These results suggest that if there is a solar signal it would likely be non-stationary (arguably masked by other sources variability) and/or non-linear (e.g. confined to Grand Maxima of solar activity). The alternative hypothesis is that the decadal variability of extreme surges is due to other causes, including internal variability"*
*As for the final sentence of this section, we do not think that we need numerical facts to support this. It just states that, regardless of the causes of the observed interannual-to-interdecadal variations in the frequency of surge events (solar, internal variability or other), one cannot reject that large variability will also occur in the future. In the revised text we have clarified that we are talking about surges and not flooding, whose future evolution is dominated by an increasing trend. "It is reasonable that, superimposed on the increasing frequency of Venice flooding due to the mean sea level rise, the frequency of extreme surges will experience large interannual-to-decadal variations in the future, as it has been observed in the recent period. However, the causes of this variability are still uncertain".*

**Line 354-358: This is an issue of time-framing, i.e., the choice of the right temporal window to trace statistically significant trends. Moreover, the approach based on stationarity or non-stationarity is also a big issue. Please elaborate and discuss further.**

*This review considers the frequency of floods using an extremely long time series of nearly 150 years duration of daily sea level -- from 1872 to 2018. Its analysis confirms previous studies that after subtracting the long term sea level mean, the frequency of extremes has no sustained trends (at multidecadal time scales) in spite of the presence of large fluctuations at multiple time scales. we have rephrased the sentence as "In summary, the amount of current evidence shows that while the frequency of floods has clearly progressively increased in time after the mid-twenty century, there is no clear indication of a sustained trend at multi-decadal time scales in either the frequency or the severity of extreme meteorological events. "*

**Line 407: All the presented material is in the form of an inventory of past hard work reported in older papers, but it does not integrate all the datasets together in a composite, coherent way to produce new knowledge on Venice flooding.**

*We insist that this is a review article. Further, the discussion of the literature has been complemented by the replica of previous analysis using new datasets (ERA5 in Figs. 2-4), longer time series that have been made recently available (Figs 5-7), extracting information specific for the Venetian floods from recent global datasets*

*(Figs. 8 and 9). We stress that all these figures have never been published before and are based on new data. The assessment of the scientific literature, complemented by the analysis of the most recent events, it has highlighted that the superposition of several different factors is fundamental for extreme sea levels, which was never highlighted in the previous literature. We further reinforce with longer time series that the increase in the frequency of extreme sea levels since the mid 19th Century is explained by relative sea level rise, with no long term trend in the intensity of the atmospheric forcing. Analogously, future regional relative mean sea level rise will be the most important driver of increasing duration and intensity of Venice floods through this century, overwhelming the small decrease in marine storminess projected during the 21 century. This will clearly pose unprecedented challenges for future flood management and the maintenance of effective coastal defences.*

**Lines 418-422: This analysis seems irrelevant to Venice city center floods and inundation of the surrounding areas. The wave set-up is a surf zone sea-level parameter in nearshore areas, but what would be important for flooding is the wave run-up on the coast. This is a different task to perform as it would require a huge amount of beachfront and coast cross sections treated with several different empirical relations depending on run-up calculation over engineered or natural beach types. Furthermore, as high waves are dissipated by depth-limited breaking and the specifics of the Venice lagoon topography do not allow very high waves to attack the waterfront, wave-induced would not be a crucial factor.**

*In general, we are a bit confused by the suggestion of the reviewer that wave run-up is relevant while wave set up is not, being the latter a component of the former, eventually increasing the maximum on-shore elevation reached by the waves. Our answer to comment 2 has clarified why wave run-up is not relevant. The wave set-up (the increase of mean sea level produced by wave breaking) might be relevant if it initiates sufficiently offshore to affect the sea level at the lagoon inlets. This has been argued having happened in the extreme flood of 4 November 1966 and being possible in future extreme storms. This is why it is cited in the review. We will add this short discussion to the paragraph considered in this reviewer's comment.*

**Figure 7: The Figure's results are most likely reported as is in Barriopedro et al. 2010.**

*Figure 7 provides an revisited version of a similar figure in Barriopedro et al. (2010), by considering a much longer series (since 1872) than that employed therein (since 1948), as well as updated data for the last decade (up to 2018, as compared to 2008, in Barriopedro et al. 2010). As stated above (comment on L276-282), our results confirm those reported in Barriopedro et al. (2010) for the second half of the 20th century, but also illustrate new findings (lack of coherence between the frequency of surges and the 11-yr solar cycle during the 21st century and before the mid-20th century).*

**Figure 8: This kind of information is already reported in Vousdoukas et al. (2017).**
**Figure 9: Plagiarism detected. This is the exact same Figure as Vousdoukas et al. (2018)'s**

*No plagiarism at all. It is the format of the figures that is similar, while the information is different.*
*Vousdoukas et al (2017) describes the results for the whole European coastline and some areas of the Mediterranean, and Vousdoukas et al (2018) describes the global 100y-ESL. Figs. 8 and 9 provide the results for the Venetian coastline (see line 393 of our manuscript). In order to stress the differences we have further restricted the area to the box from lon 12.1°W to 12.9°W; and from lat 43.8°N and 45.8°N. The new figures are here below and are clearly different from the previously published results. We will clearly explain this in the revised text*

[Figure]

*Figure 9 Break-down of projected 100y-ESL contributions in the North-West Adriatic Sea and of their uncertainty, under RCP4.5 (a, c, e) and RCP8.5 (b, d, f). Projected increase of the 100y-ESL from changes in climate extremes, the high tide water level, as well as from SLR contributions from Antarctica, land-water, Greenland, glaciers, dynamic sea level (DSL), glacial isostatic adjustment (GIA), and steric-effects (a, b); variance (in m2) in components (c, d) and fraction of components' variance in global 100y-ESL change. Colors represent different components as in the legend and values express the median at the Venetian coastline.*

[Figure]

*Figure 8 Time evolution of the 100y-ESL in the North-West Adriatic Sea under RCP4.5 (blue) and RCP8.5 (red). Lines show the corresponding medians and colored areas express the 5th-95th percentiles (very likely range).*

**Fig. 5. Moreover, it is not clear how this Figure's results correlate with local or even regional projections of mean Sea Level Rise focused on the northern Adriatic.**

*The title of the subsection where fig.5 is used is "Characteristics of cyclones producing storm surges and floods of Venice". This title should already make clear that this figure is not relevant for projections of mean sea level (which is further not the object on this review article). We will extend the information in our manuscript to better explain how this figure has been obtained and what is its meaning (see our answer to the comment of the Reviewer on Lines 276-282)*

**Figure A.1: Plagiarism detected. This is the same Figure as Barriopedro et al. (2010)' Fig. 2. It is also not clear how these results relate to Venice city center floods.**

*Also in this case there is no plagiarism at all. First, Fig. A1 is based on the residual daily series of MSL (used for the identification of surge events, as described in L276-282), while Fig. 2 in Barriopedro et al. (2010) uses the seasonal frequency of high surge events. An updated version of the latter is provided in Fig. A2. However, Fig. A2 uses data for 1924-2018, therefore providing an assessment over a longer period than that analyzed in Barriopedro et al. (2010). Fig. A2 also allows identifying periodicities at much lower frequencies than in Barriopedro et al. (2010). The results of Fig. A2 over the 1948-2008 period are similar to those reported therein, supporting that the 99.5th percentile of the daily residual series of MSL captures well the high surge events defined in Barriopedro et al. (2010) from hourly data. For the new examined interval (not addressed in Barriopedro et al. 2010), we do not identify significant and robust decadal periodicities in support of an obvious 11-yr solar cycle effect in the frequency of surge events (in agreement with Figure 7).We openly refer to the previous stu*dy in the text and explain that we are repeating the same analysis using a longer time series. The allegation of plagiarism is f*alse.*

**Table 1: This is an interesting feature. Percentages of contribution by each component to the total RSL would be beneficial to the reader for supervisory purposes. Is that new knowledge or reported elsewhere? It should be clarified.**

*This table is a new compilation computed explicitly for this review article. We agree to integrate it adding percentages of the different contributions. The second part with this information is the following:*

| Percentage | | | | | | | | |
|---|---|---|---|---|---|---|---|---|
| 1936-04-16 20:35:00 | 147 | 14 | 10 | 43 | 1 | 18 | 7 | 7 |
| 1951-11-12 07:05:00 | 151 | 28 | 1 | 30 | 2 | 26 | 4 | 10 |
| 1960-10-15 06:55:00 | 145 | 21 | 3 | 44 | 2 | 8 | 8 | 14 |
| 1966-11-04 17:00:00 | 194 | -6 | 11 | 56 | 8 | 10 | 10 | 11 |
| 1968-11-03 06:30:00 | 144 | 23 | 7 | 32 | 2 | 15 | 6 | 15 |
| 1979-02-17 00:15:00 | 140 | 24 | -2 | 28 | 6 | 18 | 11 | 16 |
| 1979-12-22 08:10:00 | 166 | 10 | 8 | 48 | 9 | 8 | 2 | 13 |
| 1986-02-01 03:00:00 | 159 | 18 | 14 | 30 | 3 | 11 | 9 | 14 |
| 1992-12-08 09:10:00 | 142 | 30 | 6 | 21 | 1 | 24 | 1 | 16 |
| 2000-11-06 19:35:00 | 144 | 11 | 4 | 49 | 1 | 12 | 6 | 18 |
| 2002-11-16 08:45:00 | 147 | 29 | -5 | 33 | 1 | 14 | 10 | 18 |
| 2008-12-01 09:45:00 | 156 | 25 | 14 | 27 | 1 | 12 | 3 | 19 |
| 2009-12-23 04:05:00 | 143 | 15 | 22 | 16 | 3 | 12 | 10 | 22 |
| 2009-12-25 03:00:00 | 145 | 20 | 15 | 15 | 2 | 14 | 12 | 21 |
| 2010-12-24 00:40:00 | 144 | 24 | 1 | 26 | 3 | 16 | 8 | 22 |
| 2012-11-01 00:40:00 | 143 | 14 | 1 | 38 | 1 | 19 | 6 | 22 |
| 2012-11-11 08:25:00 | 149 | 32 | -3 | 42 | 1 | 1 | 5 | 21 |
| 2013-02-11 23:05:00 | 143 | 27 | 10 | 27 | 0 | 4 | 10 | 22 |
| 2018-10-29 13:40:00 | 156 | 16 | 1 | 32 | 8 | 19 | 3 | 22 |
| 2018-10-29 19:25:00 | 148 | -22 | 16 | 51 | 9 | 20 | 3 | 23 |
| 2019-11-12 21:50:00 | 189 | 19 | 3 | 22 | 20 | 11 | 7 | 18 |
| 2019-11-13 08:30:00 | 144 | 33 | 3 | 10 | 5 | 16 | 10 | 24 |
| 2019-11-15 10:35:00 | 154 | 31 | 3 | 16 | 1 | 18 | 10 | 22 |
| 2019-11-17 12:10:00 | 150 | 23 | 0 | 23 | 7 | 15 | 10 | 23 |
| 2019-12-23 08:45:00 | 144 | 26 | 27 | 5 | 1 | 10 | 8 | 24 |
| AVERAGE | | 19 | 7 | 31 | 4 | 14 | 7 | 18 |

*We thank the Reviewer for the **Editorial Comments** and we will introduce the required corrections and clarifications in the revised version of our manuscript*

---

## Author Comment (AC2) · 18 Mar 2021

**Author responses to Reviewer#1 comments for the manuscript:**
**"Extremes floods of Venice: characteristics, dynamics, past and future evolution"**

**The paper touches an interesting and relevant topic because of the recent flooding events in Venice and the start of the operation of the Mose barriers. In general terms I think the paper would convey a stronger message provided that:**

*We thank the Reviewer#1 for providing comments and suggestions that will help us to improve the manuscript. Reviewer's comments are in bold characters, authors' responses and proposed changes to the manuscript are in slant characters.*

**a. It emphasizes what is actually new in the paper and the advances with respect to previous publications and the state of the art.**

*This is a review article and, indeed, the manuscript is classified as a "Review article" in the journal submission system. We thought this was however clear considering the initial sentence of the manuscript "This paper reviews current understanding on the extreme water levels that are responsible for the damaging floods affecting the Venice city center ...". Further, the description of this special issue, which is available in the journal web page, clearly writes that "This special issue is composed of three review papers, addressing three different and complementary aspects of the hazards causing the flood of Venice. Review paper 1 describes the tools [...] Review paper 2 describes the factors leading to extreme events, their past evolution, and expected future levels under a climate change perspective. Review paper 3 considers the evolution of the mean relative sea level [...]". We apologize for having missed "Review article:" in the manuscript title and we are sorry for the following confusion. In order to further emphasize that this is a review article and avoid any misunderstanding, we agree that this information should be added to the revised title.*

*Further, the article complements the discussion of the literature with updates of previous analysis using new datasets (ERA5 in Figs. 2-4) and longer time series that have been made recently available (Figs 5-7), and by extracting information specific for the Venetian floods from recent global datasets (Figs. 8 and 9). We stress that all these figures have never been published before and are based on new data. The assessment of the scientific literature, complemented by the analysis of the most recent events, demonstrated that the superposition of several different factors is fundamental for the occurrence of the extreme sea levels in Venice This was never highlighted in the previous literature and constitutes a major conclusion of our article.*

**b. It discusses the various contributions to sea level in Venice, distinguishing between internal and external variability.**

*This review is part of a special issue consisting of three complementary reviews. One of them (nhess-2020-351, Zanchettin et al.) considers trends in the relative sea level and the reader is addressed to it for related issues. Anyway, concerning the Reviewer's comment about a distinction between variability external and internal at the Venice lagoon, different considerations are valid depending on the location within it. The position of the city in the central part of the lagoon makes it very marginally affected by internal fluctuations (which would be very important for Chioggia at the southern end of the lagoon). This will be clarified by expanding section 2.3, complemented by the new version of Fig.1, which allows clarifying several specific issues of the sea level extremes in Venice, which have been mentioned by the Reviewers.*

[Figure]

*Figure 1: left panel: bathymetry of the Adriatic Sea with the position of Venice and arrows denoting the directions of the two main wind regimes affecting the North Adriatic. Right panel: morphology of the lagoon of Venice with the three inlets connecting it to the Adriatic Sea, and the position of the city and of Chioggia. The red box (which includes the whole lagoon in its northern part) denotes the area represented by the data in Figs. 8 and 9.*

**c. The same regarding the non-linear and non-stationary effects in the different sea level components.**

*We admit that scientific literature is never  fully explicit  on this issue. Practically, in many past predictions with hydrodynamical models only the meteorological forcing was used and the astronomical tide was either added to the model results to get the actual prediction or subtracted to the observations for model validation. The astronomical tide was in turn predicted using a numerical program based on harmonic analysis. Examples of this approach and of its success are Lionello et al. (2006), Bajo et al. (2007), Mel and Lionello (2014) among many others.  In general the average depth of the north Adriatic (about 35m) compared with the typical amplitude of the combined astronomical tide and storm surges (about 1m) is the basic explanation that have allowed ignoring nonlinear interaction between these components (see also point 3 below)*
*refs*
*Lionello, P., Sanna, A., Elvini, E., & Mufato, R. (2006). A data assimilation procedure for operational prediction of storm surge in the northern Adriatic Sea. Continental shelf research, 26(4), 539-553.*
*Bajo, M., Zampato, L., Umgiesser, G., Cucco, A., & Canestrelli, P. (2007). A finite element operational model for storm surge prediction in Venice. Estuarine, Coastal and Shelf Science, 75(1-2), 236-249.*
*Mel, R., & Lionello, P. (2014). Storm surge ensemble prediction for the city of Venice. Weather and forecasting, 29(4), 1044-1057*

**More specifically, I include some remarks to strengthen the message of the paper:**

1. **When talking about the timing of the surges with respect to the astronomical tide and free oscillations, some further discussion on the possibility to lower high sea level by modifying the resonance period could be interesting. As a matter of fact, part of the literature on how to avoid harbour long wave resonance could be applied in here.**

*Harbor resonances (2-10 minute periods) have never been associated with sea level extremes in Venice. The extreme sea level propagates through the three inlets and are mostly attenuated by the lagoon hydrodynamics.*

*Section 2.3 will be expanded to clarify this issue and the absence of internal resonances. The shallow nature of the lagoon (on average 1.5 meters deep) damps out the possible oscillations very fast. We comment more extensively on this issue in point 5 below.*

*Different coastal defense strategies at the basin scale have an effect on seiches and tides in the perspective of sea level rise (Lionello et al., 2005) and a short note on this is present in section 4.2 of the manuscript. Lionello et al. (2005) have considered two opposite and extreme strategies to contrast sea level rise: a full compensation strategy preserving the present coastline by dams and a no compensation strategy allowing the free inland expansion of the sea. The former strategy shortens the resonant period moving it away from the period of tides, whose amplitude would be reduced as well as surge heights. The latter strategy, on the contrary, would increase the tidal range and the surge heights.*

2. **When presenting (section 2.1) the astronomic tides and other components, it would be nice to relate the maximum tidal amplitudes at the Northern shore of the basin to antinodes.**

*We assume that the Reviewer means the antinodes of seiches. In fact, the period of seiches and of tides is close to resonance in the Adriatic Sea and it will be affected by climate change (Lionello et al., 2005). A note about this will be added to section 2.1 when describing astronomical tides and seiches and to section 4.2 (see our answer to comment 1)*

3. **When presenting (section 2.2) the different contributions to relative extreme sea level the paper would benefit from a more in depth discussion regarding the non-linear feeding between the different components. In particular discussing the cases of constructive versus distractive interference.**

*It is well known that tidal and non-tidal components have a certain degree of interaction in shallow water areas with large tidal excursions where shoaling and other non-linear effects are significant (Horsburgh and Wilson, 2007, and references therein). Non-linear interactions of the tide and non-tidal residuals were recently investigated at a global scale by Arns et al. (2020) highlighting a small negative, but small, effect of tide-surge interaction on extreme sea levels in the northern Adriatic Sea. However, in the northern Adriatic Sea, given the relatively small importance of tidal excursions (about 1 m) compared to the local water depth (average depth of about 35 m), it is reasonable to neglect the effect of tides on the surge propagation. Such an assumption has been confirmed by several high-resolution numerical studies demonstrating that tide-surge interactions are negligible, even during the most severe events (Roland et al., 2009; Cavaleri et al., 2019), besides those already mentioned in our answer to point c.*

4. **When presenting (section 2.2) the RSL peaks with at least 1.40m that occurred in November 2019 on 12, 13, 15, 17 I think this deserves a further discussion on the possibility of a 2-day resonance period.**

*No relevant resonance has occurred in this sequence of high water level events. To clarify this, the following text will be added in the 4th paragraph of section 2.2: "After the exceptionally high water on 12 November, three successive events with water level values higher than 140 cm occurred in just five days. As reported in Ferrarin et al. (2021), these events were driven by three separate Sirocco wind episodes in the Adriatic Sea, which did not trigger any significant seiche oscillations in the Adriatic Sea. Similarly to what happened on 12 November, these flood events were determined by the overlapping of the maximum meteorological contribution, the tide peak and a persistent high monthly mean sea level in the northern Adriatic"*

5. **When discussing the propagation of the sea level signal into the interior of the lagoon (section 2.3) it appears that the surge signal propagates nearly without damping while other sea level components appear to experience significant damping. This should be discussed more in depth and explained in "physical" terms.**

*Long period oscillations do effectively propagate undisturbed into the lagoon, while the short ones are dumped and sligh amplification occurs in the intermediate range (centered around 5 hours). In the hypothetical case that the inlet depth will be lowered all periods below 12 hours would be heavily damped. Therefore, lowering the depth of the inlets would be a possibility to lower the sea level maxima inside the lagoon. Clearly, there are other implications (e.g., shipping, strong erosion in the inlets) that will make this solution problematic. The figure below shows the amplification factor (in percentage, so that values higher/lower than 100 correspond to amplification/attenuation) of sea level oscillations in the Venice city center with respect to their amplitude at the lagoon inlets, as a function of their period (shown in the x axis, in seconds). In absence of friction a strong resonance would occur in the range from 3 to 11 hours, with a large amplification at about 5 hours (green curve). In reality, friction dampens the resonance and prevents all oscillations below 5 hours to penetrate in the lagoon (violet curve). Drastically reducing the depth of the inlets to 6 m would extend such attenuation up to approximately 12hours (light blue curve). The figure uses output from the model described in Umgiesser et al. (2004) (already cited in the manuscript)*

[Figure]

6. **When presenting the evolution of mean sea level pressure fields during intense surge events (section 3.1) further discussion on the cause-effect relationship between the position of the low pressure centre and the pressure gradient could provide an alternative to numerical model predictions and also advance the understanding on these phenomena.**

*We agree that the pressure distribution has strong impacts on the intensity of the wind fields, their spatial structure and direction over the Adriatic Sea, which affects substantially which part of the Adriatic coastline is most affected by the storm surge (Međugorac et al., 2018). The first predictions of floods in Venice were indeed based on an autoregressive model based on the values of cross-basin SLP differences (Tomasin and Frassetto, 1979). We will add this information and a comment on the structure of the wind fields in section 3.1. The revised versions of Figs. 2 and 3 including insets with the wind field over the Adriatic Sea are provided below.*
*ref: Tomasin, A., & Frassetto, R. (1979). Cyclogenesis and forecast of dramatic water elevations in Venice. In Elsevier Oceanography Series (Vol. 25, pp. 427-438). Elsevier.*

[Figure]

*Figure 2: The main panels show the composite of SLP fields (in hPa, left color bar) based on ERA5 data associated with storm surges higher than 50 cm in Venice (see Table 1). The insets show the corresponding wind fields over the Adriatic Sea (m/s, right color bar). The time lags chosen for the composites are 36, 24 , 12 hours before and 12, 24 hours after the peak of the event. The green dot shows the location of the city of Venice.*

[Figure]

*Figure 3: Same as figure 2, except it is based on the events in Table 1 with storm surge height lower than 50 cm*

**7.** **When discussing the future evolution of extreme sea levels (section 4.2) some further discussion on the projection of surges should be included. In fact, in a former section, mention is made on the possibility that surges will get smaller while here the possibility of non-significant changes or significant reduction are presented on an equal basis. This evolution of surges should also be related to the projections of storm trajectories and peak intensities in this part of the Mediterranean, since both are expectedly related.**

*The summary final sentences of section 4.1 will be slightly rephrased as: "the amount of current evidence shows that the frequency of floods has clearly progressively increased through time after the mid-twenty century. However, there is no clear indication of a sustained trend at multi-decadal time scales in either the frequency or the severity of extreme meteorological events. The frequency of extreme meteorological events is characterized by a substantial interannual and interdecadal variability, which explains differences among studies that have considered different periods and different thresholds. The long term increase of flood frequency is largely caused by the relative mean sea-level rise (connected to both climatic change and land subsidence, see Zanchettin et al., 2020, in this special issue)". This review considers the frequency of floods using an extremely long time series of nearly 150 years duration of daily sea level - from 1872 to 2018. Theanalysis confirms previous studies that after subtracting the long-term sea level mean, the frequency of extremes has no sustained trends (at multidecadal time scales) in spite of the presence of large fluctuations at multiple time scales.*

*The studies presented in section 4.2 are not fully comparable with each other in that some of them (e.g., Lionello et al., 2017) considered separated contributions from SLR and changes of storminess, while others (e.g., Vousdoukas et al., 2017; Vousdoukas et al., 2018) considered directly the overall change of extreme water levels. Further, Vousdoukas et al. considered the 100-year return values, while Lionello et al. considered annual maxima and 5 and 50-year return values. When considering only the effect of marine storminess results suggest non-significant changes or even a significant reduction of the intensity of future surges, which might amount to up to about 5% for the RCP8.5 emission scenario at the end of the 21st century. All these studies agree on the future increase of relative mean sea level being the dominant factor for changes in extreme sea level events.*
*We are confident that a revised manuscript with this partial rephrasing of sections 4.1 and 4.2 clarifies the point raised by the Reviewer.*

8. **When discussing (section 5) the limited consideration of wave set up so far I think it would be worthwhile to add the role of infra gravity waves both cross shore and edge waves since they can also contribute small variations to mean sea level which may be critical for the resulting flooding damages.**

*Both Reviewers have raised the issue of the importance the wave induced components and we fully agree that a more extended explanation on its lack of relevance for the floods of Venice will be beneficial.*
*The city of Venice is located in the center of a large and shallow lagoon, with an approximate extension of 500 km$^2$ and an average depth of about 1 meter. The lagoon is connected to the Adriatic Sea by three inlets (500-1000 m. wide and from 8 to 17m. deep), through which high water levels propagate from the open sea, along a complex pattern of very shallow areas and canals (from 2 to 20 meters deep) to the city center. The lagoon is separated from the sea by two long (about 25 km in total) narrow (less than 200 m average width) and sandy islands, reinforced with artificial barriers in the most vulnerable parts. The elevation of these islands is such that they separate the lagoon from the open sea also during the most extreme events, with the exception of the 4th November 1966 flood, when they were breached in several points.*
*The floods of Venice do not occur because water overtops coastal barriers or defenses. Therefore, wave run-up and infra-gravity waves and nearshore processes (though certainly relevant along the sea-side front of Lido under some conditions) have never been considered when computing sea level extremes inside the lagoon. Wave set-up at the Adriatic shore has been estimated only during some extreme events (e.g., De Zolt et al., 2006), but not inside the lagoon inlets.*
*The elevation of the natural barriers separating the lagoon from the Adriatic Sea has so far prevented overtopping caused by wave run-up and infragravity-waves, except in the 1966 flood when waves may have contributed to total water levels. In the future this is unlikely to change as barrier islands will continue being protected by coastal defences and maintained by beach nourishment. Hence, waves do not need to be considered. It cannot be excluded that these factors will become relevant in case of extreme sea level rise in the future, but present evidence is that waves do not need to be considered.*

9. **When presenting (lines 430 and following) the projected attenuation of storm surges again it should be discussed whether the projection represents a decreasing trend and relating that to the projected wave conditions.**

*Lionello et al. (2017) show substantial consistency of the projected attenuation of waves and surges. This information will be added in the revised first paragraph of section 4.2*

10. **Finally there are some "typos" that should be corrected (e.g. line 36 or line 265).**
*We agree that the manuscript requires checking for typos and also some marginal rephrasing*

---

## Author Response (AR1)

Dear Editor,

I have uploaded the revised version of the manuscript. On the behalf of all authors, I thank the reviewers for their comments, which have greatly helped to improve the quality of this review and avoid possible misunderstandings on its content. Some paragraphs have been relocated within the text, but none of them has been deleted. In fact, in order to clarify some issues, several new paragraphs have been added. The bibliography format has been modified according to the journal template (but these changes are not marked). Note that the track changes version might not highlight the corrections of all typos.

Concerning your comment:

"**With respect to the effects of resonance, like seiches, your response to remark 1 of reviewer 1 suggests that they are damped out very fast. In the same response, it is suggested that the seiches are modified by different coastal defense strategies. It must be clear in the revised manuscript, how these statements fit together**."

*The seiches that are affected by coastal defense strategies are those of the whole Adriatic Sea (see final paragraph of section 4.2) with periods of about 21 and 11 hours. We have added "Adriatic Sea seiches" in this paragraph to highlight this. Reviewer 1 mentioned harbor resonances (who have typically 2-10 minute periods), eventually representing internal modes of the lagoon. We have added Figure 3 and the new 4rth paragraph to section 2.3 in order to discuss the response of the lagoon to an external periodic sea level forcing. This figure shows that resonances inside the lagoon might occur in the range from 10 to 5 hours, but they and all periods below 12 hours are heavily damped.*

"**With respect to the reviewer's suspicions of plagiarism of figures, I suggest that you briefly mention the relationship to the published papers' figures in the text or captions.** "

*Your suggestion has been adopted in the figure captions.*

"**For Appendix 3, I couldn't find its mentioning in the text**"

*In the revised version, it is mentioned in the last paragraph of section 3.3*

Please, find here below our detailed answers to the reviewers.

Best regards

Piero Lionello

**Author responses to Reviewer#1 comments for the manuscript:**
**"Extremes floods of Venice: characteristics, dynamics, past and future evolution"**

**The paper touches an interesting and relevant topic because of the recent flooding events in Venice and the start of the operation of the Mose barriers. In general terms I think the paper would convey a stronger message provided that:**

*We thank the Reviewer#1 for providing comments and suggestions that have helped us to improve the manuscript. Reviewer's comments are in bold characters, authors' responses are in slant characters, excerpts from the manuscript are in red characters.*

**a. It emphasizes what is actually new in the paper and the advances with respect to previous publications and the state of the art.**

*This is a review article and, indeed, the manuscript is classified as a "Review article" in the journal submission system. This is clarified in the initial sentence of the manuscript "This paper reviews current understanding …" and it is stated in the description of this special issue in the journal web page. We apologize for having missed "Review article:" in the title and we are sorry for the following confusion. In order to avoid any misunderstanding, this information has been added to the title "Extreme floods of Venice: characteristics, dynamics, past and future evolution (review article)"*

*The article complements the discussion of the literature with updates of previous analyses using new datasets (ERA5 in Figures.4-6) and longer time series that have been made recently available (Figs 7-9), and by extracting information specific for the Venetian floods from recent global datasets (Figs. 10 and 11). We stress that all these figures have never been published before. Further, the assessment of the scientific literature, complemented by the analysis of the most recent events, demonstrates that the superposition of several different factors is fundamental for the occurrence of the floods in Venice. This was never highlighted in the previous literature and constitutes a major conclusion of our article.*

**b. It discusses the various contributions to sea level in Venice, distinguishing between internal and external variability.**

*This review is part of a special issue consisting of three complementary reviews. One of them (nhess-2020-351, Zanchettin et al.) considers the trends of the relative sea level and the reader is addressed to it for related issues. Concerning the distinction between variability external and internal at the Venice lagoon, different considerations are valid depending on the location within it. The position of the city in the central part of the lagoon makes it very marginally affected by internal fluctuations (which would be very important for Chioggia at the southern end of the lagoon). The text of section 2.3 has been expanded, it includes a new Figure 3 showing the response of the Lagoon to externally forced oscillations and is complemented by the new version of Fig.1, showing the morphology of the lagoon. We think that this clarifies the specific issues that have been mentioned by Reviewers#1 (see our detailed answers below).*

**c. The same regarding the non-linear and non-stationary effects in the different sea level components.**

*We admit that, to our knowledge, scientific literature is never fully explicit on this issue. Practically, in many past predictions with hydrodynamic models only the meteorological forcing was used and the astronomical tide was either added to the model results to get the actual prediction or subtracted to the observations for model*

*validation. The astronomical tide was in turn predicted using a numerical program based on harmonic analysis. Examples of this approach and of its success are Lionello et al. (2006), Bajo et al. (2007), Mel and Lionello (2014) among many others. In general the average depth of the north Adriatic (about 35m) compared with the typical amplitude of the combined astronomical tide and storm surges (about 1m) is the basic explanation that have allowed ignoring nonlinear interaction between these components (More details in our answer at point 3 below)*

**More specifically, I include some remarks to strengthen the message of the paper:**

1. **When talking about the timing of the surges with respect to the astronomical tide and free oscillations, some further discussion on the possibility to lower high sea level by modifying the resonance period could be interesting. As a matter of fact, part of the literature on how to avoid harbour long wave resonance could be applied in here.**

*Harbor resonances (2-10 minute periods) have never been associated with sea level extremes in Venice. The extreme sea level propagates through the three inlets and are attenuated by the lagoon hydrodynamics. Section 2.3 has been expanded to clarify this issue and the (secondary) role of internal resonances. The friction in the shallow lagoon (on average about 1 meters deep) damps out the possible oscillations very fast. Local considered strategies, such as decreasing the depth of the inlets, would lower the water height maxima inside the lagoon, though with problematic consequences in terms of reduced shipping, water exchange and strong erosion inside the inlets. The added text is*

*"Figure 3 shows the amplification factor (percentage, values higher/lower than 100 correspond to amplification/attenuation) of sea level oscillations in the Venice city centre with respect to their amplitude at the lagoon inlets as a function of their period. This computation is based on the model of Umgiesser et al. (2004). In the present situation long period oscillations (≥24 hours) at the inlets propagate undisturbed into the lagoon, short ones (≤3hours) are very effectively damped and at intermediate periods they reach an amplification maximum of about 120% at 9 hours. Numerical experiments with the same model and no frictions suggest that this effect is caused by the combination of internal resonances occurring in the range from 10 to 5 hours with the strong friction inside the shallow lagoon. In the hypothetical case with very shallow inlets (maximum depth equal to 6 m) all periods below 12 hours are heavily damped. This shows that lowering the depth of the inlets would lower the water height maxima inside the lagoon, though with problematic consequences in terms of reduced shipping, water exchange and strong erosion inside the inlets. A 1 m RSL rise (without any change in the morphology of the lagoon) would amplify the lagoon response, showing the possibility of higher extremes in the future."*

*Different coastal defense strategies at the basin scale have been shown to have an effect on seiches and tides of the Adriatic Sea (not on the dynamics of the lagoon) in the in the perspective of sea level rise (Lionello et al., 2005) and a paragraph (revised with respect to the first version of this manuscript) is present in section 4.2 to discuss this:*

*"While the above paragraphs discuss changes due to climatic and meteorological factors, the future dynamics of tides and surges in response to sea-level rise will also depend on the evolution of the shoreline in the area. As sea levels rise, societies will have to decide whether to protect the coast and maintain the current shoreline (e.g. with coastal dams), or allow shoreline retreat. Previous studies (Lionello, Mufato and Tomasin, 2005) have shown that a protection strategy would reduce the amplitude of tides and storm surges and increase that of Adriatic Sea seiches, while allowing for permanent flooding of the low coastal areas and retreat, would increase the amplitude of the diurnal tide components and storm surges. These effects are small, but not negligible, being about 10% for the diurnal component in the case of 1 m RSL rise."*

2.  **When presenting (section 2.1) the astronomic tides and other components, it would be nice to relate the maximum tidal amplitudes at the Northern shore of the basin to antinodes.**

*We assume that the Reviewer means the antinodes of seiches. In fact, the period of seiches and of tides is close to resonance in the Adriatic Sea and it will be affected by climate change (Lionello et al., 2005). Infact, our description of seiches begins with* "Seiches in the Adriatic are standing waves with a node at the southern boundary of the basin and an antinode at the northern shore" *and added to the description of astronomical tides that* "Both diurnal and semidiurnal components have their maximum amplitude at the northern shore of the basin, in association to antinodes of seiches (see below)."

3.  **When presenting (section 2.2) the different contributions to relative extreme sea level the paper would benefit from a more in depth discussion regarding the non-linear feeding between the different components. In particular discussing the cases of constructive versus distractive interference.**

*To discuss this, a new paragraph at the end of section 2.1 and a new figure in Appendix II have been added to the manuscript. The added text is*

"It is well known that tidal and non-tidal components have a certain degree of interaction in shallow water areas with large tidal excursions where non-linear effects are significant (e.g Horsburgh and Wilson, 2007, and references therein). However, in a recent global scale investigation on the non-linear interactions between the tide and non-tidal residuals (Arns et al., 2020) only a small negative effect on extreme sea levels in the northern Adriatic Sea has been found. In fact, in the northern Adriatic Sea, given the relatively small importance of tidal excursions (about 1 m) compared to the local water depth (average depth of about 35 m), it is reasonable to neglect the effect of tides on the storm surge propagation. Such assumption is confirmed by a long past prediction practice with hydrodynamic models, where only the meteorological forcing was used and the astronomical tide was either added to the model results to get the actual prediction or subtracted to the observations for model validation. Examples of this approach and of its success are Lionello et al. (2006), Bajo et al. (2007), Mel and Lionello (2014) among many others. Such an assumption has been confirmed by several high-resolution numerical studies demonstrating that tide-meteorological surge interactions are small, even during the most severe events (Roland et al., 2009; Cavaleri et al., 2019). An example of such simulations in Appendix II and it shows that nonlinear interactions are lower than 5% at the peak of the water height."

4.  **When presenting (section 2.2) the RSL peaks with at least 1.40m that occurred in November 2019 on 12, 13, 15, 17 I think this deserves a further discussion on the possibility of a 2-day resonance period.**

*No relevant resonance has occurred in this sequence of high water level events. To clarify this, the following text will be added in the 4th paragraph of section 2.2:* "After the exceptionally high water on 12 November, three successive events with water height above 140 cm occurred in just five days. As reported in Ferrarin et al. (2021), these events were driven by three separate Sirocco wind episodes in the Adriatic Sea, which did not trigger any significant seiche. These flood events were determined by the overlapping of the maximum meteorological contribution, the tide peak and a persistent high monthly mean sea level in the northern Adriatic."

5.  **When discussing the propagation of the sea level signal into the interior of the lagoon (section 2.3) it appears that the surge signal propagates nearly without damping while other sea level components**

**appear to experience significant damping. This should be discussed more in depth and explained in "physical" terms.**

*Please, see our answer to point 1, above*

6. **When presenting the evolution of mean sea level pressure fields during intense surge events (section 3.1) further discussion on the cause-effect relationship between the position of the low pressure centre and the pressure gradient could provide an alternative to numerical model predictions and also advance the understanding on these phenomena.**

*We agree that the pressure distribution has strong impacts on the intensity of the wind fields, their spatial structure and direction over the Adriatic Sea, which affects substantially which part of the Adriatic coastline is most affected by the storm surge. The text has been revised to explain this. "In both figures [4 and 5] the pressure minimum is located in the Gulf of Genoa at the peak of the event, but in Figure 4 the cyclone is deeper and the SLP gradient along the Adriatic Sea is larger. These differences have strong impacts on the intensity of the wind fields, their spatial structure and direction in the Adriatic Sea (small panels in Figures 4 and 5), modulating the part of the Adriatic coastline that is most affected by the storm surge (Međugorac et al., 2018). Indeed, the first predictions of floods in Venice were based on an autoregressive model considering as inputs the MSLP cross-basin differences (Tomasin and Frassetto, 1979)." The new version of figures 4 and 5 includes a panel showing the wind fields in the Adriatic Sea.*

7. **When discussing the future evolution of extreme sea levels (section 4.2) some further discussion on the projection of surges should be included. In fact, in a former section, mention is made on the possibility that surges will get smaller while here the possibility of non-significant changes or significant reduction are presented on an equal basis. This evolution of surges should also be related to the projections of storm trajectories and peak intensities in this part of the Mediterranean, since both are expectedly related.**

*We have added sentences explaining that studies on projections "are not fully comparable in that some of them (e.g Lionello et al., 2017) considered separated contributions from RSL rise and changes of meteorological surges, whereas others (e.g. Vousdoukas et al., 2017 and 2018) addressed the overall change of sea level extremes. Further, Vousdoukas et al. considered the 100-year return values, while Lionello et al. (2017) considered annual maxima and 5 and 50-year return values", After this premise, we continue that "Studies assessing only meteorological surges suggest non-significant changes or a significant reduction of their intensity, which might reach about 5% for high emissions at the end of the 21st century (with consistent attenuation also of the wind wave height). This weak climate change signal is consistent with the future prevalent decrease of cyclone intensity and related wind speeds in the Mediterranean region that is suggested by most studies in spite of model-related uncertainty and sub-regional differences (see Reale et al, 2021 for a recent comprehensive update, and Lionello et al., 2008, Zappa et al., 2013, Nissen et al., 2014, Zappa et al. 2015). "*

8. **When discussing (section 5) the limited consideration of wave set up so far I think it would be worthwhile to add the role of infra gravity waves both cross shore and edge waves since they can also contribute small variations to mean sea level which may be critical for the resulting flooding damages.**

*Both Reviewers have raised the issue of the importance the wave induced components and we fully agree that a more extended explanation on its lack of relevance for the floods of Venice will be beneficial.*

*The introduction now describes the local morphology.*

*"The city of Venice is located in the centre of a large and shallow lagoon (Figure 1), covering 500 km2 with an average depth of about 1 m. Water is exchanged between the lagoon and the open sea through three inlets (500-1000 m wide and from 8 to17 m deep) and it propagates to the city centre along a complex pattern of very shallow areas and canals (from 2 to 20 m deep). The lagoon is separated from the sea by two long (about 25 km in total) narrow (less than 200 m average width) sandy barrier islands, reinforced with artificial defences in the most vulnerable parts. The elevation of these islands is such that they are not submerged during the most extreme events, with the exception of the 4th November 1966 flood, when they were breached at several points.". A panel has been added to figure 1 to show the morphology of the Venice lagoon."*

*Section 2.1 contains a new paragraph:*

*"The floods of Venice do not occur because water overtops coastal barriers or defences. In fact, the elevation of the natural barriers separating the lagoon from the Adriatic Sea has so far prevented wave overtopping, with the unique exception (already mentioned in the introduction) of the 1966 flood, when waves may have contributed to increase water height in the lagoon. Therefore, wave run-up and infra-gravity waves and nearshore processes (though certainly relevant along the sea-side front of the barrier islands under some conditions) have never been considered when computing water height extremes inside the lagoon. It cannot be excluded that these factors will become relevant under extreme sea level rise in the future, but present evidence is that waves do not need to be considered for computing the water level in the Venice city centre (Roland et al., 2009), as long as barrier islands continue being protected by coastal defences and maintained by beach nourishment."*

9. **When presenting (lines 430 and following) the projected attenuation of storm surges again it should be discussed whether the projection represents a decreasing trend and relating that to the projected wave conditions.**

*Lionello et al. (2017) show substantial consistency of the projected attenuation of waves and surges. Please, see the text in point 7,* *"Studies assessing only meteorological surges suggest non-significant changes or a significant reduction of their intensity, which might reach about 5% for high emissions at the end of the 21st century (with consistent attenuation also of the wind wave height)."*

10. **Finally there are some "typos" that should be corrected (e.g. line 36 or line 265).**

*The manuscript has been checked for typos and many sentences have been rephrased.*

Author responses to Reviewer#2 comments for the manuscript:
"Extremes floods of Venice: characteristics, dynamics, past and future evolution"

**The preprint paper (#nhess-2020-359), submitted to EGU's journal Natural Hazards and Earth System Sciences (NHESS), presents an effort to review several aspects of the flood generating mechanism in Venice city center, i.e., by superposition of astronomical tides, seiches, storm surges, meteotsunamis, etc. All these factors are reviewed in terms of individually and/or cooperatively contributing to intense and/or extreme sea level elevation events, respectively. The main outcome of the review focuses on the following findings:**

**Extreme sea level events are mostly related to storm surges due to Sirocco winds, revealing a characteristic seasonal cycle, with the most important events occurring from November to March. The most intense historical events have been produced by western Mediterranean cyclogenesis, e.g., the Gulf of Genoa. Only a few extreme events of sea levels are caused by atmospheric circulation patterns deriving from the Euro-Atlantic sector. Tidal effects of the 11-year solar cycles appear to mildly contribute to sea level extremes, hidden by the rest of the factors. Relative sea level rise seems to drive a frequency increase of extreme sea levels 1850. Consecutively, it is assessed that the intensity and duration of flood events on Venice in the 21st century, will be affected by possible regional mean sea level rise (MSLR) equalizing and even overcoming the probable enfeeblement of extremes due to the projected storminess attenuation until 2100. High uncertainty of the evolution of global scale factors inducing MSLR, such as mass contributions in the Mediterranean due to Antarctica and Greenland ice melting, does not help in robustness of future projections. Extreme value analysis based on RCP climate projections provides estimations of increase up to 65% and 160% in 2050 and 2100, respectively, for the 100-year return level events at the North Adriatic coast. Geological and geotechnical factors, such as local subsidence due to tectonics or coastal aquifer drainage or overexploitation, are not discussed at all in terms of future increase of extreme flooding.**

**This is an interesting overall review endeavor of a very significant scientific and social issue with particular local interest, but the paper in its current form does not support scientific innovation, as it does not add new knowledge of permanent value on the subject of coastal flooding in general; it presents only a few new insights on previous findings for the Venice study area. The paper mainly recapitulates and tries to interblend existing knowledge from very remarkable past articles, with a specialized focus on certain aspects of the presented problem, by world experts on the field. Yet, in its current form, it does not build robust new arguments on the investigated subject. I believe that if the Editors should consider its publication, at least a major revision should take place, rewriting most of its parts supported by novelty aspects and fresh findings. Some graphs should be omitted (as they are reported elsewhere or refer to previously published literature) and new methodologies of interconnecting the existing knowledge should be proposed and applied. Moreover, some clarifications on the followed approaches are also deserved.**

**In the following, I present my major comments and some specific remarks in tandem with editorial changes and spellcheck needed.**

*Reviewer#1 has provided comments and suggestions that have helped us to improve the manuscript and highlighted some potential misunderstandings that is important to avoid. Comment at Line 24 asking for a revised terminology has been particularly appreciated and has led to rephrasing of several parts of our manuscript. Reviewer's comments are in bold characters, authors' responses are in slant characters, excerpts from the manuscript are in red characters.*

**Major Comments:**

**1) The paper is actually a full review of all the met-ocean physical parameters and mechanisms contributing to high sea levels and eventually the generation of Venice floods. This should be clearly stated in the Title. This is not a Research Article.**

*Indeed, this is a review article. The manuscript is classified as a "Review article" in the journal submission system. This is clarified in the initial sentence of the manuscript "This paper reviews current understanding …" and it is also stated in the description of this special issue in the journal web page. We apologize for having missed "Review article:" in the manuscript title and we are sorry for the following confusion. In order to avoid any misunderstanding, this information has been added to the title "Extreme floods of Venice: characteristics, dynamics, past and future evolution (review article)"*

**2) Flooding phenomena are by default considered as extreme events in literature, yet the authors present proper analysis based on univariate extreme value theory only for the storm surges and not all the other components of sea level variations. This perspective undermines the notion of a compound event. Moreover, the wave-induced component, i.e., the run-up, adding to the total sea-surface height, especially near the coastal front, is left out. Of course, its influence is limited to areas near the waterfront, whereas all the other components (surges, RSLR, etc.) can cause spatially extended inundation, yet all the above need to be discussed and explained to the reader.**

*A main novelty resulting from this review is the importance of the superposition of several different factors for understanding extreme water heights. The different factors are extensively described in section 2.1 and their relevance is assessed in section 2.2 (see also the new version of Table 1 including an evaluation of their percentage contribution to the total water height). However, this is a review and NOT a research paper and being based on existing literature cannot include a multivariate probabilistic analysis, as this is not available in published papers. We anticipate that there is no major river with its exit inside the lagoon and therefore compound events resulting from fluvial and coastal floods should not be considered. We agree that a multivariate approach is important and we have added in the first paragraph of the conclusions that "Furthermore, a multivariate statistical model that describes extreme water levels as a function of the various contributions would provide a more complete characterization of extreme water levels.". However, to perform such an analysis is beyond the scope of this review paper.*

*The comment of the reviewer on the role of waves shows that, indeed, a more extended explanation on the lack of relevance of this component for the floods of Venice will be beneficial for the paper.*

*The introduction now describes the local morphology.*

*"The city of Venice is located in the centre of a large and shallow lagoon (Figure 1), covering 500 km2 with an average depth of about 1 m. Water is exchanged between the lagoon and the open sea through three inlets (500-1000 m wide and from 8 to17 m deep) and it propagates to the city centre along a complex pattern of very shallow areas and canals (from 2 to 20 m deep). The lagoon is separated from the sea by two long (about 25 km in total) narrow (less than 200 m average width) sandy barrier islands, reinforced with artificial defences in the most vulnerable parts. The elevation of these islands is such that they are not submerged during the most extreme events, with the exception of the 4th November 1966 flood, when they were breached at several points.". A panel has been added to figure 1 to show the morphology of the Venice lagoon."*

*On this basis, it is clear why run-up is not relevant, as explained in a new paragraph in Section 2.1:*

*"The floods of Venice do not occur because water overtops coastal barriers or defences. In fact, the elevation of the natural barriers separating the lagoon from the Adriatic Sea has so far prevented wave overtopping, with the*

*unique exception (already mentioned in the introduction) of the 1966 flood, when waves may have contributed to increase water height in the lagoon. Therefore, wave run-up and infra-gravity waves and nearshore processes (though certainly relevant along the sea-side front of the barrier islands under some conditions) have never been considered when computing water height extremes inside the lagoon. It cannot be excluded that these factors will become relevant under extreme sea level rise in the future, but present evidence is that waves do not need to be considered for computing the water level in the Venice city centre (Roland et al., 2009), as long as barrier islands continue being protected by coastal defences and maintained by beach nourishment."*

**3) Figure 4: Please elaborate on the storm track algorithm and its previous validation. Does it treat proper identification of secondary lows in the wake of e.g., "Medicanes", as NASA's storm track algorithm to avoid double backing of storm center on itself over the course of 24 hours. Please further discuss the use of storm tracking technique.**

*Section 3.1 contains the following text describing the storm track algorithm: "The tracking scheme partitions the MSLP fields in depressions, which can be considered candidates for independent cyclones, by merging all steepest descent paths leading to the same pressure minimum. Shallow secondary minima with a small area are absorbed in the large nearest system, whose trajectory is computed by associating the location of the low-pressure centres in successive maps within a minimum distance criterion until the system disappears (cyclolysis). In that way, the method detects the formation of cyclones inside the Mediterranean and, at the same time, avoids the inflation in the number of cyclones that would result from considering small, short-lived features as independent systems. This method has been extensively described in previous works (Lionello, Dalan et al. 2002, Reale and Lionello 2013, Lionello, Trigo et al. 2016) and already used in numerous studies assessing the climatology of Mediterranean cyclones, such as Lionello et al., (2016) and Flaounas et al., (2018), in the IMILAST tracking scheme intercomparison analysis (Neu et al., 2013) and in a dedicated study on the synoptic patterns leading to high water levels along the coast of the Mediterranean Sea (Lionello et al., 2019). Readers are addressed to those studies for details."*

*None of the studies mentioned here was meant to analyse medicanes. The capability of this tracking algorithm on detecting small features, such as medicanes, depends on the tuning of the parameter controlling the merging of small secondary systems in large circulation structures. However, we feel that this discussion is not really relevant for this subsection, though it might be considered in future studies in the wake of the analysis of the 19 November 2019 event.*

**4) The wind patterns in the Adriatic are defined as a crucial factor of surge-driven flood dynamics, but no data is presented to back this up. Thus, some kind of wind maps in extreme cases or anything else would help to relate wind set-up to certain flood events.**

*Following this comment of the Reviewer, Figures 4 and 5 have been modified in order to show the wind fields over the Adriatic Sea during the evolution of the floods.*

**5) The geological and geotechnical aspects of Venice floods are totally overlooked, yet the low elevation of terrestrial land in the Venice area is the main factor for inundation, rather than changes in storminess patterns. It is reported in many papers that constant geotectonic land subsidence and potential**

**overexploitation of coastal aquifers may drive sediment settlement and the urban environment's ever-evolving land sinking below MSL.**

*Loss of land level, subsidence and, in general, vertical land motions have been a key factors for increasing the vulnerability of the city in the past century contributing to about 50% of the observed RSL rise. This review article is a part of a special issue where another contribution (Zanchettin et al., 2021, nhess-2020-351) discusses extensively relative sea level rise including estimates of the past and future role of subsidence. However, we think that the role of RSL rise is clearly evidenced in our review. It is described in the 4ᵗʰ paragraph of the introduction, listed among the relevant factors in section 2.1 and 2.2 and its contribution is computed in Table 1, with its role being mentioned again at several points in the review, particularly in section 4.1 and 4.2 for its past and future importance. The conclusions states that "This review confirms the consensus concerning the key control of the frequency and severity of floods in Venice exerted by historic and future RSL rise. Hence, understanding and predicting the future evolution of extreme water heights in Venice depends critically on the availability of RSL rise projections with lower uncertainty than at present…" Finally in the abstract one reads "The historic increase in the frequency of floods since the mid-19ᵗʰ century is explained by relative mean sea-level rise. Analogously, future regional relative mean sea level rise will be the most important driver of increasing duration and intensity of Venice floods through this century, overcompensating for the small projected decrease of marine storminess". However, for a detailed discussion of subsidence and local mean sea level rise the readers are addressed to the companion review paper by Zanchettin et al. (2021).*

**6) Lines 169-176: present a classic methodology for signal processing of timeseries to separate storm surges, PAWs, meteotsunamis and IDAS, but the choice of cut-off frequency seems arbitrary, as the eventual durations of the reported phenomena are not "physically" fixed. These are known to occur at similar frequency bands (overlapping frequencies between several components) and this makes it difficult to discriminate between different phenomena, especially between surges and meteotsunamis. This should be at least discussed in terms of results' robustness.**

*A number of processes contribute to the sea-level variability and their separation enables the extreme sea levels to be interpreted. The physical differences among the different contribution is explained in section 2.1. Obviously we do not deny that the response of sea level to the atmospheric forcing is characterized by a continuous spectrum. However, the adopted methodology is actually rather robust. This is explained in section 2.2:*

*"The contributions of storm surge, PAW surge, meteotsunamis and mesoscale atmospheric variability (MAV), seiches, IDAS variability have been estimated using band-pass digital filters in the time domain assuming Fourier decomposition, following Ferrarin et al., 2021.The procedure is straightforward for seiches and tides, which can be isolated by applying band-pass filters around their known frequencies. The criteria are more complicated when considering the response of sea level to the atmospheric forcing because it is characterized by a continuous spectrum. A general distinction by Holton (2004), based on the different space-time scales of the atmospheric phenomena, considers planetary-scale (order of $10^7$ m), synoptic-scale (order of $10^5$-$10^6$ m) and mesoscale motions (order of $10^4$-$10^5$ m). At the planetary scale Rossby waves move westwards against the eastward zonal flow and are therefore characterized by relatively small speeds (1–10 m/s) and long time scales (from 10 days to 100 days). Synoptic-scale systems (mostly driven by baroclinic instability) tend to move eastwards with the mean flow and are marked by relatively large speeds (typically 10 m/s) and time scales of about a few days. Mesoscale systems (which are topographically forced or are driven by instabilities operating at that scale) have also relatively large speeds, of the order of 10 m/s (Markowsky and Richardson, 2010) and*

*characteristic time scales in the range from 10 minutes to few hours. A 10-day period for the separation between planetary and synoptic scales is supported by the cross-spectral analysis of the 500 hPa geopotential height and sea level for the Adriatic (Orlić, 1983), which show indeed high (low) coherence above (below) this threshold. A 10-hour cut-off period allows to distinguish between synoptic-scale and MAV setup (including Meteotsunamis), as the latter has time scales in the range from the 10- min period of a pure buoyancy oscillation to the 17-hour period of mid-latitude inertial oscillations (Markowsky and Richardson, 2010). On a practical basis, Ferrarin et al. (2021) have used the 10-hour threshold for separating responses to a cyclone moving in an eastward direction above the Mediterranean from a low-pressure meso-scale system travelling in a northwestward direction along the west Adriatic coast in their analysis of the 12 November 2019 even. The separation between PAW surges and IDAS variability was achieved by applying a low pass filter with the cut-off period placed at 120 days."*

*Therefore, while we agree with the Reviewer that establishing fixed thresholds is not possible, the adopted values allow for an effective separation of the different components in the northern Adriatic Sea.*

**7) Lines 183-185: Are the authors sure that these are separate events? Which is the methodology of discrimination used? Defining the same event (with several peaks) as multiple cases may insert bias to the statistics of extremes.**

*The following comment has been added to section 2.2: "*After the exceptionally high water on 12 November, three successive events with water height above 140 cm occurred in just five days. As reported in Ferrarin et al. (2021), these events were driven by three separate Sirocco wind episodes in the Adriatic Sea, which did not trigger any significant seiche. These flood events were determined by the overlapping of the maximum meteorological contribution, the tide peak and a persistent high monthly mean sea level in the northern Adriatic".* We agree that the persistent PAW surge and the possibility of storm clustering is short sequences would rise some issue when using these three maxima in extreme values analyses. However, this computation is not within the scope of this paper*

**8) In general, the submitted paper feels more like a report (Figures can be enhanced) or a review more than a new research paper. Therefore, all past data on the reported phenomena should be "sewed" together in a comprehensive narrative with new clear scientific insight on the specifics of coastal inundation in Venice city center.**

*Several figures have been redrawn considering our answers to the comments of the Reviewers. We insist that this is indeed a review article, whose utility is presenting the available knowledge and gaps, repeating past analyses using new datasets and achieving a deep insight by merging the outcomes of published papers. We appreciate the synthesis that the Reviewer #2 provided in the second and third paragraphs of the submitted comments and we feel that it shows the effectiveness of our effort.*

**Specific Comments:**

**Some literature of storm surges, waves, climatology, cyclogenesis, extremes etc. in the Mediterranean could be added to the state-of-the-art:**
**Bengtsson et al. (2006). Storm tracks and climate change. J Clim 9(15): 3518–3543.**

Calafat et al. (2012). Comparison of Mediterranean sea level variability as given by three baroclinic models. J Geophys Res 117, C02009.

Campins et al. (2011). Climatology of Mediterranean cyclones using the ERA-40 dataset. Int J Climatol 31(11): 1596–1614.

Makris et al. (2016). Climate Change Effects on the Marine Characteristics of the Aegean and the Ionian Seas. Ocean Dyn, 66(12): 1603–1635.

Fernández-Montblanc et al. (2019). Towards robust pan-European storm surge forecasting. Ocean Mod, 133: 129-144.

*Actually, we think that these references are not really relevant for this review article. Calafat et al (2012) can eventually be relevant for the companion review on sea level rise and Fernández-Montblanc et al. (2019) for that on the prediction models. Bengtsson et al (2006) and Campins et al. (2011) consider storm tracks and cyclones and addressing the related issues would require a dedicated review article. Makris et al (2016) considers a different geographical area.*

**Line 44: No keywords are provided.**

*Proposed key words are: Venice, extreme events, floods, relative sea level rise, surges, climate change, trends*

**Lines 23, 93, 98: The authors refer to planetary waves (e.g., Rossby and Kelvin waves) in the Mediterranean. Maybe this terminology could be avoided to prevent possible misinterpretations. The Kelvin waves' mechanics could be approximately used to interpret small sea-level oscillations induced by large-scale tidal motions in the elongated Adriatic, but classic planetary wave motions usually refer to equatorial Rossby waves in global scale basins, such as the Atlantic Ocean etc., rather than a closed, marginal, regional aquatic body, as the Mediterranean Sea. Planetary waves depend heavily on global thermoclines etc. and their periods of oscillation are of monthly or yearly scales. This is hardly the case in the Mediterranean. The PAWs referred to in Lines 130 and on, are well established motions, but more likely treated as meteorologically driven long waves of fine temporal scales (hours to days) rather than actual planetary waves. The authors themselves do a good job clarifying that in Lines 133-135.**

*A distinction should be made between the planetary oceanic waves (POWs) and the planetary atmospheric waves (PAWs). We have documented the response of Adriatic sea level to the forcing provided by PAWs, whose wavelength ranges from 6000 to 8000 km, characterized by small speeds (1–10 m/s) and long time scales (from 10 days to 100 days). POWs in the Mediterranean and Adriatic Seas are not mentioned at all. The revised manuscript is very clear on this (see sections 2.1 and 2.2). The use of the PAW terminology is established in the literature and we do not think this is a source of confusion.*

**Line 24 and elsewhere in the text: The authors use the term Sea Level Anomaly (SLA) for any sea level variation investigated in the paper. However, in literature, the SLA term usually refers to large-scale long-term (even to climatological scales of analysis) deviations of the Mean Sea Level (MSL) from earth's geoid, not the episodic, short-term, meteorologically induced, coastal sea level elevations that the paper mainly discusses. According to NOAA, "A sea level anomaly reveals the regional extent of anomalous water levels in the ocean that occurs when the 5-month running average of the interannual variation is at least 0.1 meters**

**(4 inches) greater than or less than the long-term trend. The interannual variation is the monthly MSL after the trend and the average seasonal cycle are removed. The anomalies are usually mapped by month, using the mid-point of the 5-month running average. When the 5-month average is more than 0.1 meters above the trend, it is indicated as a positive anomaly...". Thus, I would recommend using the term sea surface height or anything similar.**

*We appreciate this comment of the reviewer that has actually produced a rather substantial change in the terminology used in the paper and in its internal consistency. When reviewing the paper, in fact, we have decided to modify our initial approach, maintained in the former open discussion. The terminology is explained in the first paragraph of section 2 and in the related new figure 2 (added to the revised version of the paper):*

*"Extreme floods of Venice are caused by extremes or high-end values of the local instantaneous thickness of the ocean, hereafter called water height. The water height is defined as the difference between the instantaneous sea level and a local reference level, both measured with respect to a fixed reference level (which could be the reference ellipsoid, the geoid, or a geocentric reference frame). In Venice, the local reference level moves vertically, because of land subsidence. The water height and the total thickness of the water column differ by a constant value, which is the depth of the sea bottom with respect to the local reference level (Figure 2). Water height extremes and sea level extremes differ because the latter do not consider the effect of subsidence, which is important in Venice."*

*And at the end of the first paragraph of section 2.1*

*"Local RSL rise is the increase of local sea level relative to the local solid earth surface (Figure 2 and Gregory et al., 2019) and it can be directly estimated by averaging local tide gauge data over a conveniently long period. It is caused by vertical land movements and changes of local MSL. The evolution of the RSL and IDAS in Venice, and of their different contributions is described in Zanchettin et al. (2021) in this special issue. The addition of storm surges, meteotsunamis and PAW surges represents the meteorological surge contribution to the water height anomalies. The combined effect of seiches, astronomical tides and meteorological surge is generically referred to as detrended water height in this manuscript (Figure 2), meaning that variability at seasonal and longer time scale is subtracted."*

**Line 51 and 149: please provide the reference period of determining the RSLR in Venice. It is essential information for determining the robustness of the values presented, depending on the timeframe of continuous observations.**

*The reference is described in the first paragraph of section 2.2: "Since 1919 observations have been referenced to their mean level over the 1884-1909 period (central year 1897), which is the local reference level used for water-height values and is usually called 'Zero Mareografico Punta Salute' (ZMPS)."*

**Lines 276-282: Please elaborate on the methodology used here (RMSD of which parameter, explain k-means analysis, etc.). Are the authors sure that these are extreme events? Is the analysis based on some robust EVA method? Moreover, please explain how this correlates with the Venice flood events? It seems more of a cyclogenesis-surge association.**

*The analysis explains the peculiarity of the cyclones that generate extreme events (only the uppermost $0.5^{th}$ percentile of the detrended water heights is considered). The analysis is based on the standardized anomalies of*

*MSLP over the Euro-Atlantic sector and 10-m wind vectors over the Mediterranean Sea. The text with the explanation required by the reviewer is in the last paragraph of section 3.1:*

*"The peculiarity of cyclones triggering storm surges is also evidenced from a cluster analysis of the daily atmospheric fields associated to the peaks above the 99.5th percentile of the daily mean detrended water height obtained with a 6-month high pass filter (Figure 7). Only peaks that are separated by at least 3 days are considered to ensure the selection of independent extreme events. To ensure a large sampling size, the analysis uses the NCEP/NCAR reanalysis data for the 1948-2018 period (Kalnay et al., 1996). A k-means clustering (e.g. Wilks 2006) of the standardized anomalies of MSLP over the Euro-Atlantic sector and 10-m wind vectors over the Mediterranean Sea has been applied to group events with similar spatial patterns. Clusters are constructed so that differences between the daily patterns are minimized within the same cluster and maximized between the clusters, using the sum of squared distances as metric. Each cluster is characterized by its centroid (the composited spatial pattern of MSLP and 10-m wind standardized anomalies for all days in the cluster). The root mean squared difference (RMSD) between the daily standardized fields of MSLP and 10-m wind vector of all considered events and their corresponding centroid measures the total spread of the partition."*

**Line 334-337: these statements seem like speculations, not numerical facts. Please elaborate or rephrase.**

*Our analyses and our sentence do not demonstrate or reject a solar influence. The statement relies on the new evidence provided in the manuscript (updated assessment of Figure 9). The correlation between solar activity and the frequency of autumn surges reported elsewhere for the late 20th century is herein captured, but it is lost when we use a longer record, and recent measurements for the 21st century. This suggests that solar influences are non-stationary (effects detected for specific periods only), non-linear (e.g. confined to the recent Grand Modern Maximum), or that the apparent association between the 11-yr solar cycle and surge events was only circumstantial (i.e. caused by other factors). We have rephrased the sentence to emphasize that we are describing possible explanations for the new results:* *"These results suggest that if there is a solar signal it would likely be non-stationary (arguably masked by other sources variability) and/or non-linear (e.g. confined to Grand Maxima of solar activity). The alternative hypothesis is that the decadal variability of extreme surges is due to other causes, including internal variability"*

*As for the final sentence of this section, we do not think that we need numerical facts to support it. It just states that, regardless of the causes of the observed interannual-to-interdecadal variations in the frequency of surge events (solar, internal variability or other), one cannot reject that large variability will also occur in the future.* *"It is plausible that, superimposed on the uncontroversial increasing frequency of Venice flooding due to the RSL rise, the frequency of extreme water heights will experience large interannual-to-decadal variations in the future, as it has been observed in the recent period. However, the causes of this variability are still uncertain."*

**Line 354-358: This is an issue of time-framing, i.e., the choice of the right temporal window to trace statistically significant trends. Moreover, the approach based on stationarity or non-stationarity is also a big issue. Please elaborate and discuss further.**

*This review considers the frequency of floods using an extremely long time series of nearly 150 years duration of daily sea level -- from 1872 to 2018. Its analysis confirms previous studies that after subtracting the long term sea level mean, the frequency of extremes has no sustained trends (at multidecadal time scales) in spite of the presence of large fluctuations at multiple time scales. We have rephrased the sentence as* *"In summary, the*

*amount of current evidence shows that while the frequency of floods has clearly progressively increased in time after the mid-twenty century, there is no clear indication of a sustained trend at multi-decadal time scales in either the frequency or the severity of extreme meteorological surges. The presence of a substantial interannual and interdecadal variability explains differences among studies, which have considered different periods and different thresholds "*

**Line 407: All the presented material is in the form of an inventory of past hard work reported in older papers, but it does not integrate all the datasets together in a composite, coherent way to produce new knowledge on Venice flooding.**

*We insist that this is a review article. Further, the discussion of the literature has been complemented by the replica of previous analysis using new datasets (ERA5 in Figures 4-6), longer time series that have been made recently available (Figures 7-9), extracting information specific for the Venetian floods from recent global datasets (Figures 10 and 11) and performing new numerical experiment (Figure 3). We stress that all these figures have never been published before and are based on new data. The assessment of the scientific literature, complemented by the analysis of the most recent events, it has highlighted that the superposition of several different factors is fundamental for extreme sea levels, which was never highlighted in the previous literature. We further reinforce with longer time series that the increase in the frequency of extreme sea levels since the mid 19th Century is explained by relative sea level rise, with no long term trend in the intensity of the atmospheric forcing. Analogously, future regional relative mean sea level rise will be the most important driver of increasing duration and intensity of Venice floods through this century, overwhelming the small decrease in marine storminess projected during the 21 century. This will clearly pose unprecedented challenges for future flood management and the maintenance of effective coastal defences.*

**Lines 418-422: This analysis seems irrelevant to Venice city center floods and inundation of the surrounding areas. The wave set-up is a surf zone sea-level parameter in nearshore areas, but what would be important for flooding is the wave run-up on the coast. This is a different task to perform as it would require a huge amount of beachfront and coast cross sections treated with several different empirical relations depending on run-up calculation over engineered or natural beach types. Furthermore, as high waves are dissipated by depth-limited breaking and the specifics of the Venice lagoon topography do not allow very high waves to attack the waterfront, wave-induced would not be a crucial factor.**

*We are a bit confused by the suggestion of the reviewer that wave run-up is relevant while wave set up is not, being the latter a component that increases the maximum on-shore elevation reached by the waves. Our answer to comment 2 has clarified why wave run-up is not relevant. The wave set-up (the increase of mean sea level produced by wave breaking) might be relevant if it initiates sufficiently offshore to affect the sea level at the lagoon inlets. The modified text of the paragraph discussing this in the final section 5 is:*

*"The actual effect of wave-set up on the water height inside the Venice lagoon remain uncertain. Some studies have computed it during individual storms affecting Venice (Bertotti and Cavaleri, 1985; Lionello, 1995; De Zolt et al., 2006) and for 100y-ESL projections (Vousdoukas et al., 2016; Vousdoukas et al., 2017) and have shown that the wave set-up contribution at the Adriatic shoreline can exceed 10 cm, but its relevance for the flooding of Venice city centre would require that it initiates sufficiently offshore to affect the sea level at the lagoon inlets.This remains to be investigated."*

**Figure 7: The Figure's results are most likely reported as is in Barriopedro et al. 2010.**

**Figure 8: This kind of information is already reported in Vousdoukas et al. (2017).**

**Figure 9: Plagiarism detected. This is the exact same Figure as Vousdoukas et al. (2018)'s**

**Figure A.1: Plagiarism detected. This is the same Figure as Barriopedro et al. (2010)' Fig. 2. It is also not clear how these results relate to Venice city center floods.**

*None of these figures is a plagiarism. Here we highlight the substantial differences between the figures shown in this review and the figures that are mentioned by Reviewer#2*

*Figure 9 (former figure 7) provides an revisited version of a similar figure in Barriopedro et al. (2010), by considering a much longer series (since 1872) than that employed therein (since 1948), as well as updated data for the last decade (up to 2018, as compared to 2008, in Barriopedro et al. 2010). As stated above (comment on L276-282), our results confirm those reported in Barriopedro et al. (2010) for the second half of the 20th century, but also illustrate new findings (lack of coherence between the frequency of surges and the 11-yr solar cycle during the 21st century and before the mid-20th century).*

*For figures 10 and 11 (formerly 8 and 9) the graphic format of the figures is similar to previous ones, but the provided information is different. Vousdoukas et al (2017) describes the results for the whole European coastline and some areas of the Mediterranean, and Vousdoukas et al (2018) describes the global 100y-ESL. Figs. 10 and 11 provide the results for the Venetian coastline. In order to stress the differences, in the revised version of the paper we have further restricted the area to the box from lon 12.1°W to 2.9°W; and from lat 43.8°N and 45.8°N. The new figures are here below and are clearly different from the previously published results. We will clearly explain this in the revised text.*

*Fig. C1 is based on the detrended daily water heights, while Fig. 2 in Barriopedro et al. (2010) uses the seasonal frequency of large meteorological surges. An updated version of the latter is provided in Fig. C2. However, Fig. C2 uses data for 1924-2018, therefore providing an assessment over a longer period than that analyzed in Barriopedro et al. (2010). Fig. C2 also allows identifying periodicities at much lower frequencies than in Barriopedro et al. (2010). The results of Fig. C2 over the 1948-2008 period are similar to those reported therein, supporting that the 99.5th percentile of the daily detrended water height captures well the events defined in Barriopedro et al. (2010) from hourly data. For the new examined interval (not addressed in Barriopedro et al. 2010), we do not identify significant and robust decadal periodicities in support of an obvious 11-yr solar cycle effect in the frequency of surge events.*

*In all these cases, describing these figures, we openly refer to the previous studies and explain that we are repeating the same analysis using a longer time series or different data. A note has also been added in the revised figure captions. All these allegations of plagiarism are false.*

**Fig. 5. Moreover, it is not clear how this Figure's results correlate with local or even regional projections of mean Sea Level Rise focused on the northern Adriatic.**

*We are confused that the reviewer searches for a link with local or regional sea level projections, which is not within the scope of this review article. The title of the subsection containing Figure 7 (figure 5 in previous version) is "Characteristics of cyclones producing storm surges and floods of Venice''. This figure shows that they can be split in two categories in relation to their position north or south of the Alps.*

**Table 1: This is an interesting feature. Percentages of contribution by each component to the total RSL would be beneficial to the reader for supervisory purposes. Is that new knowledge or reported elsewhere? It should be clarified.**

*Table 1 is a new compilation computed explicitly for this review article. Following the request of reviewer#2 we have integrated it adding percentages of the different contributions.*

**Lines 25-26: correct "by storm surges produced by the inverse barometer effect and enhanced locally by the Sirocco winds".**

*The suggested statement does not quite correspond to the content pf the paper. In section 2.1 the manuscript explains that "… the wind blowing over the shallow water areas over the North Adriatic Sea, whose contribution at the coast is typically 10 times larger than the inverse barometer effect (Bargagli et al., 2002; Conte and Lionello, 2013; Lionello, Conte and Reale, 2019). ". Therefore, the storm surge is produced by the wind and marginally enhanced by the inverse barometer effect. The effect of the wind is not "local" because the effective fetch covers a large fraction of the Adriatic Sea length. The original sentence "The largest extreme water heights are mostly caused by the storm surges produced by the Sirocco winds" is maintained.*

**Lines 50-55: Repetition of "damages and losses of a unique monumental and cultural heritage" expression.**

*This sentence has been deleted in the revised version*

**Line 72: Correct "relevant to extreme sea levels".**

*corrected*

**Line 74: "factors(see" typo.**

*corrected*

**Line 111: "is ceases" typo.**

*corrected*

**Lines 117-126: better use atmospheric pressure than air pressure.**

*Corrected across the whole text*

**Line 162: fix RSL term repetition (use only abbreviation), "...when it exceeds 80 cm"**

*corrected*

**Line 200: change expression to something like "with a rather moderate storm surge value"**

*The sentence has been rephrased*

**Lines 232-233: please rephrase (region is one of the areas…. in the region.. repetition)**

*The sentence has been changed*

**Line 266: correct in the Atlantic   Correct the way of citations' writing in the main text and specifically the use of parentheses e.g., in Lines 78, 251, 258, 268, 304, 340, 352, etc.**

*WE did our best to comply to  the style of the journal*

**Lines 272-275: This sentence is rather vague, please elaborate further on the reported findings of previous studies and how they specifically correlate with Venice city center floods.**

*The sentence has been rephrased: "More recent studies confirm that the position of the cyclone with respect to the basin is critical for storm surges in the north Adriatic, and its variation induces  a veering of the onshore wind and even negative responses in sea level (Lionello, Conte and Reale, 2019)."*

**Lines 295-296: Please rephrase or clarify. Do the authors refer to the synoptic-scale wave train of very low pressures in central Europe and how they drive frequent high autumn surges?**

*We are describing the seasonal mean signatures for active years (i.e. those with at least one extreme event), as compared to quiet years (i.e. autumns with no extreme events). The composite difference of Z1000 resembles a wave train pattern, with alternating positive and negative centers, although statistically significant differences are largely restricted to the negative center over central Europe, which suggests an enhanced occurrence / intensification of cyclones into the Mediterranean during active years. This favorable seasonal mean configuration for the occurrence of surge events does not resemble the NAO. We have rephrased the sentence as follows: "The favourable seasonal pattern for the occurrence of large meteorological surges in autumn displays little resemblance to the NAO, but a negative pressure centre in central Europe, similar to that found in the daily-based composite of Figure 7a"*

**Lines 308-309: Define rho (the Spearman's correlation?) in the main text. In terms of whichparameters?**

*rho (r in the revised version)  is the Spearman's rank correlation between the times series (1948-2018) comprising the seasonal frequency of AL days and surge events. We have rephrased the sentence as follows: "Despite the strong association with AL on daily scales, the Spearman's rank correlation r between the seasonal frequency series of AL days and extreme events is low (r=0.26 for 1948-2018, p<0.05,where p is the significance level) and similar to that obtained from other less influential WRs (e.g. Zonal Regime, r=0.27; p<0.05)."*

**Line 320: correct "how such a small"**

corrected

**Line 344-345: How is the frequency determined? As event/year?**

*Yes, it is the frequency of floods*

**Line 375: maybe use "optimistic" than "low" for RCP2.6**

*We use low (RCP2.6), moderate /RCP4.5) , high (RCP8.5).WE prefer to maintain this terminology*

**Line 378: correct "depend on"**

*Rephrased: "The future variation of amplitude of tides and surges in response to sea-level rise will depend on the adaptation strategy of coastal defences…"*

**Line 390: correct "spatially averaged"**

*corrected*

**Line 387-392: More information about the employed EVA method is needed.**

*The technique is fully explained in Mentaschi et al , 2016. The authors call it Transformed-Stationary (TS) methodology for non-stationary EVA. Their approach consists in first transforming a non-stationary time series into a stationary one to which the stationary EVA theory can be applied; and successively reverse-transforming the result into a nonstationary extreme value distribution. WE actually think that the methodological technical details are not relevant for this review paper*

**Line 863: Explain OND as October-November-December seasonal period.**

*Done*

**Figure 5: Write No. of cases on the graphs before the actual numbers.**

*Done*

**Figure 7: Clette et al. 2014 is missing from the Reference list.**

*Done*

**Line 897: rephrase "Heavy".**

*The caption has been corrected*

**Line 918: correct "As Figure A.1".**

*Corrected*